# Systematic review of the uncertainty of coral reef futures under climate change

**Shannon G. Klein** [1,2,3] ✉, **Cassandra Roch** [1,2,3] & **Carlos M. Duarte** [1,2,3] ✉

Climate change impact syntheses, such as those by the Intergovernmental Panel on Climate Change, consistently assert that limiting global warming to 1.5 °C is unlikely to safeguard most of the world's coral reefs. This prognosis is primarily based on a small subset of available models that apply similar 'excess heat' threshold methodologies. Our systematic review of 79 articles projecting coral reef responses to climate change revealed five main methods. 'Excess heat' models constituted one third (32%) of all studies but attracted a disproportionate share (68%) of citations in the field. Most methods relied on deterministic cause-and-effect rules rather than probabilistic relationships, impeding the field's ability to estimate uncertainty. To synthesize the available projections, we aimed to identify models with comparable outputs. However, divergent choices in model outputs and scenarios limited the analysis to a fraction of available studies. We found substantial discrepancies in the projected impacts, indicating that the subset of articles serving as a basis for climate change syntheses may project more severe consequences than other studies and methodologies. Drawing on insights from other fields, we propose methods to incorporate uncertainty into deterministic modeling approaches and propose a multi-model ensemble approach to generating probabilistic projections for coral reef futures.

Anthropogenic climate change is anticipated to propel large components of the Earth's system beyond critical climate tipping points (CTPs), initiating feedback-driven change and impacts across biophysical systems[1]. These components, known as 'tipping elements,' are distinguished by their significance in Earth's system functioning, their substantial contributions to human well-being, and their unique value[1]. A notable example is the projected dieback of the Amazon rainforest that could release gigatons of carbon into the atmosphere and accelerate global warming[1–3]. Although the concept of CTPs has been subject to debate[4,5], a recent synthesis delivered a shortlist of nine global and seven regional elements at risk[1]. Global tipping elements, such as the Amazon rainforest and West Antarctic Ice Sheet, refer to components spanning subcontinental scales that could alter the operation of

Earth's system[1]. Regional tipping elements represent biospheres expected to exhibit perpetual feedback at confined scales that have the potential to occur synchronously across subcontinental scales, including for example, the simultaneous melting of alpine glaciers[1,6]. Among the shortlisted regional elements at risk are warm-water coral reefs, which are deemed vulnerable to exceedance if global warming surpasses 1.5 °C above preindustrial levels[1,4,7].

Low-latitude reefs, as some of Earth's most biodiverse ecosystems[8], have reached a critical juncture where further deterioration could compromise global food supply, coastline protection, economic revenue, and the livelihoods of up to one billion people[9–11]. Their inclusion as a regional tipping element was based upon historical evidence of near-synchronous coral bleaching events spanning

[1]Marine Science Program, Biological and Environmental Science and Engineering Division (BESE), King Abdullah University of Science and Technology (KAUST), Thuwal 23955-6900, Kingdom of Saudi Arabia. [2]Red Sea Research Center (RSRC), King Abdullah University of Science and Technology (KAUST), Thuwal 23955-6900, Kingdom of Saudi Arabia. [3]Computational Bioscience Research Center (CBRC), King Abdullah University of Science and Technology (KAUST), Thuwal 23955-6900, Kingdom of Saudi Arabia. ✉e-mail: shannon.klein@kaust.edu.sa; carlos.duarte@kaust.edu.sa

>1000 km scales[12,13] and projections indicating progressive degradation of reefs under modest levels of global warming[14–17]. Although coral bleaching is regarded as a localized process, near-synchronous bleaching events on many of the world's reefs have occurred as a result of concomitant increases in ocean temperatures across the tropics[13]. These phenomena are expected to become more frequent, intense, last longer, and affect wider geographic areas with future warming[18–20].

The most recent CTP synthesis followed the same confidence rating system used by the Intergovernmental Panel on Climate (IPCC)[1,21]. It identified a CTP of 1.5 °C (1–2 °C, high confidence) for tropical coral reefs, with an estimated timescale of 10 years for dramatic change (with medium confidence)[1]. In high agreement with findings of the IPCC[22,23], the synthesis cited several modeling efforts using similar 'excess heat' modeling approaches as the basis of the assessment[14–17]. These approaches apply thresholds – in the form of degree heating weeks or months – that represent an accumulation of excess heat above baseline summer conditions. These thresholds are then applied to sea surface temperatures (SSTs) and forced by different emissions scenarios in an effort to retrieve the likelihood of future bleaching events[14–17]. Such models analyze the frequency of bleaching events and estimate the proportion of reef locations at risk of 'long-term degradation' or 'severe bleaching events', producing estimates with high coherence among studies[14–17]. The resulting CTP of 1.5 °C (1–2 °C) places warm-water reefs among the six elements at risk of exceeding their tipping points within the global warming range set by the Paris Agreement (1.5–<2 °C)[1]. This finding aligns with the conclusions of Working Group II's contribution to the IPCC's 6th Assessment Report (AR6)[22] and raises concerns over imminent impacts to marine biodiversity, human livelihoods, and the effectiveness of interventions to alleviate further coral reef degradation.

The earliest studies to project coral reef responses to future global warming utilized 'excess heat' threshold approaches[24–27]. Put simply, these methods operate under the notion that widespread bleaching predictably occurs when temperatures accumulate beyond a specific threshold. While many investigations show that 'excess heat' threshold metrics have strong predictive relationships with bleaching events[12,28,29], others have found these metrics to have weak predictive power when applied to historical bleaching records[24,30,31]. These inconsistencies indicate that the effectiveness of 'excess heat' threshold metrics may depend on the specific context[24]. In the mid to late-2000s, a consensus emerged that differences in bleaching susceptibility between locations were best explained by multiple modifying variables[24], which eventually led to development of alternative model-based approaches. Since then, various approaches, such as species distribution models, ecology-evolutionary models, and models of reef population dynamics have been applied. However, influential syntheses of climate change impacts largely overlook these later developments, relying on projections derived exclusively from 'excess heat' threshold approaches that apply similar assumptions and parameterizations[1,4,22,23,32].

Despite the growing body of literature projecting coral reef futures and their prominent role in assessments of climate change impacts, a comprehensive evaluation of available projections is lacking. Here, we address this requirement by conducting a systematic review of published projections of coral reef futures under climate change in isolation or in combination with other pressures. We first review existing approaches to project coral reef futures and their use in the scientific literature, and then identify key gaps in knowledge that currently contribute to uncertainties. We also disarticulate how lessons from the field of climate change science can provide pathways for improving coordination of modeling efforts toward greater certainty in projections of coral reef futures.

## Results and discussion

### Approaches for projecting coral reef futures

A search of articles in the peer-reviewed literature found 79 studies published between 1999 and 2023 that modeled coral reef responses to future climate change (Supplementary Data 1). While most studies delivered projections for distinct geographical regions (59% of the studies), a considerable proportion offered global-scale predictions (41%) (Supplementary Data 1 & Supplementary Table 1). We found six studies in our literature search that provided projections for individual reef ecosystems[33–38]. However, these studies were excluded to ensure a comparable synthesis with most other assessments at regional and global scales. The majority of articles in our database (76 of 79) could be classified into five broad categories of methodologies: 'excess heat' threshold models, population dynamic models, species distribution models (SDMs), ecological-evolutionary models, and projective meta-analyses of published data.

**'Excess heat' threshold models.** 'Excess heat' threshold models integrate thermal threshold metrics assumed to predict the likelihood of severe coral bleaching with future sea surface temperature (SST) projections to forecast future instances of bleaching events. These models usually adopt a specific frequency of events exceeding the thresholds, such as two severe bleaching events per decade[14,39,40], that is estimated to preclude long-term recovery. This assumption permits the estimation of reef cells (e.g., 0.5° × 0.5° pixels on the Earth's surface) that are at risk of 'long-term degradation'[14,15] or 'severe bleaching events'[41–43], according to the threshold and frequency of events set. Although these models have the advantage of utilizing a method that can be applied to broad geographical scales and incorporate other moderating factors without the need for detailed in situ data, they rarely perform any direct assessments of biological or ecological processes[14,15,44–46]. This approach was the most prevalent model type in our analysis (32%) (Fig. 1a) and attracted a disproportionately higher number of cumulative citations (68%) than all other model types (Fig. 1b). This trend can be partly attributed to this method's dual role as the foundation for satellite products that are used to alert the risk of coral bleaching[47,48], and its widespread adoption as the primary method for global-scale projections in the field (Supplementary Data 1 & Supplementary Table 1).

**Population dynamic models.** Articles examining the consequences of climate change on the dynamics of coral reef populations accounted for 23% of the analyzed studies (18 of 79) (Fig. 1a). Population dynamic models typically employ a process-based approach to simulate the impacts of warming on crucial ecological and biological processes. They consider factors such as coral recruitment[49,50], colony growth[51,52], coral basal mortality[53], predation[52,54], herbivory[55], and the interactions between coral and algal populations[53], including competition for space[55,56]. By incorporating such mechanisms, population dynamic models provide detailed mechanistic frameworks of how coral reef states could change with warming. These models have been used to simulate connectivity between reef ecosystems, by considering alterations in the physical transport of coral larvae and the expected physiological impacts of warming on larvae[49,53,57]. They have also been used to evaluate the efficacy of management strategies, such as increased control of crown-of-thorns starfish (CoTS) and reductions in local nutrient inputs[54,55,58]. However, a significant drawback of these approaches is their reliance on detailed ecological and biological data, which are available only for a few taxa and locations[54]. As a result, the majority of population dynamic studies (17 out of 18) focused on regional geographical scales (Supplementary Data 1 & Supplementary Table 1). Despite accounting for nearly one-quarter of the studies in our database, these models received only 13% of the cumulative citations (Fig. 1b).

**Species distribution models.** Species distribution models (SDMs), also known as niche models, establish correlations between the occurrence or abundance of species and environmental data in geographic space. In turn, they project changes in the distribution of suitable habitat under future environmental conditions[59,60]. Nearly one-quarter of the studies in our database (23%) applied SDMs to forecast the effects of climate change on coral reefs. Despite the equal contribution of SDMs and population dynamic models to our database (Supplementary Data 1 & Supplementary Table 1), they received even fewer cumulative citations than population dynamic models, accounting for <7% of the total cumulative citations (Fig. 1b). The SDMs primarily focused on assessing changes in suitable areas for coral reefs under future climate change scenarios (78% of the SDMs) and accounted for 25% of studies offering global-scale projections (Supplementary Data 1). By identifying the conditions that support historical or present-day coral reefs and simulating future changes in environmental variables, the models project shifts in suitable habitats[61–63]. This approach permits large-scale projections that consider multiple physical parameters, even with limited field sampling[63–65], making SDMs cost-effective tools. Although the widespread adoption of SDMs amplifies their value for comparing a diverse range of responses across marine and terrestrial ecosystems[66,67], a recent systematic review showed that SDMs may have significant limitations in accurately predicting the biology of real-world populations[68].

While climate-related data (e.g., mean SST) are used in SDMs applied to coral reefs, other physical parameters such as light availability, current speed, and water depth can also be included[63,65,69]. The most common physical parameters employed in the SDMs within our database were SST and aragonite saturation[64,65,70,71], which were commonly represented by their means (Supplementary Data 2). By primarily relying on means of physical parameters, such models overlook the well-recognized importance of environmental variability in influencing coral reef responses to climate change, which includes capturing the shapes and distributions of these parameters[72,73]. Another major limitation arises from the assumption that the physical environment alone largely governs the natural distribution of warm-water reefs. This key assumption overlooks key biological and ecological processes, such as the influential role of larval dispersal and retention in shaping reef distributions[74,75], as well as the role of top-down controls in the food web[76–79]. Significantly, the biogeographic approach of inferring past ecology largely disregards the potential for species'

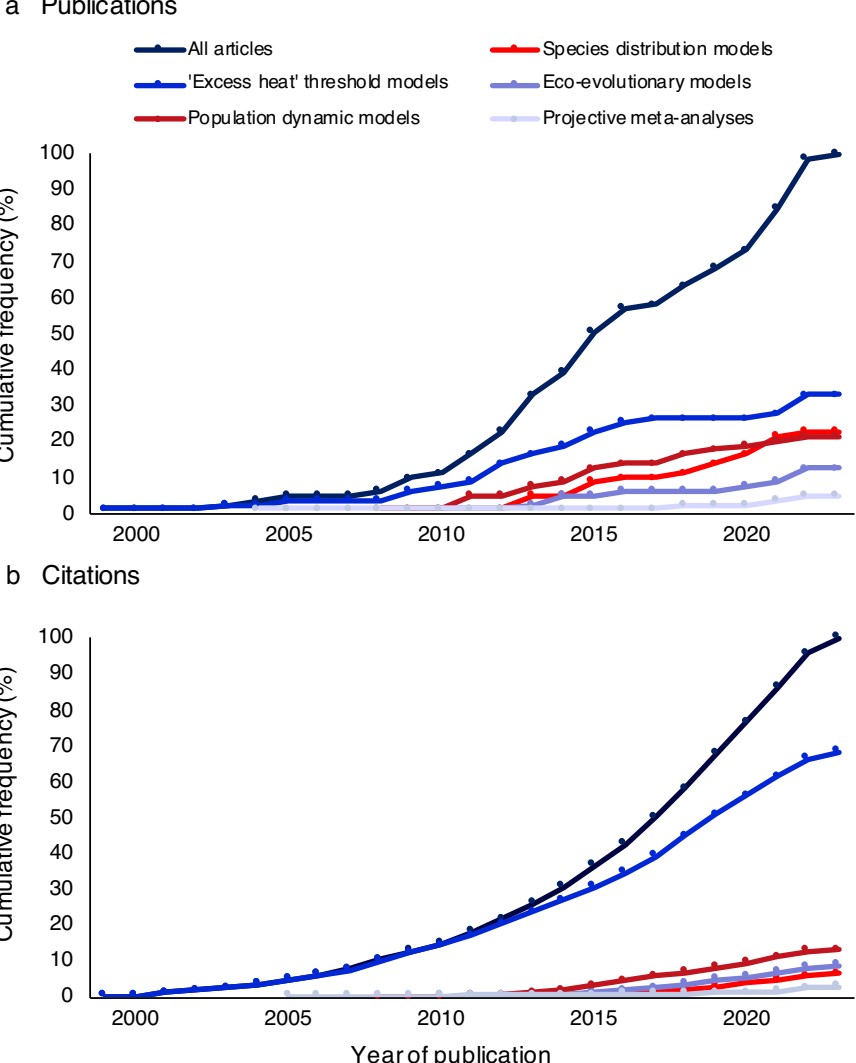

**Fig. 1 | Frequency of published articles and citations across major methodologies.** Cumulative frequency of **a** all published articles (*n* = 79) and articles classified into five broad categories of methodologies, and **b** citations of all published articles and articles classified into the same five categories. Citations were extracted from the Thomson Reuters Web of Science database. Source data are provided as a Source Data file.

niches to evolve through adaptive processes. This limitation can lead to an underestimation of future distributions[80].

**Ecology-evolutionary models.** Simulating the potential role of eco-evolutionary processes in helping coral reefs to adapt to changing ocean conditions has gained attention in this field (12% of the studies) (Fig. 1). Eco-evolutionary models simulate the interplay between ecological dynamics and evolutionary processes in response to changing climatic conditions. The earliest eco-evolutionary models[81,82] (published in 2009 and 2013) examined how heat-tolerant symbionts – the phototrophic component of reef-building corals − could improve coral heat tolerance through changes in the symbiont community and/or evolutionary adaptation. Building upon similar frameworks used in population dynamic models, more recent studies[83–85] incorporate species interactions and their abilities to adapt and disperse across diverse environments. While nearly a third of these studies generated global-scale projections (Supplementary Table 1 & Supplementary Data 1), a significant challenge lies in the requirement for knowledge of taxa-specific traits, genetic adaptation, and ecological dynamics, which is lacking for most coral species and locations. As with population dynamic models, the reliability of their projections for non-focal taxa and regions can be influenced by the parameter estimations and assumptions incorporated[85].

While these studies do not aim to achieve spatial or ecological realism, they provide essential insights into the potential role of adaptation and key environmental drivers that help to inform conservation planning at local and regional scales[85]. For instance, recent eco-evolutionary models reveal the importance of protecting networks of reefs to facilitate the migration of heat-tolerant larvae to cooler waters, thereby facilitating evolutionary adaptation[84–86]. Despite the rising demand for conservation strategies that prioritize the adaptive capacity of coral reefs and a deeper understanding of the underlying mechanisms[87,88], these models attracted a minor proportion of the cumulative citations (Fig. 1b).

**Meta-analyses.** Another approach to project coral reef futures consolidates data from published experimental manipulations. Representing a minority of the reviewed studies (5%) (Fig. 1), the identified meta-analyses[20,89–91] shared a common aim of projecting the dual impacts of ocean warming and acidification on biological processes within reefs. They compile data from experiments that measure how corals and other coral reef taxa respond to conditions that simulate future warming and acidification scenarios. These data are then utilized to parameterize models for estimating future coral responses under various representative concentration pathway (RCP) scenarios. The specific purposes of these meta-analyses range from estimating changes in numerous biological responses of corals[20] to those that exclusively focused on alterations in coral calcification processes[91] or reef-wide calcium carbonate production[89,90]. Although data from coral reef monitoring, rather than controlled experiments, arguably offer more realistic insights into how reefs will respond to further warming, our understanding of how reef organisms will react to ocean acidification is primarily based on manipulative experiments[92,93]. Thus, one significant advantage of these approaches is their capacity to consolidate the wealth of data derived from experiments to estimate how future warming and acidification will interact and impact the biological responses of reef organisms.

By aggregating data from numerous independent studies, meta-analyses can help to resolve discrepancies among experimental designs, locations, and species by uncovering overall patterns across studies[94]. However, the data underlying the projections from meta-analyses originate from short-term experiments[20,89–91], which fail to measure important elements of resilience such as genetic adaptation[20] and the complex ecological feedbacks that operate in natural reef environments[95]. It is important to acknowledge, however, that other model types similarly rely on results from short-term acidification experiments to parameterize their models[14,15,96]. Overall, the cumulative frequency of citations based on these meta-analyses aligned with their rarity in the field, equating to <3% (Fig. 1b).

**Other emerging approaches.** There are several other approaches to examine coral reef vulnerability to future climate change. In addition to the five approaches outlined above, one study adopted a spatial modeling approach to project the combined effects of warming and sea-level rise on the future coral reef growth rates in the South China Sea[97]. Another integrated linear extension rates of corals from three different islands in the same region with future SSTs to forecast coral growth rates[98]. One study used historical bleaching and sea surface temperature records to project future bleaching probabilities in the Indo-Pacific[99], while another regional-scale study focused on larval connectivity and identified conservation areas with lower risks of coral bleaching in the Amani Islands of southern Japan[100].

A standardized method for assessing the risk of ecosystem collapse, the International Union for Conservation of Nature (IUCN) Red List of Ecosystems (RLE), represents an emerging method[101–104]. The RLE offers a standardized classification system that utilizes thresholds for key variables to integrate diverse data[101,105,106]. One study applying this method used various coral reef datasets to model interactions within western Indian Ocean reefs under future warming[104]. The study reported varying levels of regional vulnerability to ecosystem collapse, ranging from 'critically endangered' to 'vulnerable' across the 11 ecoregions examined[104]. However, the ecosystem model's assessment excluded data on fishing pressure and rates of sedimentation, among other variables, due to data scarcity across countries and regions. There are at least four studies applying this method to coral reefs in the Caribbean[101,102], meso-America[103], and the western Indian Ocean[104]. Together, they emphasize the need for improvements in the consistency of monitoring efforts and advocate for the development of a unifying framework to enable more conclusive risk assessments.

In summary, the discussed approaches span a spectrum from simplistic models that minimize complexity to those incorporating detailed mechanistic frameworks that address complex ecological and evolutionary processes. While the latter approaches provide a deeper understanding of the effects of climate change on essential ecological and biological processes in warm-water reefs, their practical utility is constrained by limited data availability.

## How heat stress is modeled

Severe marine heatwaves that trigger mass coral bleaching events are expected to become more intense, frequent, last longer, and affect wider geographical areas as the planet continues to warm[18,20,107]. While the approaches discussed thus far encompass five distinct approaches for forecasting coral reef futures, the underlying procedures for modeling heatwaves and their impacts on reefs can be classified into two overarching techniques (*sensu*[24]). The first technique utilizes thermal stress thresholds, which are defined as metrics requiring a variable, such as SST, to surpass a pre-determined value[24]. For studies to be classified as threshold techniques, the use of these metrics had to form the primary framework of the models that delivered projections. The second technique represents approaches that abandon the central threshold concept to focus on empirical relationships between continuous variables. Articles classified as using this approach could use thermal stress thresholds, however, they had to be included as one of numerous variables examined[24]. Only one study in our database could not be classified as using either technique. The study integrated various data sources, environmental variables, and analytic techniques, including regression and association methods for projecting future coral cover[72].

Our analysis revealed that more than half of all the studies (53%) employed thermal threshold techniques as the primary basis for their

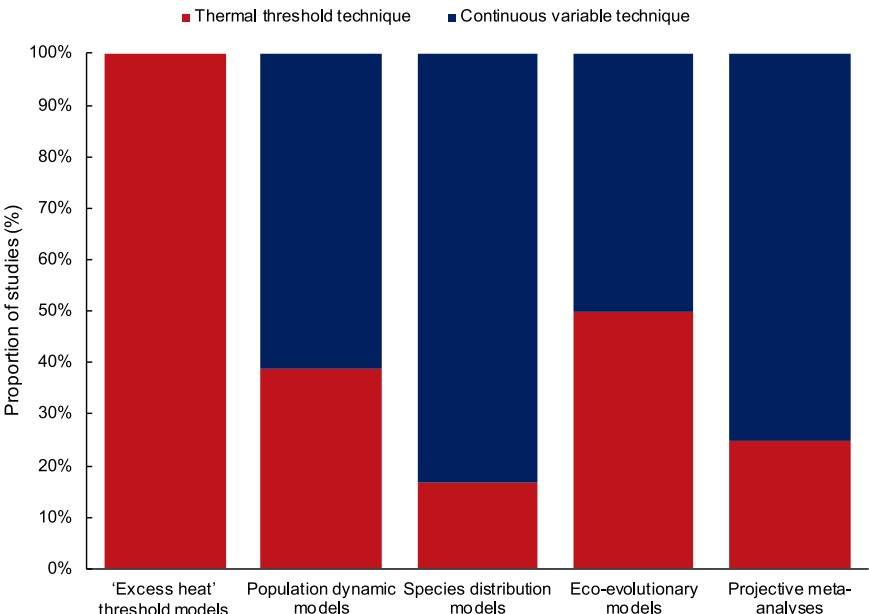

**Fig. 2 | Proportion of studies applying thermal threshold versus continuous variable techniques to model heat stress.** Proportion of studies classified as using either a thermal threshold or continuous variable technique across the five broad categories of methodologies (*n* = 74 articles). Note: one article could not be classified as using either of the threshold or continuous variable techniques[72] and four studies could not be classified as one of five major methodologies. These five studies were excluded to enable a meaningful analysis, but see Supplementary Data 2. Source data are provided as a Source Data file.

projections (Supplementary Data 2). Besides the exclusive use of this method in 'excess heat' threshold models, around 40% of population dynamic and eco-evolutionary models also relied on thresholds as the basis for their projections (Fig. 2). Across the five major approaches (Fig. 2), the most common threshold metrics applied were degree heating weeks (DHWs) or months (DHMs), which calculate values representing both the intensity and duration of heat stress events in singular metrics.

One explanation for variations in the efficacy of thresholds metrics in explaining realized coral bleaching is their inability to capture different marine heatwave characteristics, such as peak temperatures, duration, and rates of heating. For instance, a historical assessment spanning from 1985 to 2017 examined variations in SST and showed that increases in accumulated heat stress, as measured by two common threshold metrics, were predominantly attributed to longer heating events affecting wider areas[108]. However, the study could not detect changes in peak temperatures or event frequencies during the analyzed period, indicating the limitations of the metrics in capturing changes in different heating variables[108]. A study by McClanahan et al.[73] evaluated the effectiveness of heatwave variables in explaining bleaching severity on 226 coral reefs and found that the DHW metric explained 9% of the model variance. In contrast, peak temperatures, the duration of cool temperatures, and temperature bimodality were found to be stronger predictors of bleaching severity. Several empirical studies have also reported that variables representing different marine heatwave characteristics were best at predicting changes in coral cover[72,109,110]. For example, a study investigating the power of 27 environmental factors in explaining changes in coral cover on Indian Ocean reefs reported that temperature anomalies, temperature variation, and the duration of cyclones were the best predictors[109].

We found only one study projecting impacts on coral reefs that directly compared model outputs derived from both thermal threshold and continuous variable techniques[72]. This study revealed that a DHW-based model projected more severe declines in coral cover in the Indian Ocean compared to the multivariate approach that integrated variables characterizing historical and future patterns of stressors. The findings suggested that patterns of acute and chronic stressors could be more influential than cumulative heat stress in predicting future coral cover in certain regions[72], further highlighting the importance of variable selection procedures in the modeling process. Although there is substantial uncertainty in how climate change will morph future thermal regimes, global databases of marine environmental data provide many useful exposure and modifying variables for this purpose[72,111].

While thermal threshold metrics have acknowledged limitations, they remain vital for established programs forecasting coral bleaching risk using satellite-based products. Work has already been done to test how well different degree heating algorithms explain coral bleaching patterns at local, regional, and global scales in an effort to improve their efficacy (e.g. refs. 112–115). New configurations of the operational DHW algorithm hold promise in improving their ability to predict instances of observed bleaching[112,113,116], although the extent to which adapted algorithms improve predictability depends on the focal region and spatial scale of the test. This suggests that researchers could consider adapting different degree heating algorithms to pinpoint the most appropriate stress metric for their geography. Subsequently, these customized algorithms could be confidently applied to projection models for the focal region. Several studies in our database have shown how threshold choice affected their model outputs[26,40,46,112,117,118]. For example, one study reported divergent estimates of bleaching onset timing when different inter-annual variation thresholds were used[117].

Another major consideration is the future efficacy of threshold-based metrics in reliably approximating instances of coral bleaching or changes in other coral reef metrics. This is because most coral reefs have already experienced a complex legacy of exposure to disturbance and it is presently unclear by how much and to what extent organisms have adapted or will adapt in future[119,120]. A recent study examined intrapopulation variability of heat tolerance in corals from the western Pacific Ocean[121]. The study demonstrated that the most heat-tolerant corals in their study required double the heat stress to induce bleaching compared to their least-tolerant corals. When these differences in heat tolerance were translated into contrasting DHW

thresholds and applied to an ambitious emissions scenario (SSP2 −4.5), the study reported that the most heat-tolerant corals could potentially experience annual bleaching events up to 17 years later than their less-tolerant counterparts[121]. Overall, greater confidence in coral reef projections will depend on an increased number of projections derived from methods that incorporate robust variable selection procedures and a deeper understanding of how the thermal tolerances of corals and other coral reef taxa may evolve under escalating stress levels over time.

## Addressing uncertainty through more coordinated modeling efforts

Uncertainties in coral reef projections stem from various sources that compound in the steps involved in generating the projections[122]. The sources range from variations in the climate system that impact various modeling tools, such as General Circulation Models (GCMs)[122,123], to uncertainties in how future socioeconomic policies and technologies will affect future emissions trajectories[124]. Uncertainties related to the models themselves pertain to the model structure and parameter settings used[122,125], which both rely on knowledge of the specific physical and ecological processes affecting how coral reefs will respond in the future.

While the models reviewed here vary in their complexity and underlying methodologies, most rely on deterministic rules to establish cause-and-effect relationships (Supplementary Data 2). Such deterministic models do not directly incorporate uncertainty[126] and are inherently limited in their ability to account for uncertainties stemming from interactions between physical and ecological factors inherent to coral reefs. In contrast, models employing probabilistic relationships can accommodate natural variation and uncertainty in model input values and parameters by considering potential value ranges and associated probabilities[127]. While probabilistic models may be deemed most suitable for capturing uncertainties in how coral reefs will respond in the future[127–129], this field still faces significant issues in establishing robust connections between key coral reef metrics and satellite-derived data[72,73,109,110]. These difficulties ultimately hinder the reliability and utility of probabilistic models.

The question of how to account for uncertainty of deterministic models poses a significant challenge. Uncertainty associated with the model structure, specifically uncertainty about the cause-and-effect relationships, is often difficult to quantify because this requires the comparison of model outputs with real-world observations[127]. However, it is possible to evaluate and then use uncertainty caused by the model's input values and parameters. The probable range of model outputs can be examined by analyzing how these outputs behave when model input values are changed within plausible ranges[127,130]. Some studies have evaluated how choices in model inputs and different assumptions affect the outputs of coral reef models (e.g. refs. 55,72,130–133), though differences in model outputs are seldom used to produce formal estimates of uncertainty arising from model inputs. One approach to doing this is to conduct a formal uncertainty analysis of different model outputs. A straightforward method for conducting such an analysis is by applying Monte Carlo methods, where variations in model inputs are drawn randomly, and the resulting model outputs are treated as a random sample of the model output distribution[127] (e.g. refs. 55,130). Although effective in helping to incorporate uncertainty into deterministic models, this approach requires a substantial number of model runs.

Another approach is to apply a sensitivity analysis – a common method to understand how changes in input values and/or parameters of a model affect its output[127]. Essentially, these analyses aim to pinpoint the input parameters to which the model output is most sensitive. For example, if plausible changes in an input parameter value induce large variations in the model output, this indicates that the parameter value is highly uncertain. Conversely, if model outputs

remain stable, the analysis will indicate that the parameter value has low uncertainty. These analyses can become computationally expensive when all possible parameter values and their interactions are tested in a step-wise manner. However, there are techniques to reduce the number of model runs[127,130]. For instance, sensitivity analyses have been applied to coral reef models by testing only the highest and lowest plausible values of the biological parameters and adjusting single parameter values by ±10%[130,131].

While the model sensitivity analyses described above offer ways to account for uncertainty caused by the model's input values and parameters within an individual study, it is possible to address system and model uncertainty using multiple independent models in an ensemble approach[127]. For instance, in the field of climate change science, atmospheric scientists initially faced issues with fragmented data and disparate deterministic models when modeling the Earth's response to increasing greenhouse gas emissions[134]. By the 1980s, coordinated data collection from weather stations and satellites improved the accuracy of atmospheric-ocean GCM models[135]. By 1988, the IPCC formed and used the Coupled Model Intercomparison Project (CMIP) to coordinate simulations using the same emissions scenarios and model outputs[136]. This ensemble approach combined diverse deterministic model types across research groups to generate reliable probabilistic statements[134]. While acknowledging that climate scientists only model a single system compared to the thousands of interdependent and locally-adapted species comprising coral reefs, adopting a multi-model ensemble approach to generate probabilistic projections for coral reef futures is feasible[123,134,137]. This, in turn, would help to highlight major sources of variation and better characterize the extent of uncertainty of coral reef futures under climate change. However, applying an IPCC ensemble-like approach would initially necessitate improved coordination among modelers and the selection of common output metrics and emission scenarios.

Despite the growing body of studies forecasting coral reef futures, there is presently no broad consensus on the optimal variables for projecting coral reef vulnerability[102–104]. This is reflected in the diversity of variables used and the large proportion of studies delivering projections with metrics that prove challenging to translate to real-world observations (Supplementary Data 1). Establishing a connection between model outputs and real-world observations is not only crucial for enhancing the practicality and usefulness of modeled projections but also enables future assessments of the models' ability to simulate past conditions. More than half of the studies employing 'excess heat' thresholds presented their projections in terms of fractions of reef cells at risk (52%), while SDMs typically provided estimates in terms of fractions of reef cells with suitable habitats or relative changes in habitat suitability (69%) (Supplementary Data 1). Although coral cover serves as a widely used and accessible indicator for this purpose[104,138], projections of coral cover were delivered in less than a third of all published studies in our database (29%).

While coral cover represents the most frequently simulated metric directly linked to real-world observations, its effectiveness as a singular measure is constrained[104]. The simplicity and accessibility it provides comes with trade-offs, as it fails to encompass other crucial aspects of reef health, including changes in community compositions of corals, algae, and other key taxa essential for ecosystem functioning. Transitions in coral communities in the western Indian Ocean and the Great Barrier Reef have marked significant ecological shifts in response to climate change[139,140], highlighting the requirement for coordinated simulations of numerous common reef variables to better capture future coral reef vulnerability. Present coral reef assessment and monitoring efforts, however, suffer from differences in methods and the resulting datasets[104]. Recommendations for unifying frameworks to select common metrics to capture different dimensions of ecosystem integrity and risk of collapse across ecosystems already exist[141].

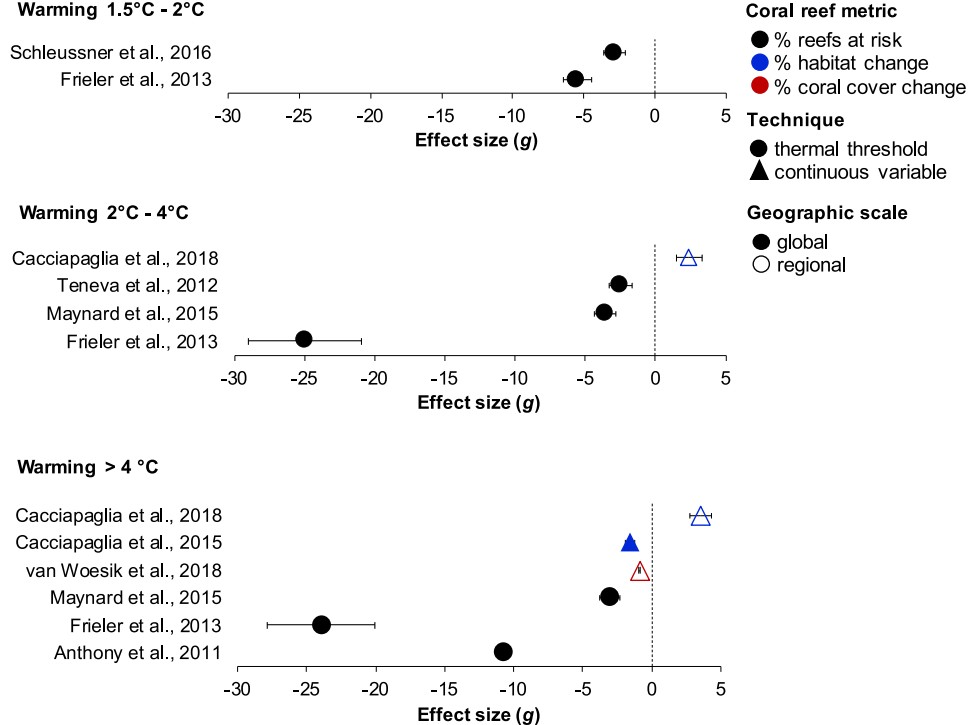

**Fig. 3 | Comparative effect-size analysis of projected impacts on coral reefs among a small subset of available studies and three warming scenarios.** Calculated mean effect sizes (Hedges' $g \pm$ 95% CIs) represent the magnitude of projected impacts on model outputs (i.e., coral reef metrics) across three global warming scenarios (1.5–2 °C, 2–4 °C, and >4 °C). Model outputs (mean $\pm$ 1 Std) used in this analysis were extracted from $n = 39$ individual modeled scenarios across eight published studies, and represented in Fig. 4. Mean effect sizes were derived from differences between projected estimates of coral reef metrics for the end-of-century (2090–2100) and the baseline period (2000–2015) (cf. "Methods" section).

Hedges' g, a common effect-size metric ranging from $-\infty$ to $+\infty$, signifies no impact at zero, positive values indicate ecological benefits, and negative values signify adverse effects. The 95% CIs represent variability among scenarios within each study and warming scenario (Supplementary Data 3). Analyzed coral reef metrics include percent reef cells at risk (black), percent habitat change (blue), and percent coral cover change (red). Circles and triangles denote studies using thermal threshold and continuous variable techniques for modeling heat stress, respectively. Open symbols represent global-scale projections, while closed symbols denote regional-scale projections. Source data are provided as a Source Data file.

However, coordination to select key metrics specific to coral reef ecosystems for this purpose is still lagging. This recommendation is further emphasized by studies utilizing the IUCN RLE classification system to evaluate the risk of coral reef ecosystem collapse, which call for enhanced coordination in monitoring coral reefs and improved data quality and quantity[102–104].

There is currently no formal consensus on the most suitable emissions scenarios for modeling coral reef futures. While the number and type of emissions scenarios varied, the most frequently used scenario in our database was RCP8.5 (CMIP5), representing a high-emission scenario of ~4.5 °C global warming by the end of the 21st century (Supplementary Data 2). Most studies applied two emissions scenarios, typically comparing RCP8.5 (CMIP5) with a scenario of lower radiative forcing such as RCP2.6 or RCP4.5 (CMIP5) (Supplementary Data 2). Though subject to debate, recent analyses show that observed trends in global $CO_2$ emissions are substantially lower than those simulated by high-emission baseline scenarios such as RCP8.5 (CMIP5)[142–145]. These studies suggest that this divergence could widen throughout this century and conclude that such scenarios should no longer serve as reference high-emission scenarios[142–145]. Given these developments and the release of the IPCC's AR6 report[22], there is a pressing need for coordination to select the common emissions pathways for modeling coral reef futures. This urgency is underscored by the introduction of new socioeconomic pathways representing novel levels of radiative forcing (1.9, 3.4, 7.0 W m$^{-2}$), already incorporated into recent projections for coral reefs (e.g. refs. 17,146).

## Comparison with a prevailing diversity in methodologies

A major challenge in synthesizing existing projections stems from the diversity of coral reef metrics simulated and emissions scenarios used. In other fields, meta-analyses have been employed to compile published projections and compare the direction and extent of modeled impacts across studies using diverse metrics[147,148]. These syntheses adopt standardized effect-size metrics such as Hedges' g. Calculated based on relative differences between impacted and baseline (or control) scenarios and weighted for variance, these metrics offer a uniform measure for assessing the magnitude of anticipated effects[147,149]. We focused on the three most commonly projected coral reef metrics (fractions of reef cells at risk, fractions of reef cells deemed habitable, and changes in coral cover). However, due to reporting limitations in most published articles, we could extract requisite data from only 39 modeled scenarios across eight studies (Fig. 3 & Supplementary Data 3). We therefore consider this analysis to be exploratory in nature to encourage future efforts, rather than providing definitive or conclusive results.

Figure 3 illustrates mean effect sizes representing the direction and magnitude of expected impacts on coral reef metrics across the selected studies. The distribution of effect sizes among the studies is influenced by a combination of factors: (1) varying assumptions on key drivers, such as choices in future emissions scenarios, (2) methodological differences, including the choice of simulated coral reef metric and the type and parameterization of the model, and (3) how the results were reported, such as the number of, and agreement among individual scenarios within each study. To help disentangle these

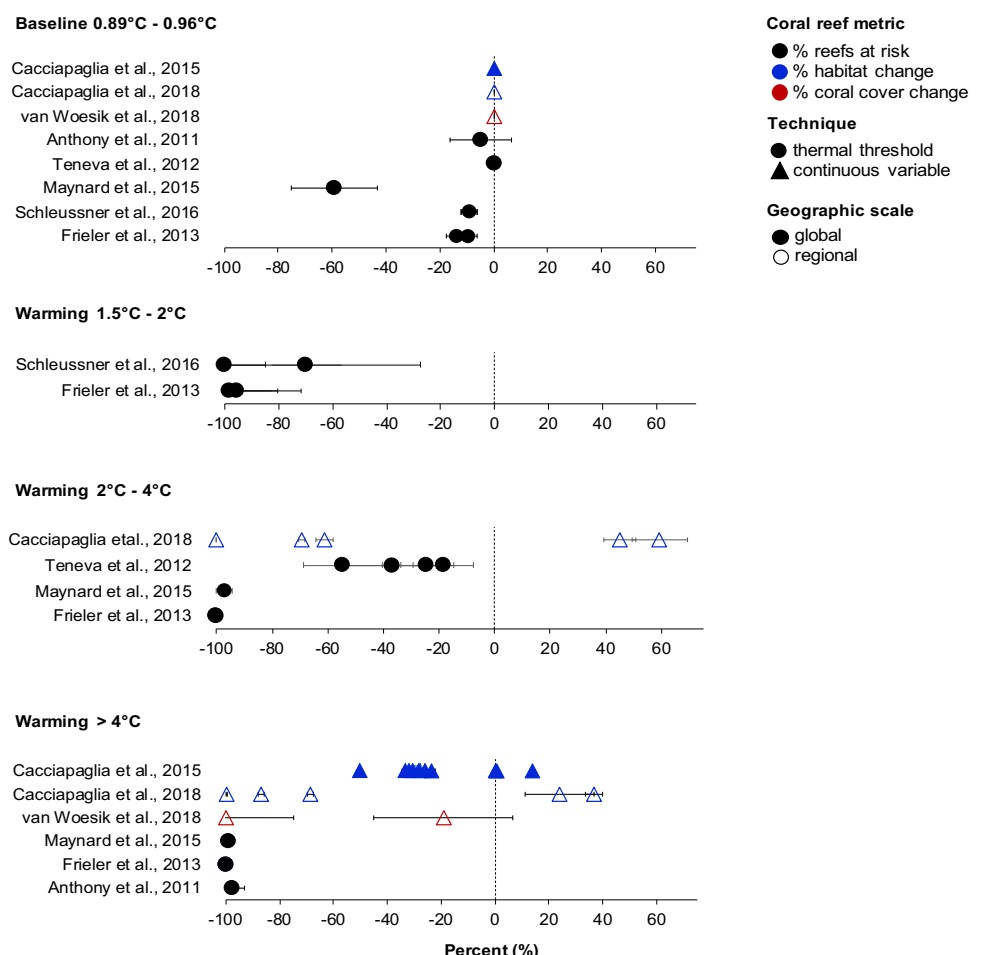

**Fig. 4 | Percent changes in coral reef metrics representing the model outputs used in the analysis of Fig. 3.** Percent change in mean estimates (±1 standard deviation) of model outputs (i.e., coral reef metrics) used in the analysis presented in Fig. 3. Model outputs were extracted from *n* = 39 modeling scenarios across eight published studies and converted into percent change for ease of interpretation. Mean estimates of coral reef metrics for historical global warming levels of 0.86–0.96 °C represent the baseline period of the years 2000–2015. Mean estimates of coral reef metrics categorized into for future warming scenarios of 1.5–2 °C, 2–4 °C, and >4 °C represent projections at the end of the century (years

2090–2100). Negative values for percent reef cells at risk (black), percent habitat change (blue), and percent coral cover change (red) signify adverse ecological impacts compared to a baseline of 0% (no effect), while positive values indicate a positive effect direction, such as projections estimating increases in reef cell habitat availability. Circles and triangles denote studies using thermal threshold and continuous variable techniques for modeling heat stress, respectively. Open symbols represent global-scale projections, while closed symbols denote regional-scale projections. Supplementary Data 3 provides a comprehensive list of individual scenario descriptions. Source data are provided as a Source Data file.

factors, we aligned model outputs to baselines years between 2000 and 2015 (0.86–0.96 °C) and three end-of-century warming scenarios (1.5–2 °C, 2–4 °C, and >4 °C), which categorized the various emissions scenarios used (Supplementary Data 3). We further categorized each study based on whether it employed a thermal threshold or continuous variable technique in modeling heat stress and whether it presented global or regional-scale projections (Fig. 3).

Nearly all studies projected negative impacts on the coral reef metrics, but the relative sizes of these effects differed (Fig. 3). Articles that used thermal threshold techniques tended to yield more negative effect sizes than alternative methods (Fig. 3). Among the threshold studies in the 2–4 °C scenarios, Teneva[117] produced a relatively small effect size, aligning with the study's less severe and more variable projections of reef cells at risk (Figs. 3 & 4). The projections by Teneva et al.[117] cannot be easily compared with other threshold studies reviewed here because of various methodological differences. In

contrast to the other studies[14], which applied global temperature thresholds to estimate future bleaching frequencies, Teneva et al.[117] used bleaching observations from Reef Base to test prediction methods in which thermal thresholds were determined by historical SST variability. Accounting for historical climate experience might explain why the projections by Teneva et al.[117] deviated from most other threshold studies in the analysis (Fig. 4). Importantly, Teneva et al.[117] also defined reef cells at risk as grid cells characterized by at least a 50% probability of experiencing 5-year mild or severe bleaching events by 2100.

In future scenarios characterized by >4 °C warming, articles applying thermal threshold techniques consistently projected that >93% of global reef cells will be at risk by the end of the century[14,42,55] (Fig. 4). However, the study by Maynard et al.[42] generated a notably smaller effect size than Frieler et al.[14] and Anthony et al.[55] (Fig. 3). Given that the effect sizes were based on relative differences between the

baseline and end-of-century scenarios and weighted for variance, this discrepancy may be explained by the more severe and variable baseline impacts modeled by Maynard et al.[42] (Fig. 4). The relatively large effect size for Frieler et al.'s[14] projections occurred because of the absence of any variance with the study's drastic projections of reef cells at risk (−100%, ±0 Std) (Fig. 4). In contrast to the threshold studies, articles employing continuous variable techniques produced effect sizes that were relatively modest, but variable (Fig. 3). This can be attributed to the high variability in model outputs across the individual scenarios, reflecting the distinct characteristics of different coral reef provinces (Fig. 4). For example, projected changes in suitable habitats under >4 °C ranged from −99.9% (±3.1 Std) to +36.8% (±3.1 Std), and changes in coral cover varied from −100% (± 25.4 Std) to −19.2 (±25.8 Std) (Fig. 4).

How do our findings relate to the IPCC's projections for coral reefs? The IPCC's AR6 Summary for Policy Makers anticipates that coral reefs will decline by 70–90% at 1.5 °C global warming, exceeding 99% at 2 °C (with high confidence)[22,150]. Although the IPCC reports lack a definition for coral reef "decline," their assessments draw on projections from Schleussner et al.[15] and Frieler et al.[14]. The two studies exhibit high agreement, collectively estimating that between 69.7% (±42.2 1 Std) and 100% (±15.2 1 Std) of coral reef cells will be at risk under scenarios of 1.5–2 °C (Fig. 4). We find that these projections generated effect sizes similar to those generated by alternative methodologies under even the most pessimistic warming scenario (Fig. 3). This suggests that the studies serving as a basis for recent climate change impact assessments[1,22,23] might project more severe consequences for coral reefs than other approaches.

The main reason for the high coherence between Schleussner et al.[15] and Frieler et al.[14] is the minimal differences in their approaches to modeling the frequency of bleaching events across global reef cells. Both articles used the same model type and made analogous assumptions, including their selection of global thermal thresholds and frequency of heating events expected to impede reef recovery. Overall, there are several factors that may explain the high variation in expected outcomes for coral reefs. In contrast to Schleussner et al.[15] and Frieler et al.[14], differences in model types, model parameterization, and assumptions are likely important factors explaining differences in the extent of expected impacts. While our analysis has limitations, it underscores the importance of exercising caution when drawing conclusions from a limited number of key studies and emphasizes the need for enhanced coordination to transition toward a multi-model ensemble approach.

## Reporting uncertainty and metrics of model outputs

One of the fundamental, yet basic steps toward improving future syntheses of modeled projections is the adherence to essential reporting standards. While all studies in our database provided ample data to facilitate interpretation of the study outcomes, most (89% of studies) failed to report basic metrics for model outputs or sufficient extractable data for measures of variation to be converted into the same units. In many cases, challenges arose from the display of results in figures and geographical maps that were not accompanied by adequate supplemental information reporting extractable values. While there is an increasing emphasis on depositing empirical data into online repositories (e.g., Dryad, Figshare, and Zenodo), this is rarely required for model outputs. Recognizing the necessity for reporting and metadata availability standards, other fields focused on projecting climate change impacts to biological systems have implemented agreed-upon standards[66,67,134]. For instance, the IUCN established preliminary reporting standards for species threat assessments based on SDMs[151], which have been further refined in subsequent publications[66,67].

Another vital component of studies projecting coral reef futures is clarity over units of the metrics projected. Modeling studies simulate changes using a diverse set of metrics that vary according to the purpose of the study and ultimately communicate the extent and nature of expected impacts on coral reefs. However, a lack of clarity over the ecological or biological meaning of the projected variable and the exact outcomes anticipated for coral reefs constrains the usefulness of projections in guiding effective decision making, management, and conversation efforts.

While the vast majority explicitly define the metrics simulated, some earlier studies provide indistinct descriptions (Supplementary Data 2). For instance, several 'excess heat' threshold models simulate the frequency of severe bleaching events to deliver projections as the proportion of coral reef cells (e.g., 1° × 1° grid cells on the Earth's surface) at risk of 'long-term degradation' or 'severe bleaching events'[14–17]. However, there is presently no agreed nomenclature of such states for coral reefs, raising uncertainty as to their exact meaning and the consequences involved. In some cases, subjective terms affect the communication of projections in influential assessments of climate change impacts, where terms like 'losses of coral reefs'[152], 'corals being lost'[23], and 'coral reefs at risk'[23] are used interchangeably without accompanying definitions. These terms could, in theory, be understood to imply a range of outcomes for coral reefs, ranging from reductions in live coral to the ecological collapse of entire reef ecosystems. Clear and well-defined nomenclature is especially important to in the context executive summaries addressing policymakers and other stakeholders. In summary, establishing and adhering to standards for the comprehensive reporting and communication of projections, including associated uncertainties, would facilitate more conclusive syntheses of coral reef projections in the future. This may also involve setting standards for publishing metadata.

## Toward ecologically relevant and restoration-compatible spatial scales

The recent establishment of ambitious goals to restore biodiversity (Kunming−Montreal biodiversity framework) has ignited a race to identify effective strategies assisting decision-makers in implementing successful mitigation and intervention efforts for coral reefs[153]. The capacity of projection models to guide these strategies, however, is challenged by the difficulties they face in detecting changes at practical scales[17,154]. Almost half of the studies in our database (49%) provided projections at geographical resolutions lower than 0.25° latitude × 0.25° longitude (Supplementary Data 2). In practical terms, this roughly corresponds to grid cells with an area of 770 km² at the equator − a size that is orders of magnitude larger than a typical coral reef.

There are two main approaches to improve the spatial resolution of global and regional models: statistical and dynamical downscaling procedures[155] (Table 1). Statistical downscaling estimates local-scale climate variables from larger-scale climate models using statistical methods, whereas dynamical downscaling uses regional numerical models to simulate local conditions at a higher spatial resolution based on global climate model outputs[156]. Among the 19 studies in our database that applied downscaling techniques, the majority (85%) used statically downscaled models to formulate their projections (Supplementary Data 2). While statistical techniques are computationally inexpensive, one major drawback is their inherent assumption that patterns between large- and local-scale climates observed today will remain unchanged in the future[157] (Table 1). This assumption introduces substantial uncertainty across decadal time frames[157]. On the other hand, dynamical techniques explicitly model ocean dynamics and are more likely to capture the key processes involved[156] (Table 1). These dynamical procedures, however, can still inherit biases present in the large-scale climate models and face challenges in considering how ocean dynamics may change over time[157,158] (Table 1).

We found only one study that compared the performance of statistical and dynamical downscaling procedures. The study by

**Table 1 | Advantages and limitations of statistical and dynamical downscaling procedures. Adapted from ref. 196**

| | Advantages | Limitations |
|---|---|---|
| Statistical downscaling | • Computationally inexpensive and requires minimal expertise | • Assumes constant relationship between local and large-scale climate through time |
| | • May correct for biases in GCMs | • May not capture climate mechanisms |
| | • Can be applied in data-scarce regions | • Limited ability to capture variability and extremes |
| | • More flexibility in models and scenarios | |
| Dynamical downscaling | • Simulates climate mechanisms and more likely to capture key processes involved | • Computationally demanding, requires specialized expertise, and longer run-time |
| | • No assumptions of the relationship between current and future climate conditions | • Biases present in GCMs can extend and propagate to regional scales |
| | • Technology advances constantly improving availability of regional climate models | • Results can be sensitive to uncertain parameterizations |
| | | • Limited flexibility, often tied to specific models and scenarios |

Hooidonk and colleagues compared models of annual coral reef bleaching in the Caribbean that were downscaled using either statistical or dynamical procedures[156]. While there was a high level of agreement between the projections produced by the two techniques, the dynamically downscaled model detected an earlier onset of annual severe bleaching linked to future changes in regional currents. In contrast, the statistical procedure failed to detect these changes due to its inability to capture local-scale features, such as eddies, which influence warming levels leading to coral bleaching[156]. Although these results suggest that dynamical downscaling may outperform statistical methods, further assessments of the relative costs and benefits of the two techniques are warranted (Table 1). Downscaling techniques, however, ultimately introduce an additional source of uncertainty. Fortunately, the spatial resolution of global models is expected to improve in the near term with the introduction of new data streams, including higher-resolution satellites (e.g., Himawari[159]) coming online. This enhancement will improve sea surface temperature (SST) data resolution and reduce the reliance on downscaling approaches[160].

**Geographical bias in modeled projections**
It is well-documented that coral reef responses to climate change vary across major coral reef provinces[13,161,162]. However, when analyzing the landscape of climate projections, it becomes evident that there are substantial geographic gaps that require attention (Fig. 5)[163,164]. A significant portion of the research provides global-scale projections, which offer a broad perspective on climate patterns and anticipated changes across coral reefs worldwide. While these global-scale projections provide valuable insights into overall trends, they lack the necessary resolution and accuracy to provide detailed and reliable information at more practical scales for management and intervention purposes[17,154]. In contrast, regional-scale models usually benefit from region-specific data and typically offer projections with finer spatial detail, addressing the need for more localized information to inform conservation efforts[156].

Our analysis shows that the availability of regional-scale models is inconsistent across the world's coral reefs. Provinces such as eastern Australia and the Caribbean have received considerable attention and have well-documented projections using various modeling approaches (Fig. 5). However, other equally important coral reef provinces, including the eastern Pacific (Costa Rica, Ecuador, and Mexico), the western Atlantic (Brazil's northeastern coast), the Indian Ocean, and the Arabian Seas, lack regional-scale models (Fig. 5)[165]. These understudied regions thus heavily rely on less-tailored global assessments for projections of future reef impacts in these locations. Many of these provinces also suffer from a limited number of studies and diversity of modeling approaches (Fig. 5). For example, projections for the Arabian seas, the western Atlantic, and the eastern Pacific are exclusively based on SDMs, which involve key assumptions and

limitations. Coral reef scientists are increasingly aware of this issue. Addressing these gaps necessitates targeted efforts to enhance the resolution and accuracy of global-scale projections, while simultaneously expanding the scope and diversity of regional and local-scale projections and monitoring efforts. Such efforts are already underway and essential in providing decision-makers with actional information to manage climate change impacts on coral reefs at global, regional, and local scales[166,167].

**Beyond the impact of warming**
Although climate change is acknowledged as a dominant driver of coral reef degradation, it is clearly not the only threat. The extensive list of pressures includes ocean acidification[168], sea-level rise[169], deoxygenation[170], cyclones[171], pollution[172] as well as numerous biotic pressures such as disease[42], pest species[173], and overfishing[174]. However, the vast majority of studies in this review modeled the impacts of warming alone or warming in combination with only one other stressor (76% of studies) (Supplementary Data 1). In reality, coral reefs are subject to ongoing climate change and a complex interplay of numerous interacting pressures that operate across various temporal and spatial scales.

Coral reef research has allocated significant effort to projecting and understanding the combined impacts of climate change and ocean acidification on coral reefs (Supplementary Data 1 & Supplementary Table 2). On the other hand, our analysis revealed that 16 studies in the database considered pollution to some extent, and four studies considered fishing pressure in their projections (Supplementary Data 1 & Supplementary Table 2) Although ocean acidification will undoubtedly have discernable effects on coral reefs[175], there are no practical solutions available to mitigate ocean acidification, apart from the urgent reduction of greenhouse gas emissions[176,177]. In contrast, elevated nutrients and fishing pressure are now well recognized to increase the susceptibility of coral reefs to heatwaves[172,178,179], and measures to address these pressures are effective and practical[180,181]. Local-scale management actions to minimize pollution and regulate fishing have already demonstrated success in reducing cumulative impacts to coral reefs[180–183], particularly in Pacific nations where actions to manage reefs have been implemented for centuries[184].

A similar pattern exists for evaluating how pest species and disease will interact with climate change to shape the future of coral reefs. In our analysis, we found only two investigations that delved into the role of coral disease outbreaks in influencing coral reef futures under climate change (Supplementary Data 1 & Supplementary Table 2)[34,42], with one of these studies being limited to a simulation of a single reef. The global study highlighted that future warming is likely to heighten coral susceptibility to disease and identified specific locations where targeted management could be implemented[42]. Although excluded from our analysis due to the absence of future climate change

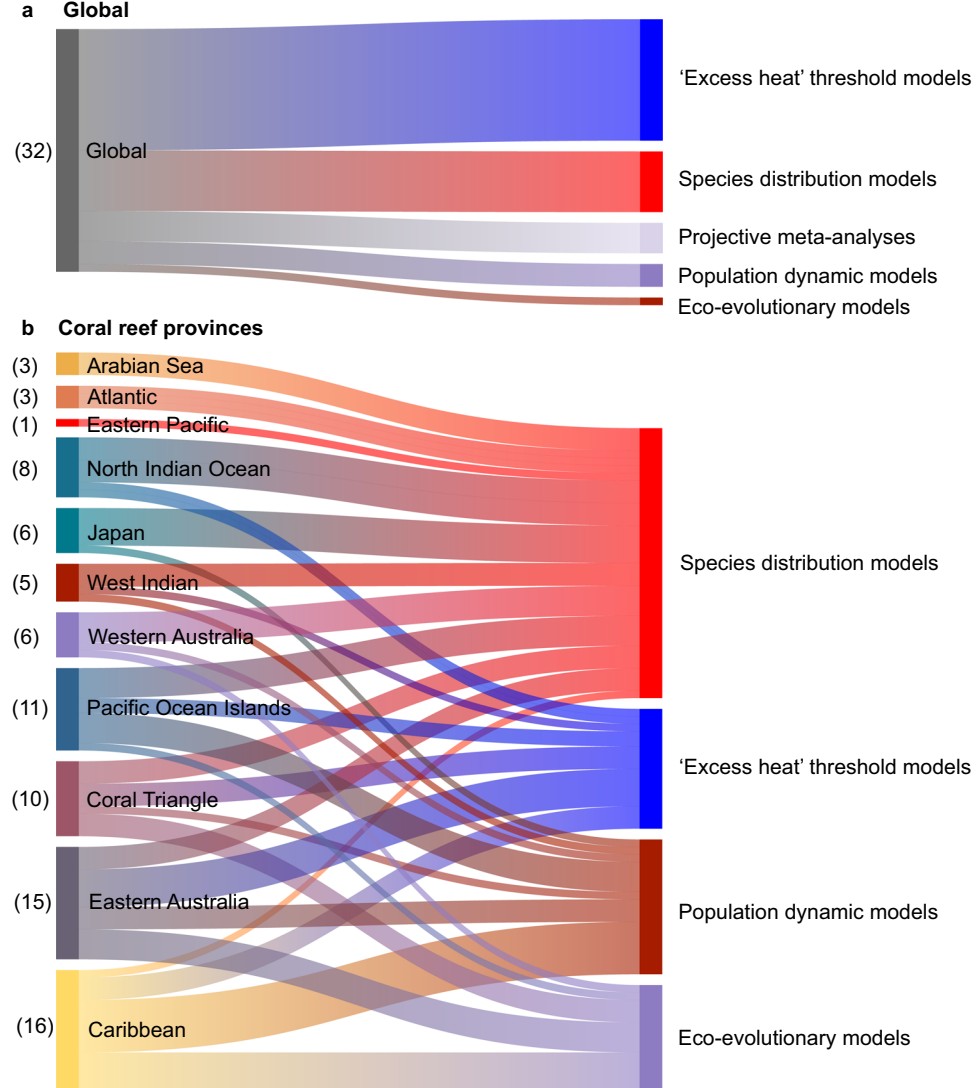

**Fig. 5 | Association between major coral reef provinces and applied approaches used to project coral reef futures. a** represents the distribution of modeling approaches used at a global-scale, and **b** represents the association between coral reef provinces and the main methodologies used. The specific flow width is proportional to the number of research articles applying each of the five main methods, while the numbers in parentheses indicate the total count of articles that generated projections for global reefs (**a**) or each reef province (**b**). See Supplementary Data 1 for a full description of the focal geographic regions for each study included in database (*n* = 74). This diagram has been generated using the online tool: Visual Paradigm (https://online.visual-paradigm.com). Source data are provided as a Source Data file.

projections, numerous predictive models serving as early warning systems for coral diseases exist[185–187]. These early warning system models have identified crucial drivers of disease outbreaks in various regions, which could prove useful for refining existing models projecting coral disease outbreaks under future climate change scenarios. Similarly, we identified only one study that simulated the impact of a pest species in climate change scenarios for coral reefs (Supplementary Data 1 & Supplementary Table 2). This study assessed the potential effectiveness of management strategies in addressing outbreaks of CoTS and reducing cumulative impacts on the Great Barrier Reef[54]. The urgency to address this area of uncertainty is underscored by the ongoing coral disease outbreak in the Gulf of Mexico, which poses a severe threat to coral reefs in the region[188,189]. Disease outbreaks are becoming increasingly concerning, affecting not only coral reefs but also other marine life[190,191], highlighting the need for urgent attention and action.

With the growing recognition of the need for intervention measures, particularly in line with the Kunming–Montreal biodiversity framework's objective of restoring 30% of degraded habitats by 2030, projection models are likely to play a crucial role in guiding these endeavors. Our analysis points toward a possible need to shift the focus of future modeling experiments to better guide actions to manage and restore coral reefs. This does not imply that modeling studies should neglect stressors like ocean acidification, which are expected to have long-term impacts with limited practical solutions. Instead, modelers could consider prioritizing the inclusion of management and intervention scenarios, including coral reef restoration, that integrate the modeled effects of global and regional pressures. Just three of the 79 studies reviewed here included potential intervention scenarios. Two of these studies explored unconventional geoengineering solutions[96,192], while one simulated the potential benefits of demographic restoration and assisted evolution in enhancing reef resilience[83].

In summary, projections of coral reef futures at global, regional, and local scales play a crucial role in informing discussions and policy-making at various levels of governance. While recognizing the diverse

objectives and methods employed in the reviewed articles, there is a clear need for greater coordination in efforts to project coral reef futures. Robust projections are vital for decision-makers and policy-makers to implement effective strategies for coral reef management and restoration, helping us achieve our climate, biodiversity, and sustainable development goals. The recommendations presented here propose tangible steps toward a greater understanding of the uncertainty surrounding coral reef futures while also promoting transparency in reporting projections and communicating them to decision-makers. Crucially, the success of these endeavors will depend on interactive communication between the scientific community, policymakers, and local end-users.

## Methods

### Literature search and study selection

We searched the Thomson Reuters Web of Science database (http://www.webofknowledge.com) to identify studies projecting the impact of climate change on shallow tropical and sub-tropical coral reefs. The search was performed on March 6, 2023, and retrieved 2705 peer-reviewed articles. Our literature search strategy followed the guidelines of PRISMA (Preferred Reporting Items for Systematic Reviews and Meta-analyses)[193] (Supplementary Fig. 1). To synthesize the initial database, we screened the title, abstract, and display items of each article, resulting in the identification of 2073 potentially eligible articles to be included in our database (Supplementary Fig. 1). Publications were then selected based on the following criteria: (1) projections represented the responses of tropical and/or sub-tropical coral reefs to future levels of warming alone or in combination with any other drivers, (2) future emissions pathways and/or warming scenarios used to force the simulations were stated, and (3) projections were modeled across more than one reef site to be included in the database. The final database consisted of 79 peer-reviewed articles published between 1999 and 2023.

### Data extraction

We initially extracted the key characteristics of each study, including the focal variable(s) simulated, model inputs, spatial scale, and focal geographic area. We classified the models into five broad categories of methodologies: (a) 'excess heat'/threshold models, (b) population dynamic models, (c) species distribution models, (d) ecological-evolutionary models, and (e) meta-analyses of published data (see the Main text for definitions). In a few cases where studies could not be categorized, the model type was recorded as 'other' (Supplementary Data 1). We further classified the studies according to the underlying techniques used to simulate heat stress on reefs, as either threshold techniques or continuous variable techniques (see the Main text for definitions). We recorded each study's purpose, underlying methodological approach, key assumptions, spatial resolution, and application of downscaling techniques (Supplementary Data 2). Finally, we acknowledged the diverse range of approaches used to simulate coral reef futures by summarizing the key advantages and limitations of each study (Supplementary Data 2).

### Study criteria and data analysis

A major objective of our study was to examine and compare the magnitude of projected impacts and estimated uncertainties across different model types. Meta-analyses offer a valuable approach to aggregate evidence from multiple studies to provide a comprehensive overview of current modeled projections[149]. The database of 79 studies was considered for inclusion in the exploratory meta-analysis based on specific criteria (view supplementary methods for detailed list and Supplementary Fig. 1). Briefly, to enable a meaningful analysis, we identified the three most common coral reef metrics used as model outputs in our database. The first unit, usually expressed as a percentage of reef cells at risk of repeated severe bleaching events

(or 'long-term degradation'[14,15]), was a common model output of 'excess heat' threshold models (Supplementary Data 1). Both population dynamic and ecological-evolutionary model types frequently projected changes in percent coral cover, whereas species distribution/niche models usually simulated fractional changes in habitat suitability (Supplementary Data 1). Among those, only studies that provided: (1) sufficient data for projection estimates and uncertainty measures to be reliably extracted or calculated, (2) reported end-of-century projections, and (3) used a baseline period between 2000 and 2015, were selected for the exploratory meta-analysis. In cases where projection and uncertainty estimates were only presented in figures, values were extracted using PlotDigitizer (plotdigitizer.com), where possible. When projection estimates and uncertainties were reported as proportional values between 0 and 1, we converted these values to percentages ranging from 0 to 100.

Among the initial pool of 79 studies, eight studies were identified as containing quantitative data that could be extracted and compared in our analysis. As such, due to the low number of studies included, we consider this analysis to be exploratory in nature. For each study, we calculated Hedges' $g$ effect sizes and variance for all individual scenarios/trajectories (39 scenarios in total) (Supplementary Data 3). The signs of the effect sizes (positive or negative) were adjusted to align with the effect direction reported by the individual studies. In this adjustment, a negative effect size denotes a negative ecological response, while a positive effect size indicates a positive ecological response (Supplementary Methods). Hedges' $g$ quantifies the difference between the means of two groups divided by the pooled standard deviations and was calculated as follows:

$$g = (X_P - X_B) \times \frac{J}{s.d._{pooled}} \tag{1}$$

where $X_P$ and $X_B$ are the estimate of end-of-century projections and baseline data, respectively. $J$ corrects for bias attributed to different sample sizes by differentially weighting studies as follows:

$$J = 1 - \left( \frac{3}{(4 \times (N_P + N_B - 2) - 1)} \right) \tag{2}$$

Where $N_P$ and $N_B$ are the number of models used for projections and baselines.

The $s.d._{pooled}$ was calculated as follows:

$$s.d._{pooled} = \sqrt{\left( \frac{(N_P - 1) \times (s.d._{P})^2 + (N_B - 1) \times (s.d._{B})^2}{(N_P + N_B - 2)} \right)} \tag{3}$$

Variance for each scenario was calculated as:

$$V_g = \left( \left( \frac{N_P + N_B}{N_P \times N_B} \right) + \left( \frac{g^2}{2 \times (N_P + N_B)} \right) \right) \tag{4}$$

All calculations were computed using the metafor package (v. 4.2-0) in R (v. 4.3.0)[194].

### Reporting summary

Further information on research design is available in the Nature Portfolio Reporting Summary linked to this article.

## Data availability

The source data supporting Figs. 1–5 are available in the Source Data file. Supplementary Data Files 1–3 provide a summary of all other data generated by this study, and the complete database is deposited in Dryad (https://doi.org/10.5061/dryad.4f4qrfjkp)[195]. Source data are provided with this paper.

## Code availability

The R script needed to produce the analysis has been deposited in Dryad (https://doi.org/10.5061/dryad.4f4qrfjkp)[195].

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

## Acknowledgements
King Abdullah University of Science and Technology (KAUST) supported this research through baseline funding to C.M.D.

## Author contributions
S.G.K. and C.M.D. conceptualized the study and conceived the study design. S.G.K and C.R. extracted the data, conducted the analyses, and prepared the figures and tables. S.G.K. and C.M.D. wrote the manuscript, with substantive contributions from C.R. All authors contributed to interpretation, revisions, and editing.

## Competing interests
The authors declare no competing interests.
