## [Peer Review File · Nature Communications]

Systematic review of the uncertainty of coral reef futures under climate changeREVIEWERS' COMMENTS:

Reviewer #1 (Remarks to the Author):

This is a pretty good summary of limitations regarding the modeling of climate change on reefs. Overall I agree with the majority of the authors' conclusions and the identification of major limitations, I was not convinced that a clear pathway forwards has been suggested. In the majority of cases there is an ask for more studies and data, which is fine but does not necessarily help someone prioritize where to focus.

I was not compelled by the conclusion that we should be using hierarchical models. This attracts very little explanation in the text and seems to suffer a number of limitations - just as any other approach does. These are not discussed. In general a hierarchical Bayesian model is nothing special - just a statistical analysis at multiple levels. It still has the same limitations of any statistical approach in that it is often not mechanistic and therefore it is difficult to project values outside the bounds used to create the model. And while good statistical models include a mechanistic equation, not all such equations are truly mechanistic. For example, the logistic, which forms the foundation of population modelling, is just a mathematical convenience for an s-shaped function. It doesn't intrinsically include mechanisms.

To me an important omission of the paper is the consideration of study objectives - is it about generic principles or more site-specific projection of possible futures? Tackling generic principles with simple models is appealing and it is relatively simple to capture the sources of variability in the models. BUT they do not adequately represent the biological uncertainty in their behaviour. I have seen many examples of practitioners and other scientists reading far too much realism into the results of such models, even to the point of using them for prediction. How should we handle the relegation of an ecosystem to a simple representation? What sorts of questions should be tackled with these models? Moreover, if we reduce the ecosystem to a cartoonish simplification, are the models able to tell us anything novel? To be clear, I use such models myself so I'm not against them but feel like the limitations need to be clear.

My other concern is that the handling of uncertainty is still problematic. I like the way the paper articulates major sources of this. But I am yet to see a clear pathway for how to communicate such massive uncertainty in a meaningful way that doesn't undermine the purpose of the entire modelling exercise. Simply being able to capture and propagate uncertainty in a stochastic manner across the model system does not necessarily help someone make a decision. It really needs to return to the objective of the study and how can uncertainty analysis inform the analysis / decision / conclusion. Indeed, I often hear colleagues asking for greater analysis of uncertainty but yet there are few cases where such analyses have added significant value. To me the paper has an opportunity to make a contribution here. I like Fig 2 and the treatment of uncertainty so far. Can you go further and provide a clear pathway for its use?

A few comments on the paper:

The section on eco-evo models does not criticize any of the limitations of such models which makes things a little uneven. They have massive uncertainty in terms of the taxa responsible for adaptation (coral, symbionts, etc), the complexity of traits and genes that might encode thermal tolerance, and the biological simplification of reality in the ecology.

I felt that fig 1 needed further discussion to make its value clearer to the paper.

Agree with the discussion of limitations of measuring thermal stress. You missed a recent paper by Lachs 2021 that did a global analysis of bleaching fitting a better DHW algorithm. Not a big deal but it complements the McClanahan stuff.

Reviewer #2 (Remarks to the Author):

The paper reviews the literature on coral responses to thermal stress and argues there are many sources of uncertainty that make future predictions difficult. This topic is important as some people outside of the field have claimed high confidence in predictions of coral reef response models to climate change (i.e., McKay et al. 2022). Those in the field will be aware of the “gaps” in confidence that the authors elucidate. In many ways this “review”

summarizes many of the discussion sections of these papers and brings them together in a single perspective. This is a potentially useful product. However, I have many problems with the paper that would require a major revision if they can be addressed. Because critical readers cannot check or confirm to the authors conclusions as currently written, the paper should be rejected. That is there is a poor connection between the authors conclusions and the cited literature and no quantitative analysis of the literature.

A major problem is the lack of a systematic review that would force the authors to consider the full literature in this field. The ideas and citations are typical of this field where some of the same authors and papers get cited repeatedly, even if they are not the best papers on these subjects. Often, they reflect poor science but get cited because they are provocative. The climate change literature is full of provocative papers that will not stand up to scrutiny. Then, many papers directly relevant to the topics are ignored and not cited even though they make very strong contributions to very specific issues. There are many recent papers that are not cited and cases where papers are cited in the reference section but not in the text where the contributions were made. The text suffers this problem throughout and leads to a disappointing coverage of issues. The authors probably know this, but they repeat this behavior as it may be accepted to be provocative rather than analytically rigorous.

Most of the text comes to sweeping conclusions without citing the studies from where some of these conclusions arose. I suggest the authors see several recent reviews, meta-analysis, and papers of which I list some below. They should also give specific examples of studies and consider putting these supporting and unresponsive studies in a table where we can consider the evidence ourselves. They should not come to categorizations and conclusions themselves based on an incomplete review of the literature. They need to quantify and disarticulate the many papers and responses of corals in these models. For example, being clear about the differences in thresholds, bleaching responses, coral cover, genomic and community change. Responses tend to get lumped together and yet they are not the same. Words like uncertainty are not well defined and so we are not certain what is being evaluated and by what means. This makes the paper hard to review for its scientific veracity and should lead to rejection.

The figures are not that useful or hard to interpret. The authors need to make it clear what they did, for example, to produce figure 1. This seems to be a result without a method that can be checked for veracity. Figure 3 is not helpful and should be based on quantified information rather than a cartoon or schematic drawing. The supplementary table was disappointing as it seemed so like the table in the text. I was hoping to find evidence for these categorizations with specific citations of literature. This table seems to lack quantification and is the opinion of the authors, based on an incomplete reading of the literature.

The English composition is weak, the paper is not a pleasure to read, and the paper should be rewritten once the literature is reviewed systematically and updated.

There are several additional issues that I outline below in more detail.

Abstract:

I don't find the abstract very informative as it lacks details. It reflects or sounds similar to many of the discussion sections of papers on climate and corals, where there is a call for better resolution and knowledge of adaptation. It seems like a time to give more details and specific suggestions.

“that existing projections reflecting a near-complete loss”

The authors need to be specific about which projections. There are different methods used to make predictions, but most are excess heat predictions with some that include ocean acidity.

Introduction

Many paragraphs are not really paragraphs. English composition need attention throughout the manuscript, many short and long paragraphs are intermixed without much structure.

Why not set up the introduction relative to this paper, rather than the usual Hughes papers?

Armstrong McKay,. 2022. Exceeding 1.5 C global warming could trigger multiple climate tipping points. *Science* 377:eabn7950.

L45 – Some of the citation seems selective or old in this rapidly moving field. For example, the increased frequency of thermal thresholds is repeated in many papers but the evidence is not strong. See, for example, this paper where events are show to be not more frequent but larger scale and longer duration.

Skirving, 2019. The relentless march of mass coral bleaching: A global perspective of changing heat stress. *Coral Reefs* 38:547 - 557.

L48 – local stressors often include background conditions. Why say otherwise when there are many papers that have specifically examined this topic. None cited. This is just one of many examples.

2021 Environmental variability and threshold model's predictions for coral reefs. *Frontiers in Marine Science* 8:1774.

L49: The 50% decline is not support here.

Tebbett, S. B., 2023. Benthic composition changes on coral reefs at global scales. *Nature Ecology & Evolution* 7:71-81.

L52 – how was this review done? The citation appears quite selective, so it is not clear that the authors have actually reviewed the full literature. See this more systematic review.

Coral responses to climate change exposure. *Environmental Research Letters* 17:073001.

Table 1 might have some specific papers cited as examples. This table seems to be the creation of the authors without any analysis of publication statistics, such as word use of variables, etc. It is not really a meta-analysis but more about the authors interpretation and opinions about the literature. I see it as subjective and likely incomplete.

L62 – It this a title? Seems to be an incomplete sentence, so maybe put them in italics or bold to be clear they are title to sections.

Any model that predicts the frequency of bleaching events must contend with the fact that frequency has not changed in most regions, as described by Skirving. These models require rates of SST rise and threshold information, so best to describe them by how they were built. The paper lacks much consideration of how these models were built, what were their key assumptions, etc.

L64 – What models specifically are being discussed. There are a variety of model and many these days are using 4 DHW and not modelling at the month. Thus, the use of month and 2 degrees sounds like a selection of some of the older models. A 1-degree threshold is very common for bleaching studies, so make it clear if this is a cumulative or simple threshold. The 1 degree was first suggested here. However, many recent studies do not support this suggestion. See for example, papers by DeCarlo.

Hoegh-Guldberg, O., and B. Salvat. 1995. Periodic mass-bleaching and elevated sea temperatures: bleaching of outer reef slope communities in Moorea, French Polynesia. *Marine Ecology Progress Series* 121:181-190.

L 67 - Reefs at Risk is this the title and findings in the report, or is this a general statement based on the unknown papers being describe here? There are many studies reporting different threshold, so be clearer about what you are reviewing.

L68 – this is important but not clear as written. Thresholds are to predict bleaching not coral survival, so do not assume too much about their predictions unless addressing specific studies. Is the 99% lost are is it that they will bleach every year? This vagueness in responses is a problem that needs to be rectified.

L75 – This text seems dated in the context of some recent papers, such as.

McManus, L. C., 2021. Evolution and connectivity influence the persistence and recovery of

coral reefs under climate change in the Caribbean, Southwest Pacific, and Coral Triangle. *Global Change Biology* 27:4307-4321.

Would niche models fit this category? Do they require another category?

van Woesik, R., S. Köksal, A. Ünal, C. W. Cacciapaglia, and C. J. Randall. 2018. Predicting coral dynamics through climate change. *Scientific Reports* 8:17997.

Cacciapaglia, C., and R. van Woesik. 2018. Marine species distribution modelling and the effects of genetic isolation under climate change. *Journal of Biogeography* 45:154-163.

L103 -are they predicting losses of coral cover or bleaching event frequencies? This paper connects bleaching to coral losses but I don't believe the authors are referring to this paper.

Cornwall, C. E., et al. 2021. Global declines in coral reef calcium carbonate production under ocean acidification and warming. *Proceedings of the National Academy of Sciences* 118:e2015265118.

Figure 1 what does coral performance mean and how is it related to coral losses discussed in the text? Why would performance be a percentage? How many studies (n=?) were used to make these uncertainty estimates? What does uncertainty mean? This figure will not tolerate much scrutiny as presented.

Gap text – English composition is weak here in that there are many long paragraphs lacking structure.

L137 – changes in what, the characteristics of MHWs or in impacts on corals? It would be good to describe the underlying structure of the models, so one can understand the predictions better. What variables are being modelled, what are the key assumptions, etc. MHW seem to be defined by the duration of heat but what about the extent of the heat. The duration of warm events will occur in the future but coral sensitivity is also not likely to be responding to the same thresholds. The threshold for bleaching is increasing over time.

Do these models consider this change in thresholds?

Sully, S., D. E. Burkepile, M. K. Donovan, G. Hodgson, and R. van Woesik. 2019. A global analysis of coral bleaching over the past two decades. *Nature Communications* 10:1-5.

L169. One wonders how the Skirving paper compares to the Li and Donner paper, especially in terms of the frequency of temperatures that pass certain thresholds. Are the authors being selective here or interpreting frequency correctly?

Gap 2 – is this variability in the models or the sensitivity of the corals or both. Not clear as written. The authors need to distinguish spatial from taxonomic variability. The word coral embodies a great deal of variability in taxa, evolution, and ecology, and yet this is not made clear.

L185 – The same data set indicated that coral taxa are having different adaptation rates and therefore reversal of responses over time. The authors could separate these responses in space, time, and taxa better in this text.

McClanahan. 2020. Highly variable taxa-specific coral bleaching responses to thermal stress. *Marine Ecology Progress Series* 648:135 - 151.

L193 – See the Tebbett et al. paper cited above for coral cover patterns over time and how the Indo West and Indian Ocean are showing little effects over time. Also, see the McClanahan *Frontiers in Marine Science* 2021 paper above where two models are compared and where large differences are reported.

The responses as differences in thresholds, bleaching, and coral cover are very seldom disarticulated in the paper, and yet they are not the same thing. Thresholds have been shown to differ geographically, but this is not covered or evidence given despite a number of papers on the subject.

DeCarlo, T. M. 2020. Treating coral bleaching as weather: A framework to validate and

optimize prediction skill. PeerJ 8:e9449.

Coral is composed of many life histories with various responses. This blending of concepts creates vagueness that makes the paper hard to follow.

Darling, E. S., 2019. Social–environmental drivers inform strategic management of coral reefs in the Anthropocene. *Nature Ecology & Evolution* 3:1341-1350.

L254 – One wonders how these mechanistic models compare with empirical data. There are some comprehensive models, so not sure I would call one the most comprehensive models. A number are not cited in this paper.

MacNeil, M. A., 2019. Water quality mediates resilience on the Great Barrier Reef. *Nature Ecology & Evolution* 3:620 - 627.

L269 – I would suggest the *Frontiers in Marine Science* as doing most of what is claimed not to have been done here. Empirically, consider the findings described in these papers as well as the one above.

Zinke, J., J. P.. 2018. Gradients of disturbance and environmental conditions shape coral community structure for south-eastern Indian Ocean reefs. *Diversity and Distributions* 24:605-620.

Vercammen, A., J. 2019. Evaluating the impact of accounting for coral cover in large-scale marine conservation prioritizations. *Diversity and Distributions* 25:1564-1574.

McClanahan, T. R. 2020. Coral community life histories and population dynamics driven by seascape bathymetry and temperature variability. Pages 291-230 in B. Reigl and P. W. Glynn, editors. *Advances in Marine Biology: Population Dynamics of The Reef Crisis*. Academic Press, London, UK.

Gap 5 -see recent papers by Lisa McManus for good recent examples. Also see Selmoni for

compilation papers for empirical examples.

A good example of seascape genomics can be found here:

Selmoni, O., E. Rochat, G. Lecellier, V. Berteaux-Lecellier, and S. Joost. 2020. Seascape genomics as a new tool to empower coral reef conservation strategies: An example on north-western Pacific *Acropora digitifera*. *Evolutionary Applications* 13:1923-1938.

Ecological shifts are summarized here but Tebbett et al. 2023 found little evidence for few algal shifts apart from the Caribbean.

Selmoni, O., G. Lecellier, V. Berteaux-Lecellier, and S. Joost. 2022. Worldwide analysis of reef surveys sorts coral taxa by associations with recent and past heat stress. *Frontiers in Marine Science* 9:948336.

L378 – one of the greatest uncertainties is that many empirical studies are finding thermal thresholds are not predicting coral cover. So, most thermal stress models must be wrong. Here is a recent example but also see Vercaemmen and McClanahan and Azali as other good examples.

Santana, E. F. 2023. Turbidity shapes shallow Southwestern Atlantic benthic reef communities. *Marine Environmental Research* 183:105807.

I am not sure figure 2 is adding much and may just be confusing. Reconsider how to present this concept of uncertainty and the 7 topics covered.

L466 – machine learning is being used to solve these problems, see Santana, McClanahan and Azali, and Vercaemmen as uncited sources of this approach.

Reviewer #3 (Remarks to the Author):

This paper reviews four modelling approaches for projecting future coral and reef responses to thermal stress, with a focus on the uncertainties embedded in the approaches and the need to be cognizant of these for making generalized comments about the potential futures for coral reefs. In this it provides a useful update of the state of science.

I make a number of comments to help clean up the manuscript – I think the analysis is useful and the findings are a timely reminder on the things to consider. But at the end, noting comments on Table 1 I'm not sure this is an authoritative systematic review of modelling studies, or more based on the authors' knowledge of the field – and with a bias towards the Great Barrier Reef. In this, the findings may not be as robust as if systematic review were done. In fact, lines 52-53 are the only 'methods section' in the paper, and the authors must add specification here of how they canvassed the literature to select these studies.

Significant but not challenging edits are suggested here.

Detailed comments, by line

13 should insert 'earth system tipping elements' otherwise it's not clear or explicit

20 'intersect with less-severe projections' ... this implies intersection of different projections with very different outcomes, but it's really that the individual projections have very broad uncertainty ranges. Make this match much more explicitly the main finding.

22 'within the next few years' ... there's nothing in the text about this, so either delete here, or incorporate into the discussion/conclusions

24-25. "evaluate whole-reef responses in the context of multiple pressures, and predict adaptive processes with greater " see my comment in gap 2, but you also address the scaling from organisms via population/community to reef, so consider how to incorporate

this based on the revision made.

26, 'time critical' is colloquial, do you mean 'urgent'?

64 - 2oC threshold of degree heating months (2o C-months). Mixup of units in this phrase

117-127 very well put

177-210- Gap 2 addresses variability in 'coral reef responses' and does mention 'coral' rather than 'coral reef' responses. I feel it needs to be more carefully worded and explicitly acknowledge the variation that starts with individual genes and genotypes, then aggregates through species populations and multi species communities to a 'coral reef' response (which really means the aggregate of coral communities in a defined spatial area, the 'reef'). Otherwise, 'coral reef' means all the non-coral species as well, which is not what the section is about.

243- wouldn't coral diseases and competition be more logical direct-interactors for bleaching responses? Not necessary to change, but the two listed strike me as less direct/priority choices.

277 is co-correlated a word? Either correlated, or covary?

282 "some reef ecosystems may be adapting naturally" ... rephrase as it is certain that species and ecosystems adapt, the question is in what ways and how much.

283 "coral bleaching tolerances increased by ~0.5o C despite increases in the intensity and frequency of heatwaves over the same timeframe" ... isn't this BECAUSE OF!! Must be rephrased.

297 "a pathway for stabilization of radiative forcing that assumes climate policies" . All of them assume policies, the issue is which policy?

381-394 the discussion on uncertainty is good, but I don't see the value of the figure. It may be useful if the source figures are familiar to the reader, but not particularly to this unfamiliar reader. The y axis orientation is not obvious, nor the overlaps or scaling along the x axis.

This should also be figure 3 not 2, so the in-text citations need correcting also.

425 there's no table 2

444 but it might be good to note that the 50 'reefs' in that study is quite a large scale region (100s of km) compared to the 'reef' implied in this study, which is more at the 11-10 km scale?

451 why this sudden introduction of 'random' here, without discussion it in the text. The question about the uncertainties is appropriate but understanding and reducing these is more the point than suddenly wondering if they're random?

470-472 ... repetitive use of the word 'efforts'. I think the first one can be deleted, as it could have multiple meanings.

Table 1. I wonder how fully comprehensive this table is (and see general comment on methods) ... is it studies that the authors are aware of? They don't list a systematic review approach to ensure they picked up everything that might be relevant, so more on this in the methods would be useful. There does appear a strong bias to the GBR once the global studies are discounted, which limits the more general relevance.

Table 1- I don't know if this class of study fits, but they certainly use several of the models described here - the Red List of Ecosystems might fall somewhere near groups 2 and 4 in approach, but also use group 1, and some explicitly use models in the fifth criterion. There are 4 studies now - on Caribbean, meso-American reef, western Indian Ocean and Colombian reefs. A further factor of importance is that the study wants to advance coherence across studies, and use of model results in decision-making, and the RLE incorporates this in its methodology and approach (approaches to combine disparate

results) and targeted outputs for policy/management. It is now also one of the few headline indicators in the new CBD biodiversity targets, so is a top priority indicator for application in coming years and including it will help with the policy relevance of the paper.

The RLE as a type of assessment? Are there other similar approaches based on risk?

Incorporates several of the approaches identified here, particularly if crit E can be evaluated as in MAR

Coral reef RLE refs:

Keith, D. A. et al. Scientific foundations for an IUCN Red List of Ecosystems. *PLoS ONE* 8, e62111 (2013).

Bland, L. M. et al. Using multiple lines of evidence to assess the risk of ecosystem collapse. *Proc. R. Soc. B* 284, 20170660 (2017).

Obura D, et al (2021) Vulnerability to collapse of coral reef ecosystems in the Western Indian Ocean. *Nat Sustain.* <https://doi.org/10.1038/s41893-021-00817-0>

Uribe, E. S., Luna-Acosta, A. & Etter, A. Red List of Ecosystems: risk assessment of coral ecosystems in the Colombian Caribbean. *Ocean Coast. Manag.* 199, 105416 (2021).

LOCATION	COMMENT	RESPONSE AND ACTION(S)
Reviewer 1		
Overall summary	This is a pretty good summary of limitations regarding the modeling of climate change on reefs. Overall, I agree with the majority of the authors' conclusions and the identification of major limitations, I was not convinced that a clear pathway forward has been suggested. In the majority of cases there is an ask for more studies and data, which is fine but does not necessarily help someone prioritize where to focus.	We thank Reviewer 1 for their thorough and constructive review of our manuscript. We agree that the previous manuscript did not provide a clear pathway forward. To address this, we undertook a full systematic review of the literature to portray the current-state of the field, identify key gaps in knowledge, and make clear recommendations. Briefly, we conducted a systematic review following the Preferred Reporting Items for Systematic Reviews and Meta-Analyses (PRISMA) guidelines and identified 79 articles providing projections for coral reefs. In doing so, we show that there is a lack of coordinated efforts and make recommendations for the need to consolidate efforts towards common goals. These recommendations now include the need to select robust, agreed-upon variables to project reef vulnerability to climate change, which would enhance comparability among modelling efforts and their compatibility with real-world observations (Pages 13-14, Lines 318-342). We undertook an exploratory meta-analysis of projections to examine variations in the magnitude of effects of projected effects. Crucially, we used a meta-analytical method to minimize confounding differences in approaches to show that the extent of projected impacts varies considerably among the methods and projection variables employed (Pages 14-18, Lines 344-432). As a result, a key conclusion of our review is the need to establish improved coordination to achieve consistency across modelling efforts, for which we make specific recommendations. We further show that most modelling studies lack the necessary resolution to provide insight for local management and efforts, for which there is increasing demand and discuss methods that will help improve projection resolution (Page 18-20, Lines 435-467). We also identified clear geographic biases of higher-resolution modelling efforts towards few regions (including Eastern Australia and Caribbean), providing a clear pathway forward to future modelling efforts in understudied coral reef provinces (Pages 19-20, Lines 470-500). By analyzing the stressors investigated in the reviewed articles, we also show that most models fail to include scenarios of management and intervention and provide key examples of how modelling approaches have been used to successfully inform decision-makers (Pages 21-22, Lines 502-552). Overall, we highlight the current strengths and limitations of the field, discuss the benefits of more coordinated efforts, and identify key areas that need urgent attention.

Major comment 1	I was not compelled by the conclusion that we should be using hierarchical models. This attracts very little explanation in the text and seems to suffer a number of limitations - just as any other approach does. These are not discussed. In general, a hierarchical Bayesian model is nothing special - just a statistical analysis at multiple levels. It still has the same limitations of any statistical approach in that it is often not mechanistic and therefore it is difficult to project values outside the bounds used to create the model. And while good statistical models include a mechanistic equation, not all such equations are truly mechanistic. For example, the logistic, which forms the foundation of population modelling, is just a mathematical convenience for an s-shaped function. It doesn't intrinsically include mechanisms.	We agree that this recommendation was inappropriate and removed it from the text. Instead, we now identify the major approaches used to simulate coral reef futures via a systematic review of the literature and review the key strengths and limitations of each approach (Pages 5-10, Lines 95-247).
Major comment 2	To me an important omission of the paper is the consideration of study objectives - is it about generic principles or more site-specific projection of possible futures? Tackling generic principles with simple models is appealing and it is relatively simple to capture the sources of variability in the models. BUT they do not adequately represent the biological uncertainty in their behaviour. I have seen many examples of practitioners and other scientists reading far too much realism into the results of such models, even to the point of using them for prediction. How should we handle the relegation of an ecosystem to a simple representation? What sorts of questions should be tackled with these models? Moreover, if we reduce the ecosystem to a cartoonish simplification, are the models able to tell us anything novel? To be clear, I use such models myself so I'm not against them but feel like the limitations need to be clear.	We agree that we did not adequately acknowledge study objectives in the previous version of our manuscript, which is crucial to their interpretation. We now discuss the key strengths and limitations of the major approaches and provide examples of how some do not aim for realism, but rather provide crucial insights into how ecological and biological processes may change in the future. In this section of the text, we also provide practical examples of particular studies have been used to inform decision-makers. See example on Pages 8-9, Lines 197-201. Finally, we acknowledged the diverse range of studies by summarizing the major objectives of each study in our database, which is included in Supplementary Data 1.
Major comment 3	My other concern is that the handling of uncertainty is still problematic. I like the way the paper articulates major sources of	We agree with Reviewer 1 that a major risk of reporting such massive uncertainties is its potential to undermine modelling efforts. As such, we have removed this recommendation from the text. In our

this. But I am yet to see a clear pathway for how to communicate such massive uncertainty in a meaningful way that doesn't undermine the purpose of the entire modelling exercise. Simply being able to capture and propagate uncertainty in a stochastic manner across the model system does not necessarily help someone make a decision. It really needs to return to the objective of the study and how can uncertainty analysis inform the analysis / decision / conclusion. Indeed, I often hear colleagues asking for greater analysis of uncertainty but yet there are few cases where such analyses have added significant value. To me the paper has an opportunity to make a contribution here. I like Fig 2 and the treatment of uncertainty so far. Can you go further and provide a clear pathway for its use?

systematic review of the field, we detected major inconsistencies in the reporting of projection estimates, uncertainty measures, and statistical power (Pages 15, Lines 357 -365 & Page 18, Lines 419-432). Given the critical importance of these model statistics for interpretation, repeatability, and wider syntheses (such as the one attempted in our review) we now advocate for improved reporting practices. We provide examples of how agreed-upon standards have been implemented in other fields as a clear pathway towards more coordinated efforts (Page 17, Lines 419-432). An excerpt from this section of the text reads as follows:

“One of the fundamental steps towards improving future syntheses of modelled projections for coral reefs is the adherence to essential reporting standards. While all studies in our database provided ample data to facilitate interpretation of the study outcomes, most (89% of studies) failed to report either statistical power or sufficient data for uncertainty measures to be converted into uniform metrics. In many cases, challenges arose from the display of results in figures and geographical maps that were not accompanied by adequate supplemental information reporting extractable values. Recognizing the necessity for reporting standards, other fields focused on projecting climate change impacts to biological systems have implemented agreed-upon standards¹⁻³. For instance, the International Union for Conservation of Nature (IUCN) established preliminary reporting standards for species threat assessments based on SDMs⁴, which have been further refined in subsequent publications^{1,2}. In light of the observed difficulties encountered in our analyses, we urge scientists projecting coral reef futures under climate change to develop and implement standards for the effective communication of projections and uncertainty associated with scenarios and models. “

We removed Figure 2 to address critique from Reviewer 2.

Minor comment 1

The section on eco-evo models does not criticize any of the limitations of such models which makes things a little uneven. They have massive uncertainty in terms of the taxa responsible for adaptation (coral, symbionts, etc.), the complexity of traits and genes that might encode thermal tolerance, and the biological simplification of reality in the ecology.

As suggested, we now discuss common limitations of eco-evolutionary models. The text reads as follows (Page 8, Lines 184-196):

“...Building upon similar frameworks used in population dynamic models, more recent studies⁵⁻⁷ incorporate species interactions and their abilities to adapt and disperse across diverse environments. While nearly a third of these studies formulated global-scale projections (Supplementary Tables 1 & 2), a significant challenge lies in the requirement for knowledge of taxa-specific traits, genetic adaptation, and ecological dynamics, which is lacking for most coral species and locations. As with population dynamic models, the reliability of their projections for non-focal taxa and regions can be influenced by parameter estimations and assumptions incorporated into these models⁷.

While these studies do not aim to achieve spatial or ecological realism, they provide essential insights into the potential role of adaptation and key environmental drivers that help to inform conservation planning at local and regional scales⁷...”

Minor comment 2	I felt that fig 1 needed further discussion to make its value clearer to the paper.	We removed Figure 1 and conducted a more robust, meta-analytical approach. This approach provided standardized measures to quantify the magnitude of the effect and enable comparisons while minimizing confounding differences. The results of this analysis are now delivered in Figure 3 and discussed on Pages 14-17, Lines 344-417. We also provide a figure showing the underlying data used in the analysis (Figure 4).
-----------------	---	--

Minor comment 3	Agree with the discussion of limitations of measuring thermal stress. You missed a recent paper by Lachs 2021 that did a global analysis of bleaching fitting a better DHW algorithm. Not a big deal but it complements the McClanahan stuff.	We thank Reviewer 1 for highlighting this important paper and now reference the study on Page 12, Lines 297-301. This section of the text reads as follows: “...It should be acknowledged, however, that many threshold based-techniques have evaluated how threshold choice affects model output ⁸⁻¹³ . For example, one study reported divergent estimates of bleaching onset timing when different inter-annual variation thresholds were used ⁸ . Overall, these findings further emphasize the importance of variable selection procedures in the outcome and power of the modelling process ¹⁴⁻¹⁸ .”
-----------------	---	--

Reviewer 2

Overall summary	The paper reviews the literature on coral responses to thermal stress and argues there are many sources of uncertainty that make future predictions difficult. This topic is important as some people outside of the field have claimed high confidence in predictions of coral reef response models to climate change (i.e., McKay et al. 2022). Those in the field will be aware of the “gaps” in confidence that the authors elucidate. In many ways this “review” summarizes many of the discussion sections of these papers and brings them together in a single perspective. This is a potentially useful product. However, I have many problems with the paper that would require a major revision if they can be addressed. Because critical readers cannot check or confirm to	We thank Reviewer 2 for their constructive critique of our work. As suggested, we conducted a full systematic review of the literature to provide a robust evaluation of the field (please see our response to Reviewer 1, Comment 1 for further details), which has required significant effort and time. Accordingly, our conclusions and recommendations are based on findings derived from the systematic review. We further provide supplemental information supporting the key findings in the text to enhance transparency and use of the outcomes presented (see Supplementary Information & Supplementary Data 1). Please see below for specific responses and actions, however, the text has undergone significant revisions and many of the specific comments are no longer applicable, as the statements originated the comments have been removed. We thank Reviewer 2 for these specific comments because these critiques have been considered in the writing of the new version.
-----------------	---	---

	the authors conclusions as currently written, the paper should be rejected. That is there is a poor connection between the authors conclusions and the cited literature and no quantitative analysis of the literature.	
Major comment 1	A major problem is the lack of a systematic review that would force the authors to consider the full literature in this field. The ideas and citations are typical of this field where some of the same authors and papers get cited repeatedly, even if they are not the best papers on these subjects. Often, they reflect poor science but get cited because they are provocative. The climate change literature is full of provocative papers that will not stand up to scrutiny. Then, many papers directly relevant to the topics are ignored and not cited even though they make very strong contributions to very specific issues. There are many recent papers that are not cited and cases where papers are cited in the reference section but not in the text where the contributions were made. The text suffers this problem throughout and leads to a disappointing coverage of issues. The authors probably know this, but they repeat this behavior as it may be accepted to be provocative rather than analytically rigorous.	We agree with Reviewer 1 and thoroughly reviewed the literature to incorporate missing articles. We conducted a systematic review to ensure our conclusions are based on an analytical assessment. We further agree with Reviewer 1's sentiments that certain contributions are often ignored despite making significant and rigorous contributions to the field. To address this, we undertook an analysis of citation practices. This analysis (shown in Figure 1 on Page 6) revealed that 'excess heat' threshold models are disproportionately cited relative to other rigorous approaches. We compliment this analysis by reviewing the objectives, key strengths, and limitations of the five major approaches (Pages 5-10, Lines 95-247).
Major comment 2	Most of the text comes to sweeping conclusions without citing the studies from where some of these conclusions arose. I suggest the authors see several recent reviews, meta-analysis, and papers of which I list some below. They should also give specific examples of studies and consider putting these supporting and unsupportive studies in a table where we can consider the evidence ourselves. They should not come to categorizations and conclusions themselves based on an incomplete review of the literature. They need to quantify and disarticulate the many papers and responses of corals in these models. For example, being clear about the differences in	Please see Reviewer 2, Comment 1. We specifically address Reviewer 2 and 3's concerns regarding the need to quantify and disarticulate the variety of currencies used to project coral reef vulnerability by examining which variables (and units) were used. This section of the text reads as follows (Pages 13-14, Lines 319-342): "...Despite the growing body of studies forecasting coral reef futures, there is no broad consensus on the optimal variables for projecting coral reef vulnerability¹⁹⁻²¹. This lack of consensus is reflected by discrepancies in the type of variables that have been simulated (Supplementary Table 1). Ideally, different models should simulate common variables that can be readily measured in monitoring programs. This, in turn, would allow for meaningful model validation and comparisons between independent models, which

thresholds, bleaching responses, coral cover, genomic and community change. Responses tend to get lumped together and yet they are not the same. Words like uncertainty are not well defined and so we are not certain what is being evaluated and by what means. This makes the paper hard to review for its scientific veracity and should lead to rejection.

are crucial for increasing the accuracy and precision of projected climate change impacts on biological systems^{3,22,23}. These sentiments are further emphasized by calls of Red List of Ecosystems-based studies for more coordinated monitoring of coral reefs and improved data quality and quantity¹⁹⁻²¹.

Our analysis shows that a large proportion of the reviewed studies formulated their projections using variables that are challenging to verify against real-world observations (Supplementary Table 1). More than half of the studies employing 'excess heat' thresholds presented their projections in terms of fractions of reef cells at risk (52%), while SDMs typically provided estimates in terms of fractions of reef cells with suitable habitats or relative changes in habitat suitability (69%) (Supplementary Table 1). Although coral cover serves as a widely used and accessible indicator for this purpose^{19,24}, projections of coral cover were delivered in less than a third of all published studies (29%). Even within these studies, inconsistencies arose in the measures utilized and semantic ambiguity as to what the absolute magnitude of coral cover these measures referred to. The inconsistent measures included proportions of coral cover, proportional changes in coral cover, probability of changes in coral cover, and the probability of low coral cover (Supplementary Table 1). Such discrepancies in metric choices can present challenges when comparing model projections and translating them into field measurements of coral cover."

We also clarify key differences in the major approaches to clarify that different studies vary in their objectives and the variables projected, so as not lump projections together (see Pages 5-10, Lines 95-247, Pages 13-14, Lines 319-342, and throughout).

We now clarify throughout the text which measures of uncertainty we are referring to.

Major comment 3

The figures are not that useful or hard to interpret. The authors need to make it clear what they did, for example, to produce figure 1. This seems to be a result without a method that can be checked for veracity. Figure 3 is not helpful and should be based on quantified information rather than a cartoon or schematic drawing. The supplementary table was disappointing as it seemed so like the table in the text. I was hoping to find evidence for these categorizations with specific citations of literature. This table seems to lack quantification and is the

We agree that Figure 1 and the underlying procedures used required a more rigorous approach. We thus used our systematic review database to identify articles that provided sufficient data to be included an exploratory meta-analysis of projections (see Methods on Page 24-26, Lines 574-643). This approach permitted an examination of variations in the magnitude of effects of projected effects. Although exploratory, we show that the extent of projected impacts varies considerably among the methods and projection variables used (Pages 14--17, Lines 344-405). Importantly, we consider this analysis exploratory because our dataset was restricted by the unavailability of projection estimates, measures of model output uncertainty (i.e., standard deviation, 95% CI, range), and statistical power. The limitations of this

	opinion of the authors, based on an incomplete reading of the literature.	analysis are discussed on Pages 14--17, Lines 344-405 and we provide supplemental information (Supplementary Table 3) that lists the studies and data sources.
Major comment 4	The English composition is weak, the paper is not a pleasure to read, and the paper should be rewritten once the literature is reviewed systematically and updated.	As suggested, we made significant revisions to enhance readability of the text and conducted a systematic review of the literature.
Specific comment 1 (Abstract)	I don't find the abstract very informative as it lacks details. It reflects or sounds similar to many of the discussion sections of papers on climate and corals, where there is a call for better resolution and knowledge of adaptation. It seems like a time to give more details and specific suggestions.	We rewrote the abstract to reflect the finding of the systematic review, which now includes specific details and suggestions. The abstract now reads (Page 1, Lines 13-33): “Influential syntheses of climate change impacts consistently conclude that limiting global warming to 1.5°C is unlikely to save most of the world’s coral reefs. However, they fail to elaborate on the extent of uncertainties and confidence levels associated with projections supporting these conclusions. To address this, we synthesized two decades of research effort projecting coral reef futures under climate change using a systematic review approach. Our quantitative analysis of the resulting 79 articles revealed a surprising lack of coordination among studies, which poses a significant challenge in integrating the results to reach reliable and comparable conclusions. We found that discrepancies in the magnitude of projected impacts were influenced by the choice of approach and response variables. These findings suggest that relying on projections from a few key studies could result in conclusions that misrepresent the slate of available projections, calling for caution and more transparent acknowledgment of uncertainty measures around projection estimates. To improve coordination and enhance the consensus across future projections, we advocate for standardized reporting of projections and measures of associated uncertainty, as well as improved coordination among studies to focus on variables that can be measured and verified through existing efforts to monitor coral reefs. Our review revealed significant geographical gaps in higher-resolution models for specific regions and emphasize the incompatibility of most projections with practical scales for management and intervention strategies, emphasizing the need for more downscaled projections. We stress the potential value of models as tools for informing management and interventions, but their effectiveness depends on coordinated efforts to close existing knowledge gaps and raise scientific confidence in coral reef projections.”
Specific comment 2 (Abstract)	“that existing projections reflecting a near-complete loss” The authors need to be specific about which projections. There	This statement was removed from the text.

	are different methods used to make predictions, but most are excess heat predictions with some that include ocean acidity.	
Specific comment 3 (Introduction)	Many paragraphs are not really paragraphs. English composition need attention throughout the manuscript, many short and long paragraphs are intermixed without much structure.	The manuscript has undergone significant review to enhance its structure and organization.
Specific comment 4 (Introduction)	Why not set up the introduction relative to this paper, rather than the usual Hughes papers?	We thank Reviewer 2 for the nice suggestion and rewrote the introduction in the context of Armstrong's 2022 paper of climate tipping points. (Pages 3-4, Lines 36-92).
Specific comment 5 (Introduction, Line 45)	L45 – Some of the citation seems selective or old in this rapidly moving field. For example, the increased frequency of thermal thresholds is repeated in many papers but the evidence is not strong. See, for example, this paper where events are show to be not more frequent but larger scale and longer duration. Armstrong McKay,. 2022. Exceeding 1.5 C global warming could trigger multiple climate tipping points. Science 377:eabn7950. Skirving, 2019. The relentless march of mass coral bleaching: A global perspective of changing heat stress. Coral Reefs 38:547 - 557.	We thank Reviewer 2 for highlighting this point and the supporting paper, which is now cited and discussed on Page 11-13, Line 272-317-. An exert of this section of text reads as follows: (Page 12, Lines 282-302) "One explanation for the poor fit of these threshold metrics to field data is their inability to capture different marine heatwave characteristics, such as peak temperatures, duration, and rates of heating. For instance, a historical assessment spanning from 1985 to 2017 examined variations in SST and revealed that increases in accumulated heat stress, as measured by two common threshold metrics, were predominantly attributed to longer heating events affecting wider areas²⁵. However, the study could not detect changes in peak temperatures or event frequencies during the analyzed period, indicating the limitations of the metrics in capturing changes in different heating variables²⁵. A study by McClanahan and colleagues²⁶ evaluated the effectiveness of heatwave variables in explaining bleaching severity on 226 coral reefs and found that the DHW metric explained only 9% of the model variance. In contrast, peak temperatures, the duration of cool temperatures, and temperature bimodality were found to be stronger predictors of bleaching severity. Several empirical studies have also reported that approaches based on thermal thresholds face difficulties in predicting changes in coral cover^{18,27,28}. For example, a study investigating the power of 27 environmental factors in explaining changes in coral cover on Indian Ocean reefs reported that temperature anomalies, temperature variation, and the duration of cyclones were the best predictors¹⁸. It should be acknowledged, however, that many threshold based-techniques have evaluated how threshold choice affects model output⁸⁻¹³. For example, one study reported divergent estimates of bleaching onset timing when different inter-annual variation thresholds

		were used ⁸ . Overall, these findings further emphasize the importance of variable selection procedures in the outcome and power of the modelling process ¹⁴⁻¹⁸ . “
Specific comment 6 (Introduction, Line 48)	L48 – local stressors often include background conditions. Why say otherwise when there are many papers that have specifically examined this topic. None cited. This is just one of many examples. 2021 Environmental variability and threshold model’s predictions for coral reefs. Frontiers in Marine Science 8:1774.	We did not intend to imply this and revised the text to clarify. We now acknowledge the numerous papers that have examined local, background conditions, including the paper suggested.
Specific comment 7 (Introduction, Line 49)	L49: The 50% decline is not support here. Tebbett, S. B., 2023. Benthic composition changes on coral reefs at global scales. Nature Ecology & Evolution 7:71-81.	This statement was removed was from the revised manuscript. We cited this paper in other sections of the text, including Page 13, Lines 335-337.
Specific comment 8 (Introduction, Line 52)	L52 – how was this review done? The citation appears quite selective, so it is not clear that the authors have actually reviewed the full literature. See this more systematic review. Coral responses to climate change exposure. Environmental Research Letters 17:073001.	We now include a Methods section to provide details of how the systematic review was conducted. We also refer to and cite the recent systematic review by McClanahan, see example on Page 13, Lines 307-311.
Specific comment 9 (Table 1)	Table 1 might have some specific papers cited as examples. This table seems to be the creation of the authors without any analysis of publication statistics, such as word use of variables, etc. It is not really a meta-analysis but more about the authors interpretation and opinions about the literature. I see it as subjective and likely incomplete.	Reviewer 1 is correct that the previous version of our manuscript did not involve a systematic review of the literature. We undertook a complete revision to directly address these issues. Our search of the literature retrieved 2,075 articles and we identified 79 articles providing quantitative projections of coral reef futures to climate.
Specific comment 10 (Line 62)	L62 – It this a title? Seems to be an incomplete sentence, so maybe put them in italics or bold to be clear they are title to sections.	We rewrote this section of the text to improve structure (Pages 5-10, Lines 95-247).
Specific comment 11	Any model that predicts the frequency of bleaching events must contend with the fact that frequency has not changed in most regions, as described by Skirving. These models require rates of	We thank Reviewer 2 for highlighting this key point and re-wrote the text in three places. (Pages 3-4, Lines 55-82)

SST rise and threshold information, so best to describe them by how they were built. The paper lacks much consideration of how these models were built, what were their key assumptions, etc.

“The latest synthesis identified a CTP of 1.5°C (1 to 2°C, high confidence) for tropical coral reefs, with an estimated timescale for dramatic change of 10 years with medium confidence²⁹. In high agreement with findings of the Intergovernmental Panel on Climate Change (IPCC)^{30,31}, the synthesis cited independent modelling efforts using similar ‘excess heat’ modelling approaches as the basis of the assessment³²⁻³⁵. These approaches apply thresholds— in the form of degree heating weeks or months – that represent an accumulation of excess heat above baseline summer conditions. These thresholds are then applied to sea surface temperatures (SSTs) and forced by different emissions scenarios in an effort to retrieve the likelihood of future bleaching events³²⁻³⁵. Such models analyze the frequency of bleaching events and estimate the proportion of reef locations at risk of ‘long-term degradation’ or ‘severe bleaching events’, producing estimates with high coherence among studies³²⁻³⁵. The resultant CTP of 1.5°C (1 to 2°C) places warm-water reefs among the closest elements to their tipping points²⁹, aligning with the conclusions of the Working Group II’s contribution to the IPCC’s AR6 report³⁰. This finding raises concern over imminent impacts to marine biodiversity, human livelihoods and the effectiveness of interventions to alleviate further coral reef degradation.

The earliest studies to forecast coral reef responses to future global warming utilized ‘excess heat’ threshold approaches^{10,15,36,37}. Put simply, these methods operate under the notion that widespread bleaching predictably occurs when temperatures accumulate beyond a specific threshold. Although several investigations show that ‘excess heat’ threshold metrics have limited predictive power when applied to historical bleaching records¹⁵⁻¹⁷, this approach continues to prevail in the field¹⁵. In the mid to late-2000s, a consensus emerged that differences in bleaching susceptibility between locations were best explained by multiple modifying variables¹⁵, which eventually led to development of alternative model-based approaches. Since then, various approaches, such as species distribution models, ecology-evolutionary models, and models of reef population dynamics have been applied. However, influential syntheses of climate change impacts largely ignore these developments to prioritize projections derived exclusively from ‘excess heat’ threshold approaches^{29,30,38-40}.”

(Page 5, Lines 108-123)

“‘Excess heat’ threshold models integrate thermal threshold metrics assumed to predict the likelihood of severe coral bleaching with future sea surface temperature (SST) projections to forecast future instances of bleaching events. These models usually adopt a specific frequency of events exceeding the thresholds,

such as two severe bleaching events per decade^{12,32,41}, that is estimated to preclude long-term recovery. This assumption permits the estimation of reef cells (e.g., 0.5° × 0.5° pixels on the Earth’s surface) that are at risk of ‘long-term degradation’^{32,33} or ‘severe bleaching events’⁴²⁻⁴⁴, according to the threshold and frequency of events set.”

(Page 12, Lines 282-289)

“One explanation for the poor fit of these threshold metrics to field data is their inability to capture different marine heatwave characteristics, such as peak temperatures, duration, and rates of heating. For instance, a historical assessment spanning from 1985 to 2017 examined variations in SST and revealed that increases in accumulated heat stress, as measured by two common threshold metrics, were predominantly attributed to longer heating events affecting wider areas²⁵. However, the study could not detect changes in peak temperatures or event frequencies during the analyzed period, indicating the limitations of the metrics in capturing changes in different heating variables²⁵...”

We also acknowledged the diverse range of studies by summarizing the major objectives, strengths and limitations of each study in our database, which is included in Supplementary Data 1.

Specific comment 12
(Line 64)

L64 – What models specifically are being discussed. There are a variety of model and many these days are using 4 DHW and not modelling at the month. Thus, the use of month and 2 degrees sounds like a selection of some of the older models. A 1-degree threshold is very common for bleaching studies, so make it clear if this is a cumulative or simple threshold. The 1 degree was first suggested here. However, many recent studies do not support this suggestion. See for example, papers by DeCarlo. Hoegh-Guldberg, O., and B. Salvat. 1995. Periodic mass-bleaching and elevated sea temperatures: bleaching of outer reef slope communities in Moorea, French Polynesia. Marine Ecology Progress Series 121:181-190.

Please see below for our response to Reviewer 2, Comment 14, which addresses this critique.

Specific comment 13 (Line 67)	L 67 - Reefs at Risk is this the title and findings in the report, or is this a general statement based on the unknown papers being describe here? There are many studies reporting different threshold, so be clearer about what you are reviewing.	We agree and clarify this section of the text to explain how these models produce estimates of fractions of reef cells at risk and provide specific examples. This revised text reads as follows (P5, Lines 108-123): “Excess heat’ threshold models integrate thermal threshold metrics assumed to predict the likelihood of severe coral bleaching with future sea surface temperature (SST) projections to forecast future instances of bleaching events. These models usually adopt a specific frequency of events exceeding the thresholds, such as two severe bleaching events per decade^{12,32,41}, that is estimated to preclude long-term recovery. This assumption permits the estimation of reef cells (e.g., 0.5° × 0.5° pixels on the Earth’s surface) that are at risk of ‘long-term degradation’^{32,33} or ‘severe bleaching events’⁴²⁻⁴⁴, according to the threshold and frequency of events set.” In other sections of the text, where these studies are synthesized, we provide supplemental material listing each study to enhance transparency. See Supplementary Tables 1 & 3.
Specific comment 14 (Line 68)	L68 – this is important but not clear as written. Thresholds are to predict bleaching not coral survival, so do not assume too much about their predictions unless addressing specific studies. Is the 99% lost are is it that they will bleach every year? This vagueness in responses is a problem that needs to be rectified.	We thank Reviewer 2 for highlighting these issues. We removed the statement that these projections represent coral survival, which was incorrect and revised the text to acknowledge the underlying differences and assumptions between studies. We no longer imply that these studies use the same thresholds or assumptions. This was a clear oversight. This part of the text now reads as follows, where specific examples are provided (Page 5, Lines 108-123): “Excess heat’ threshold models integrate thermal threshold metrics assumed to predict the likelihood of severe coral bleaching with future sea surface temperature (SST) projections to forecast future instances of bleaching events. These models usually adopt a specific frequency of events exceeding the thresholds, such as two severe bleaching events per decade^{12,32,41}, that is estimated to preclude long-term recovery. This assumption permits the estimation of reef cells (e.g., 0.5° × 0.5° pixels on the Earth’s surface) that are at risk of ‘long-term degradation’^{32,33} or ‘severe bleaching events’⁴²⁻⁴⁴, according to the threshold and frequency of events set. Although these models have the advantage of utilizing a method that can be applied to broad geographical scales and incorporate other moderating factors without the need for detailed in situ data, they rarely perform any direct assessments of biological or ecological processes^{32,33,45-47}. This approach was the most prevalent model type in our analysis (32%) (Figure 1a) and attracted a disproportionately higher number of cumulative citations (68%) than all other model types (Figure 1b). This trend can be partly attributed to this method’s dual role as the foundation for satellite products that

are used to alert the risk of coral bleaching^{48,49}, and its widespread adoption as the primary method for global-scale projections in the field (Supplementary Tables 1 & 2).”

Specific comment 15 (Line 75)	L75 – This text seems dated in the context of some recent papers, such as. McManus, L. C., 2021. Evolution and connectivity influence the persistence and recovery of coral reefs under climate change in the Caribbean, Southwest Pacific, and Coral Triangle. Global Change Biology 27:4307-4321. Would niche models fit this category? Do they require another category? van Woesik, R., S. Köksal, A. Ünal, C. W. Cacciapaglia, and C. J. Randall. 2018. Predicting coral dynamics through climate change. Scientific Reports 8:17997. Cacciapaglia, C., and R. van Woesik. 2018. Marine species distribution modelling and the effects of genetic isolation under climate change. Journal of Biogeography 45:154-163.	These papers are now included in our systematic review of the literature. We agree with Reviewer 2 that species distribution models warrant another category. The text has been updated throughout to include these efforts. We thank Reviewer 2 for this suggestion.
Specific comment 16 (Line 103)	L103 -are they predicting losses of coral cover or bleaching event frequencies? This paper connects bleaching to coral losses but I don't believe the authors are referring to this paper. Cornwall, C. E., et al. 2021. Global declines in coral reef calcium carbonate production under ocean acidification and warming. Proceedings of the National Academy of Sciences 118:e2015265118.	We removed this statement from the text. However, the suggested study is cited in the text and we thank Reviewer 2 for this suggestion.
Specific comment 17 (Figure 1)	Figure 1 what does coral performance mean and how is it related to coral losses discussed in the text? Why would performance be	Please see response to Reviewer 1, Minor Comment 1 and the updated Methods section detailing our new analysis.

	a percentage? How many studies (n=?) were used to make these uncertainty estimates? What does uncertainty mean? This figure will not tolerate much scrutiny as presented.	
Specific comment 18	Gap text – English composition is weak here in that there are many long paragraphs lacking structure.	As suggested, we made significant revisions to enhance readability of the text and conducted a systematic review of the literature.
Specific comment 19 (Line 137)	L137 – changes in what, the characteristics of MHWs or in impacts on corals? It would be good to describe the underlying structure of the models, so one can understand the predictions better. What variables are being modelled, what are the key assumptions, etc. MHW seem to be defined by the duration of heat but what about the extent of the heat. The duration of warm events will occur in the future but coral sensitivity is also not likely to be responding to the same thresholds. The threshold for bleaching is increasing over time. Do these models consider this change in thresholds? Sully, S., D. E. Burkepile, M. K. Donovan, G. Hodgson, and R. van Woesik. 2019. A global analysis of coral bleaching over the past two decades. Nature Communications 10:1-5.	This section of the text has been revised, which has been renamed as “how heat stress is modelled” (Page 8, Lines 201 -260). We also acknowledged the diverse range of studies by summarizing the major objectives, strengths and limitations of each study in our database, which is included in Supplementary Data 1.
Specific comment 20 (Line 169)	L169. One wonders how the Skirving paper compares to the Li and Donner paper, especially in terms of the frequency of temperatures that pass certain thresholds. Are the authors being selective here or interpreting frequency correctly?	We agree that this should be acknowledge and discuss the findings of Skirving et al. 2019, as per our response to Reviewer 2, Specific Comment 5.
Specific comment 21 (Gap 2)	Gap 2 – is this variability in the models or the sensitivity of the corals or both. Not clear as written. The authors need to distinguish spatial from taxonomic variability. The word coral embodies a great deal of variability in taxa, evolution, and ecology, and yet this is not made clear.	We removed this section from the manuscript.
Specific comment 22 (Line 185)	L185 – The same data set indicated that coral taxa are having different adaptation rates and therefore reversal of responses over time. The authors could separate these responses in space,	We removed this section from the manuscript. Discussions of spatial variability have now been addressed in a separate section (see Pages 19-20, Lines 470-501).

time, and taxa better in this text.

McClanahan. 2020. Highly variable taxa-specific coral bleaching responses to thermal stress. *Marine Ecology Progress Series* 648:135 - 151.

Specific comment 23
(Line 193)

L193 – See the Tebbett et al. paper cited above for coral cover patterns over time and how the Indo West and Indian Ocean are showing little effects over time. Also, see the McClanahan *Frontiers in Marine Science* 2021 paper above where two models are compared and where large differences are reported.

We agree with Reviewer 2 and no longer compare how different models explain geographic variations in projections for coral reefs. Indeed, reviewer 2 is correct that these comparisons are inevitably confounded by differences in underlying model structure and objectives. As such, we take a higher-level approach to investigate geographic gaps in regional modelling efforts (see Pages 19-20, Lines 470-501).

The responses as differences in thresholds, bleaching, and coral cover are very seldom disarticulated in the paper, and yet they are not the same thing. Thresholds have been shown to differ geographically, but this is not covered or evidence given despite a number of papers on the subject.

DeCarlo, T. M. 2020. Treating coral bleaching as weather: A framework to validate and optimize prediction skill. *PeerJ* 8:e9449.

Coral is composed of many life histories with various responses. This blending of concepts creates vagueness that makes the paper hard to follow.

Darling, E. S., 2019. Social–environmental drivers inform strategic management of coral reefs in the Anthropocene. *Nature Ecology & Evolution* 3:1341-1350.

Specific comment 24
(Line 254)

L254 – One wonders how these mechanistic models compare with empirical data. There are some comprehensive models, so not sure I would call one the most comprehensive models. A

We removed this section of the manuscript.

number are not cited in this paper.

MacNeil, M. A., 2019. Water quality mediates resilience on the Great Barrier Reef. *Nature Ecology & Evolution* 3:620 - 627.

Specific comment 25 (Line 269)	L269 – I would suggest the Frontiers in Marine Science as doing most of what is claimed not to have been done here. Empirically, consider the findings described in these papers as well as the one above.	This section of the text has been removed and the recommended studies have been integrated into the revised text.
---	---

Zinke, J., J. P.. 2018. Gradients of disturbance and environmental conditions shape coral community structure for south-eastern Indian Ocean reefs. *Diversity and Distributions* 24:605-620.

Vercammen, A., J. 2019. Evaluating the impact of accounting for coral cover in large-scale marine conservation prioritizations. *Diversity and Distributions* 25:1564-1574.

McClanahan, T. R. 2020. Coral community life histories and population dynamics driven by seascape bathymetry and temperature variability. Pages 291-230 in B. Reigl and P. W. Glynn, editors. *Advances in Marine Biology: Population Dynamics of The Reef Crisis*. Academic Press, London, UK.

Specific comment 26 (Gap 5)	Gap 5 -see recent papers by Lisa McManus for good recent examples. Also see Selmoni for compilation papers for empirical examples.	We removed this section of the manuscript and these studies are included in our database.
--	---

Specific comment 27 (Line 319)	A good example of seascape genomics can be found here: Selmoni, O., E. Rochat, G. Lecellier, V. Berteaux-Lecellier, and S. Joost. 2020. Seascape genomics as a new tool to empower coral reef conservation strategies: An example on north-western	We removed this section of the manuscript and these studies are included in our database.
---	---

	Pacific Acropora digitifera. Evolutionary Applications 13:1923-1938.	
Specific comment 28 (Line 344)	Ecological shifts are summarized here but Tebbett et al. 2023 found little evidence for few algal shifts apart from the Caribbean. Selmoni, O., G. Lecellier, V. Berteaux-Lecellier, and S. Joost. 2022. Worldwide analysis of reef surveys sorts coral taxa by associations with recent and past heat stress. Frontiers in Marine Science 9:948336.	We removed this section of the manuscript.
Specific comment 29 (Line 378)	L378 – one of the greatest uncertainties is that many empirical studies are finding thermal thresholds are not predicting coral cover. So, most thermal stress models must be wrong. Here is a recent example but also see Vercaemmen and McClanahan and Azali as other good examples. Santana, E. F. 2023. Turbidity shapes shallow Southwestern Atlantic benthic reef communities. Marine Environmental Research 183:105807.	We thank Reviewer 2 for this suggestion and highlighting the newly published paper. This point is discussed on Page 12, Lines 282-302 and these studies cited, along with others.
Specific comment 30 (Figure 2)	I am not sure figure 2 is adding much and may just be confusing. Reconsider how to present this concept of uncertainty and the 7 topics covered.	We revised the text to adhere to a more systematic review format, as suggested, and thus Figure 2 was removed.
Specific comment 31 (Line 466)	L466 – machine learning is being used to solve these problems, see Santana, McClanahan and Azali, and Vercaemmen as uncited sources of this approach.	We have removed this recommendation from the text as per critique from Reviewer 1 (see response to Reviewer 1, Major Comment 1.)
Reviewer 3		
Overall summary	This paper reviews four modelling approaches for projecting future coral and reef responses to thermal stress, with a focus on the uncertainties embedded in the approaches and the need to be cognizant of these for making generalized comments about the potential futures for coral reefs. In this it provides a useful	We thank Reviewer 3 for their thorough review of our manuscript, which greatly improved the study.

update of the state of science.

Major comment 1	I make a number of comments to help clean up the manuscript – I think the analysis is useful and the findings are a timely reminder on the things to consider. But at the end, noting comments on Table 1 I'm not sure this is an authoritative systematic review of modelling studies, or more based on the authors' knowledge of the field – and with a bias towards the Great Barrier Reef. In this, the findings may not be as robust as if systematic review were done. In fact, lines 52-53 are the only 'methods section' in the paper, and the authors must add specification here of how they canvassed the literature to select these studies. Significant but not challenging edits are suggested here.	We have substantially revised the manuscript, which now reflects a systematic review. We have since removed Table 1 and include extensive supplementary information supporting the key conclusions in the main text. As suggested, we now also include a detailed Methods section of our analysis was conducted. We provide specific responses and actions to each comment below.
Specific comment 1 (Line 13)	13 should insert 'earth system tipping elements' otherwise it's not clear or explicit	We clarify this on Page 3, Lines 36-38. This section of the Introduction now reads: "Anthropogenic climate change is expected to propel elements of the Earth's system beyond climate tipping points (CTPs) that, once exceeded, are expected to induce self-perpetuating change and impacts across biophysical systems ²⁹ ."
Specific comment 2 (Line 20)	20 'intersect with less-severe projections' ... this implies intersection of different projections with very different outcomes, but it's really that the individual projections have very broad uncertainty ranges. Make this match much more explicitly the main finding.	We have removed this text and changed the method of this analysis. Specifically, we now take a meta-analytical approach which employs effect sizes. While this method does not pin-point differences in projection estimates per se, it analyses differences in the magnitude of effects. This analysis is delivered in Figure 3 and discussed on Page 14-17, Lines 345-406.
Specific comment 3 (Line 22)	22 'within the next few years' ... there's nothing in the text about this, so either delete here, or incorporate into the discussion/conclusions	We agree and removed this statement.
Specific comment 4 (Lines 24-25)	24-25. "evaluate whole-reef responses in the context of multiple pressures, and predict adaptive processes with greater " see my comment in gap 2, but you also address the scaling from	Please see below response to Reviewer 3, Specific Comment 8.

	organisms via population/community to reef, so consider how to incorporate this based on the revision made.	
Specific comment 5 (Line 26)	26, 'time critical' is colloquial, do you mean 'urgent'?	The term 'time-critical' was removed and replaced with "urgent", as suggested.
Specific comment 6 (Line 64)	2oC threshold of degree heating months (2o C-months). Mix-up of units in this phrase	This was corrected throughout the text.
Specific comment 7 (Lines 117-127)	Very well put	We thank Reviewer 3 for this kind comment. However, we have taken a different approach to this analysis and therefore have rewritten the text. We conducted a meta-analysis of projections to examine variations in the magnitude of effects of projected effects. We used this method to minimize confounding differences in approaches to show that the extent of projected impacts varies considerably among the methods and projection variables employed (Page 14-17, Lines 345-406).
Specific comment 8 (Lines 177-210)	177-210- Gap 2 addresses variability in 'coral reef responses' and does mention 'coral' rather than 'coral reef' responses. I feel it needs to be more carefully worded and explicitly acknowledge the variation that starts with individual genes and genotypes, then aggregates through species populations and multi species communities to a 'coral reef' response (which really means the aggregate of coral communities in a defined spatial area, the 'reef'). Otherwise, 'coral reef' means all the non-coral species as well, which is not what the section is about.	This section was removed from text. However, we now use the terminology "coral reef responses" consistently to refer to all variables that were simulated by the models, including some coral and non-coral variables. We also now include a section in the text that reviews the variables used in our systematic review and discuss inconsistencies. See Page 13-18, Lines 320-433.
Specific comment 9 (Line 243)	243- wouldn't coral diseases and competition be more logical direct-interactors for bleaching responses? Not necessary to change, but the two listed strike me as less direct/priority choices.	We thank Reviewer 3 for this insightful comment. Following this suggestion, we highlight the importance of both abiotic and biotic pressures, including coral disease (See Pages 21-22, Lines 502-553)
Specific comment 10 (Line 277)	277 is co-correlated a word? Either correlated, or covary?	This term was removed from the text.
Specific comment 11 (Line 282)	282 "some reef ecosystems may be adapting naturally" ... rephrase as it is certain that species and ecosystems adapt, the question is in what ways and how much.	This sentence was removed from the text.

Specific comment 12 (Line 283)	283 “coral bleaching tolerances increased by ~0.5o C despite increases in the intensity and frequency of heatwaves over the same timeframe” ... isn’t this BECAUSE OF!! Must be rephrased.	This sentence was removed from the text.
Specific comment 13 (Line 297)	297 “a pathway for stabilization of radiative forcing that assumes climate policies”. All of them assume policies, the issue is which policy?	This sentence was also removed from the text.
Specific comment 14 (Lines 381-394)	381-394 the discussion on uncertainty is good, but I don’t see the value of the figure. It may be useful if the source figures are familiar to the reader, but not particularly to this unfamiliar reader. The y axis orientation is not obvious, nor the overlaps or scaling along the x axis. This should also be figure 3 not 2, so the in-text citations need correcting also.	Thank you for this feedback. We decided to remove this figure based on critique of Review 2. Please see Reviewer 1, Major Comment 3.
Specific comment 15	425 there’s no table 2	This oversight was corrected. We have thoroughly checked our revised manuscript to ensure such errors are not repeated.
Specific comment 16 (Line 444)	444 but it might t be good to note that the 50 ‘reefs’ in that study is quite a large-scale region (100s of km) compared to the ‘reef’ implied in this study, which is more at the 11-10 km scale?	We removed this discussion point from the text.
Specific comment 17 (Line 451)	451 why this sudden introduction of ‘random’ here, without discussion it in the text. The question about the uncertainties is appropriate but understanding and reducing these is more the point than suddenly wondering if they’re random?	We agree this term was not adequately defined and was misleading as written. We now clarify and describe similar terms in our revised manuscript so as not cause confusion.
Specific comment 18 (Lines 470-472)	470-472 ... repetitive use of the word ‘efforts’. I think the first one Can be deleted, as it could have multiple meanings.	We carefully checked the revised version to avoid excessive use of this term.
Specific comment 19 (Table 1)	Table 1. I wonder how fully comprehensive this table is (and see general comment on methods) ... is it studies that he authors are aware of? They don’t list a systematic review approach to ensure they picked up everything that might be relevant, so more on this in the methods would be useful. There does appear a strong bias to the GBR once the global studies are discounted, which limits the more general relevance.	We conducted a full systematic review of the literature to provide a more quantitative analysis of the literature. We used a database of identified studies to draw key conclusions in our manuscript and provide detailed supplementary information of the specific articles used. For further details, please see Reviewer 1, Comment 1. We thank Reviewer 3 for highlighting a strong bias towards studies in the GBR in our manuscript. Given this, we undertook an investigation of the literature to understand whether geographic biases are present

in the studies providing forecasts for coral reefs. We quantified the number of studies providing regional-scale projections and their focal locations and detected a significant focus on regions such as the GBR and Caribbean (see Pages 19-21 and Figure 5). This analysis also revealed that knowledge of understudied coral reef provinces is relying on the coarser resolution of global-scale projections, or single studies that follow particular approaches with inherent strengths and limitations. We conclude from this analysis that future research should prioritize closing these gaps. Sections of this text read as follows:

“Our analysis reveals that the availability of regional-scale models is inconsistent across the world’s coral reefs. Provinces such as eastern Australia and the Caribbean have received considerable attention and have well-documented projections using various modeling approaches (Figure 5). However, other equally important coral reef provinces, including the eastern Pacific (Costa Rica, Ecuador, and Mexico), western Atlantic (Brazil's northeastern coast), the Indian Ocean, and the Arabian Seas, lack regional-scale models (Figure 5, ⁵⁰). These understudied regions thus heavily rely on less-tailored global assessments for our understanding of future reef impacts in these locations. Many of these provinces also suffer from a limited number of studies and diversity of modeling approaches (Figure 5). For example, projections for the Arabian seas, western Atlantic, and eastern Pacific are exclusively based on SDMs, which involve key assumptions and limitations. Addressing these gaps necessitates targeted efforts to enhance the resolution and accuracy of global-scale projections, while simultaneously expanding the scope and diversity of regional and local-scale projections and monitoring efforts. Such efforts are essential in providing decision-makers with actionable information to manage climate change impacts on coral reefs at global, regional, and local scales^{51,52}.”

Specific comment 20
(Table 1)

Table 1- I don't know if this class of study fits, but they certainly use several of the models described here - the Red List of Ecosystems might fall somewhere near groups 2 and 4 in approach, but also use group 1, and some explicitly use models in the fifth criterion. There are 4 studies now - on Caribbean, meso-American reef, western Indian Ocean and Colombian reefs. A further factor of importance is that the study wants to advance coherence across studies, and use of model results in decision-making, and the RLE incorporates this in its methodology and

We thank Reviewer 3 for highlighting these studies, which were missed in the previous version of our manuscript. This was an oversight. We opted not to include to RLE as a type of major assessment because of challenges in classifying them with the existing categories. Given their critical importance and method, we acknowledge this method in the text, which reads as follows (Pages 10, Lines 228-239):

“There are several other emerging approaches to assess coral reef vulnerability to climate change. In addition to the five major approaches outlined above, two studies used unique methodologies to simulate future impacts on coral and coral reef growth variables in the South China Sea^{53,54}, while one study used historical bleaching and sea surface temperature records to project future bleaching probabilities in the

approach (approaches to combine disparate results) and targeted outputs for policy/management. It is now also one of the few headline indicators in the new CBD biodiversity targets, so is a top priority indicator for application in coming years and including it will help with the policy relevance of the paper.

*Indo-Pacific*⁵⁵. Another regional-scale study focused on larval connectivity and identified conservation areas with lower risks of coral bleaching in the Amani Islands of southern Japan⁵⁶. Articles applying the Red List of Ecosystems represent an especially promising method that seeks to harmonize a diversity of coral reef variables to classify ecosystem collapse risk^{19-21,57}. These studies highlight inconsistencies in data quality and quantity and call for improvements in monitoring efforts for more definitive risk assessments. There are now at least four studies applying this method to coral reefs in the Caribbean^{21,57}, meso-America²⁰, and the western Indian Ocean¹⁹. ”

We further acknowledge their calls for greater coherence among studies, which our review further shows are a key issue (Pages 13, Lines 320-328).

Specific comment 21 The RLE as a type of assessment? Are there other similar approaches based on risk? Incorporates several of the approaches identified here, particularly if crit E can be evaluated as in MAR

Coral reef RLE refs:

Keith, D. A. et al. Scientific foundations for an IUCN Red List of Ecosystems. PLoS ONE 8, e62111 (2013).

Bland, L. M. et al. Using multiple lines of evidence to assess the risk of ecosystem collapse. Proc. R. Soc. B 284, 20170660 (2017).

Obura D, et al (2021) Vulnerability to collapse of coral reef ecosystems in the Western Indian Ocean. Nat

Sustain. <https://doi.org/10.1038/s41893-021-00817-0>

Uribe, E. S., Luna-Acosta, A. & Etter, A. Red List of Ecosystems: risk assessment of coral ecosystems in the Colombian Caribbean.

Ocean Coast. Manag. 199, 105416 (2021).

Please see our response to Reviewer 3, Comment 20 (above). We thank Reviewer 3 for highlighting these key studies, which have been discussed and cited in the text (Pages 10, Lines 228-239 & Pages 13, Lines 320-328).

References:

- 1 Zurell, D. *et al.* A standard protocol for reporting species distribution models. *Ecography* **43**, 1261-1277, doi:<https://doi.org/10.1111/ecog.04960> (2020).
- 2 Araújo, M. B. *et al.* Standards for distribution models in biodiversity assessments. *Science Advances* **5**, eaat4858, doi:doi:10.1126/sciadv.aat4858 (2019).
- 3 Urban, M. C. Projecting biological impacts from climate change like a climate scientist. *WIREs Climate Change* **10**, e585, doi:<https://doi.org/10.1002/wcc.585> (2019).
- 4 Ferrier, S. *et al.* IPBES: The methodological assessment report on scenarios and models of biodiversity and ecosystem services., (Bonn, Germany. , 2016).
- 5 DeFilippo, L. B. *et al.* Assessing the potential for demographic restoration and assisted evolution to build climate resilience in coral reefs. *Ecological applications* **32**, e2650 (2022).
- 6 McManus, L. C. *et al.* Evolution and connectivity influence the persistence and recovery of coral reefs under climate change in the Caribbean, Southwest Pacific, and Coral Triangle. *Global change biology* **27**, 4307-4321 (2021).
- 7 Matz, M. V., Trembl, E. A. & Haller, B. C. Estimating the potential for coral adaptation to global warming across the Indo-West Pacific. *Global Change Biology* (2020).
- 8 Teneva, L. *et al.* Predicting coral bleaching hotspots: the role of regional variability in thermal stress and potential adaptation rates. *Coral Reefs* **31**, 1-12 (2012).
- 9 Kalmus, P., Ekanayaka, A., Kang, E., Baird, M. & Gierach, M. Past the precipice? Projected coral habitability under global heating. *Earth's Future* **10**, e2021EF002608 (2022).
- 10 Donner, S. D., Skirving, W. J., Little, C. M., Oppenheimer, M. & Hoegh-Guldberg, O. Global assessment of coral bleaching and required rates of adaptation under climate change. *Global Change Biology* **11**, 2251-2265 (2005).
- 11 Meissner, K., Lippmann, T. & Sen Gupta, A. Large-scale stress factors affecting coral reefs: open ocean sea surface temperature and surface seawater aragonite saturation over the next 400 years. *Coral Reefs* **31**, 309-319 (2012).
- 12 Langlais, C. *et al.* Coral bleaching pathways under the control of regional temperature variability. *Nature Climate Change* **7**, 839-844 (2017).
- 13 Lachs, L. *et al.* Fine-tuning heat stress algorithms to optimise global predictions of mass coral bleaching. *Remote Sensing* **13**, 2677 (2021).
- 14 Vercammen, A. *et al.* Evaluating the impact of accounting for coral cover in large-scale marine conservation prioritizations. *Diversity and Distributions* **25**, 1564-1574, doi:<https://doi.org/10.1111/ddi.12957> (2019).
- 15 McClanahan, T. Coral responses to climate change exposure. *Environmental Research Letters* **17**, 073001 (2022).
- 16 Geneviev, L. G. C., Jamil, T., Raitos, D. E., Krokos, G. & Hoteit, I. Marine heatwaves reveal coral reef zones susceptible to bleaching in the Red Sea. *Global Change Biology* **25**, 2338-2351, doi:<https://doi.org/10.1111/gcb.14652> (2019).
- 17 Fordyce, A. J., Ainsworth, T. D., Heron, S. F. & Leggat, W. Marine heatwave hotspots in coral reef environments: physical drivers, ecophysiological outcomes, and impact upon structural complexity. *Frontiers in Marine Science* **6**, 498 (2019).
- 18 Zinke, J. *et al.* Gradients of disturbance and environmental conditions shape coral community structure for south-eastern Indian Ocean reefs. *Diversity and Distributions* **24**, 605-620 (2018).
- 19 Obura, D. *et al.* Vulnerability to collapse of coral reef ecosystems in the Western Indian Ocean. *Nature Sustainability* **5**, 104-113, doi:10.1038/s41893-021-00817-0 (2022).
- 20 Bland, L. M. *et al.* Using multiple lines of evidence to assess the risk of ecosystem collapse. *Proceedings of the Royal Society B: Biological Sciences* **284**, 20170660, doi:doi:10.1098/rspb.2017.0660 (2017).
- 21 Uribe, E. S., Luna-Acosta, A. & Etter, A. Red List of Ecosystems: Risk assessment of coral ecosystems in the Colombian Caribbean. *Ocean & Coastal Management* **199**, 105416 (2021).
- 22 Urban, M. C. *et al.* Improving the forecast for biodiversity under climate change. *Science* **353**, aad8466, doi:doi:10.1126/science.aad8466 (2016).
- 23 Nicholson, E. *et al.* Scientific foundations for an ecosystem goal, milestones and indicators for the post-2020 global biodiversity framework. *Nature Ecology & Evolution* **5**, 1338-1349, doi:10.1038/s41559-021-01538-5 (2021).
- 24 Tebbett, S. B., Connolly, S. R. & Bellwood, D. R. Benthic composition changes on coral reefs at global scales. *Nature Ecology & Evolution* **7**, 71-81, doi:10.1038/s41559-022-01937-2 (2023).
- 25 Skirving, W. J. *et al.* The relentless march of mass coral bleaching: a global perspective of changing heat stress. *Coral Reefs* **38**, 547-557, doi:10.1007/s00338-019-01799-4 (2019).
- 26 McClanahan, T. R. *et al.* Temperature patterns and mechanisms influencing coral bleaching during the 2016 El Niño. *Nature Climate Change* **9**, 845-851, doi:10.1038/s41558-019-0576-8 (2019).
- 27 McClanahan, T. R. & Azali, M. K. Environmental Variability and Threshold Model's Predictions for Coral Reefs. *Frontiers in Marine Science* **8**, doi:10.3389/fmars.2021.778121 (2021).
- 28 Santana, E. F. *et al.* Turbidity shapes shallow Southwestern Atlantic benthic reef communities. *Marine Environmental Research* **183**, 105807 (2023).
- 29 Armstrong McKay, D. I. *et al.* Exceeding 1.5oC global warming could trigger multiple climate tipping points. *Science* **377**, eabn7950, doi:doi:10.1126/science.abn7950 (2022).
- 30 IPCC. Climate Change 2022: Impacts, adaptation, and vulnerability: contribution of Working Group II to the sixth assessment report of the Intergovernmental Panel on Climate Change. Report No. 0521015006, 3056 (Cambridge University Press, Cambridge, UK and New York, NY, USA, 2022).
- 31 IPCC. Impacts of 1.5 °C global warming on natural and human systems. (Cambridge University Press, Cambridge, UK and New York, NY, USA, 2018).
- 32 Frieler, K. *et al.* Limiting global warming to 2 °C is unlikely to save most coral reefs. *Nature Climate Change* **3**, 165-170, doi:10.1038/nclimate1674 (2013).
- 33 Schlessner, C.-F. *et al.* Differential climate impacts for policy-relevant limits to global warming: the case of 1.5 C and 2 C. *Earth system dynamics* **7**, 327-351 (2016).

34 Dixon, A. M., Forster, P. M., Heron, S. F., Stoner, A. M. K. & Beger, M. Future loss of local-scale thermal refugia in coral reef ecosystems. *PLOS Climate* **1**, e0000004, doi:10.1371/journal.pclm.0000004 (2022).

35 McWhorter, J. K. *et al.* The importance of 1.5°C warming for the Great Barrier Reef. *Global Change Biology* **28**, 1332-1341, doi:<https://doi.org/10.1111/gcb.15994> (2022).

36 Hoegh-Guldberg, O. Climate change, coral bleaching and the future of the world's coral reefs. *Marine and freshwater research* **50**, 839-866 (1999).

37 Donner, S. D. Coping with commitment: projected thermal stress on coral reefs under different future scenarios. *PLoS One* **4**, e5712 (2009).

38 Hoegh-Guldberg, O., Bindi, M. & Allen, M. Chapter 3: Impacts of 1.5 °C of Global Warming on Natural and Human Systems. *Intergovernmental Panel on Climate Change (IPCC) Special report on the impacts of global warming of 1* (2018).

39 Pörtner, H.-O. *et al.* in *IPCC Special Report on the Ocean and Cryosphere in a Changing Climate (SROCC)* (IPCC, 2019).

40 Lenton, T. M. *et al.* Climate tipping points—too risky to bet against. *Nature* **575**, 592-595 (2019).

41 Hoegh-Guldberg, O. Coral reef ecosystems and anthropogenic climate change. *Regional Environmental Change* **11**, 215-227 (2011).

42 van Hooidonk, R. & Huber, M. Effects of modeled tropical sea surface temperature variability on coral reef bleaching predictions. *Coral Reefs* **31**, 121-131, doi:10.1007/s00338-011-0825-4 (2012).

43 Maynard, J. *et al.* Projections of climate conditions that increase coral disease susceptibility and pathogen abundance and virulence. *Nature Climate Change* **5**, 688-694 (2015).

44 van Hooidonk, R., Maynard, J. A. & Planes, S. Temporary refugia for coral reefs in a warming world. *Nature Climate Change* **3**, 508-511, doi:10.1038/nclimate1829 (2013).

45 van Hooidonk, R., Maynard, J. A., Manzello, D. & Planes, S. Opposite latitudinal gradients in projected ocean acidification and bleaching impacts on coral reefs. *Global Change Biology* **20**, 103-112, doi:<https://doi.org/10.1111/gcb.12394> (2014).

46 Setter, R. O., Franklin, E. C. & Mora, C. Co-occurring anthropogenic stressors reduce the timeframe of environmental viability for the world's coral reefs. *PLOS Biology* **20**, e3001821, doi:10.1371/journal.pbio.3001821 (2022).

47 Meissner, K. J., Lippmann, T. & Sen Gupta, A. Large-scale stress factors affecting coral reefs: open ocean sea surface temperature and surface seawater aragonite saturation over the next 400 years. *Coral Reefs* **31**, 309-319, doi:10.1007/s00338-011-0866-8 (2012).

48 *Allen Coral Atlas*, <<https://allencoralatlas.org/atlas/#4.41/-47.4353/128.2436>> (2023).

49 *NOAA Coral Reef Watch*, <<https://coralreefwatch.noaa.gov/>> (2023).

50 Morais, J., Medeiros, A. P. & Santos, B. A. Research gaps of coral ecology in a changing world. *Marine Environmental Research* **140**, 243-250 (2018).

51 Gurney, G. G., Melbourne-Thomas, J., Geronimo, R. C., Aliño, P. M. & Johnson, C. R. Modelling coral reef futures to inform management: can reducing local-scale stressors conserve reefs under climate change? *PLoS One* **8**, e80137 (2013).

52 Presseley, R. L. *et al.* The mismeasure of conservation. *Trends in Ecology & Evolution* **36**, 808-821 (2021).

53 Yan, H. *et al.* Regional coral growth responses to seawater warming in the South China Sea. *Science of the total environment* **670**, 595-605 (2019).

54 Zuo, X. *et al.* Spatially Modeling the Synergistic Impacts of Global Warming and Sea-Level Rise on Coral Reefs in the South China Sea. *Remote Sensing* **13**, 2626 (2021).

55 Khalil, I., Muslim, A. M., Hossain, M. S. & Atkinson, P. M. Modelling and forecasting the effects of increasing sea surface temperature on coral bleaching in the Indo-Pacific region. *International Journal of Remote Sensing* **44**, 194-216 (2023).

56 Abe, H., Kumagai, N. H. & Yamano, H. Priority coral conservation areas under global warming in the Amami Islands, Southern Japan. *Coral Reefs* **41**, 1637-1650 (2022).

57 Keith, D. A. *et al.* Scientific Foundations for an IUCN Red List of Ecosystems. *PLOS ONE* **8**, e62111, doi:10.1371/journal.pone.0062111 (2013).

REVIEWER COMMENTS

Reviewer #1 (Remarks to the Author):

When I read the response to referees I was pretty happy with the responses. However, on reading the revised paper I feel like the value of the paper has diminished somewhat. To me, the most compelling aspect of the original paper was to highlight the issue of high uncertainty in model projections and then provide a pathway forward to guide people in making use of such uncertainty. This was raised as a significant opportunity for the paper to improve. However, the revised paper has simply removed the entire section on uncertainty (i.e., the opposite of the recommendation).

Instead, the paper has now included a meta-analysis of the limited number of studies and while the results are of interest, I'm not sure that they make a hugely important contribution to the field. As the authors' admit - there is a limited dataset with which to synthesize and many of the results presented are descriptive (e.g., proportion of studies using method X vs Y to project stress). That's fine but it doesn't provide a new synthesis that moves the field onward by arriving at fresh conclusions.

To me the most valuable contribution of the paper is highlighting that IPCC tend to cite the 'excess heat' type papers and these tend to be over-represented in citations despite their obvious limitations.

On reading the revision I had a number of thoughts:

line 13 - I don't think it's correct to suggest that most studies claim that 1.5 C warming is unsustainable for reefs.

line 74 - Yes agreed several studies have found DHW to be lacking but others have found strong predictive relationships. It's likely to be context dependent. Try to be a little more balanced here.

line 315 ish - The real problem is that the efficacy of DHW type measures is bound to decline in future as most reefs have a complex legacy of exposure to disturbance. We definitely

need more studies that explore the phenotypic variety of outcomes - much like Humanes et al PRS

l325 - I don't really agree that there such huge problems in metrics. Almost all studies project coral cover (at least - though not the exceedance studies)

l 340 and 422 I find the narrative around validating projections of coral and the importance of having high power to be odd. You cannot - by definition - validate a prediction made in the future, though you can validate historical projections. Also, I would rarely if ever link power to modelling because you wouldn't use statistics for a model output (you can just add more and more runs to obtain minute confidence intervals and you typically model the entire population).

l 523 and 529. In several places the narrative complains about studies having ignored some factors (e.g., fishing) or only being done in a single locale. Yet those studies may have justified why they ignored some of these factors and why they worked at one location. Sure, that doesn't make them a great source of data for an analysis of global projections of coral reefs but they were never designed for that. So I think the narrative needs to be tightened up and make it less 'whingy'.

Reviewer #2 (Remarks to the Author):

Nature Communication - 402170

I reviewed a previous version of the manuscript. This a very different and improved paper as it is more transparent about the sources of information and they attempt to calculate effect sizes from the data that they can find in papers. There are a few areas that need to be improved. These are that the figures 3 and 4 are still not that clear and need more explanation to understand what is being presented, the abstract needs to be rewritten to be clearer about specifics, and the meta-analysis category is hard to understand how a summary of the literature would lead to predictions in the future.

The authors seem to think or assume the models in use are probabilistic, but they are simple deterministic models and therefore not easily amenable to assigning probabilities. They use deterministic equations, and they make vague categorical statements like “at risk” from these models based on some possibly arbitrary categories. While this is a problem is there really some clear ways to improve what is being done? You can ask people to try but they are constrained by relying on simple deterministic models and do not have data to estimate errors. Data are from satellites without error estimates and models are deterministic equations. None of this is measuring what is of concern, which might be coral cover in the future. Global coral cover and even bleaching compilations are poorly connected to the satellite data. Thus probabilities would be speculative until addressing their relationships.

Below are some more comments that can assist the revision.

Abstract:

I don't find the abstract a good representative of the paper. Compared to the main text, I find the text vague and sometimes repetitive with other critiques, i.e., the need to coordinate and down scale models. Is this really that important as opposed to picking the right variables and the model structures? One of the main problems with the models is nearly all based on the excess heat model, which has low certainty in predicting coral cover. The high certainty of the IPCC is based on the fact the excess heat will increase in the future, but this says very little about how corals will react.

L15 - Does the IPCC really fail to address uncertainty? The 2018 IPCC report states that “multiple lines of evidence indicate that the majority (70–90%) of warm water (tropical) coral reefs that exist today will disappear even if global warming is constrained to 1.5°C (very high confidence)” (Hoegh-Guldberg et al. 2018a). This seems clear about their confidence. Isn't the question whether there is evidence for this high confidence or not?

L18 – is coordination the right word here or is it correspondence? Important to be clear if the problem is methodological or conclusions that differ.

L22 – one wonders what the attributes of these key studies are?

L27 – I believe most projects are not even based on coral reef metrics but rather satellite observation of temperature. As written this sounds vague.

L30 Many papers mentioned geography and scale but few consider the model variables and structure, which can be more important than scale and location.

L32 – what do authors think are the gaps?

Introduction

This section is improved.

The word “elements” is quite vague to me, what are they?

L58 – how are they independent if they all used the same excess heat metric?

L86 – is it a narrow set of projections, variables, or methods. This could be better articulated.

Approaches ..

One problem with SDMs is that they are not very predictive because they often chose variables that are not what control the species. i.e., mean temperature. So, they often do not know what is “suitable”. Corals are driven more by shapes of the temperature or aragonite distributions than by their means, see Environmental variability and threshold model’s predictions for coral reefs. *Frontiers in Marine Science* 8: 1774

I find the meta analysis section a bit hard to follow, consider more explicit writing. It is not clear how a summary of findings would result in making global projections. It is not a meta analysis of projects but of some variables?

L237 – what method? Authors are discussing other methods, what specifically are these, larval dynamics?

L309 – aligned how? This sentence seems important, but I don't understand it.

L315 - What are new data streams? Do you mean variables of something else?

L326 – I am not sure what this group is, is it an IUCN group? “Red List of Ecosystems-based studies”. What is coordinated monitoring? Is it how, when, or the methods that need coordinating? There is the GCRMN but most scientists find these data impenetrable in terms of the methods and analyses. Most groups measure coral cover and sometimes “macroalgae” but not much else, which may be a problem.

Figure 3 shows the effect sizes on what? Coral cover, reef cells at risk, etc? I am not sure what (g) is, can it be explained in the legend? Does this have some time scale, such as by 2050? It is not very clear. Are the error bars based on the scenarios?

L389- What does “at risk” mean in these reports?

P419 – the problem with all these models is that they are deterministic and not probability models. So, no certainty can be calculated unless probabilities are applied to the deterministic variables that are used and yet this is unknown.

L446 – can the authors briefly differentiate “statistical and dynamic downscaling procedures, so it is clearer to the reader

Reviewer #4 (Remarks to the Author):

General:

This paper presents a systematic review of studies projecting coral reef futures under climate change scenarios and highlights gaps and areas for improvement for future research to be applicable to management. The authors undertook significant revisions to address reviewer comments in a previous round of review, which included a systematic review of

studies. Their responses to reviewers were thorough and thoughtful, and the edits are well done. This study is interesting and important, and it points out areas that are really valuable for future research. In their response, the authors have noted areas in need of additional information, as well as inconsistencies among studies, which might be resolved by greater transparency and standardization in uncertainty measures. This was an interesting read, and I think it represents an important contribution to the field with urgent implications for ongoing and future research. The authors should be commended on their revisions and overall product. I recommend it for publication with minor revisions.

Abstract:

I would recommend including more information on the meta-analysis component (8 studies) and highlighting major findings from this.

Introduction:

The explanation of climate tipping points, global and regional elements at risk, and regional tipping elements is a bit confusing in the first paragraph and could use some rephrasing.

Lines 36-37: Expected is used 2x in this sentence

Line 43: Exceedance and exceed are used back to back in this sentence – consider rewording

Lines 41-43: This sentence is confusing and would benefit from a rewrite.

Line 45-47 (1st sentence): Improper grammar – revise

Line 47: Again, reference to regional tipping element needs more explanation in the beginning

Lines 55-56: What do high and medium confidence refer to in this context?

Line 66: What does “among the closest elements” refer to? What are the other elements?
Other ecosystems?

Line 87: Projections used 2x in this sentence. Should second use be “studies” instead?

Approaches for projecting coral reef futures:

Figure 1: Interesting figure! Extra spaces in the legend text.

Line 228: What were the unique methodologies?

Line 233: What is the Red List of Ecosystems? Can you explain here?

The need for more coordinated modeling efforts:

Line 326: Can you explain “Red List of Ecosystem-based studies”

Figure 3: Interesting figure! What does “variable continuous” refer to in the technique?
Should this be continuous variable?

Figure 4: It is not clear to me from the caption why the % coral cover is pulled out and not included as the other projection variables are.

Line 385: Why is decline in quotes? Also, given the prior discussion of how “coral reefs” are measured, it would be good to include here what the definition is in the IPCC (i.e., % coral cover).

Beyond the impact of warming:

Line 528: Caldwell et al. 2016 (<https://www.mdpi.com/2072-4292/8/2/93>) may be relevant here.

Methods:

Line 615: Semicolon at end of sentence should be colon

Line 641: Please double check variance formula as written.

Supplemental:

Really thorough, clear, and well done.

LOCATION	COMMENT	RESPONSE AND ACTION(S)
Reviewer 1		
Overall summary	When I read the response to referees, I was pretty happy with the responses. However, on reading the revised paper I feel like the value of the paper has diminished somewhat. To me, the most compelling aspect of the original paper was to highlight the issue of high uncertainty in model projections and then provide a pathway forward to guide people in making use of such uncertainty. This was raised as a significant opportunity for the paper to improve. However, the revised paper has simply removed the entire section on uncertainty (i.e., the opposite of the recommendation). Instead, the paper has now included a meta-analysis of the limited number of studies and while the results are of interest, I'm not sure that they make a hugely important contribution to the field. As the authors admit - there is a limited dataset with which to synthesize and many of the results presented are descriptive (e.g., proportion of studies using method X vs Y to project stress). That's fine but it doesn't provide a new synthesis that moves the field onward by arriving at fresh conclusions.	We thank Reviewer 1 for their thorough review of our revised manuscript. We appreciate the reviewer's insightful comments and share their concern regarding the perceived diminishing value of the paper. We have now revised the text to re-integrate components of the deleted section, although in a slightly different structure, and now propose a clear pathway to navigate such uncertainty. We would also like to take this opportunity to acknowledge insightful comments of Reviewer 2 and their contribution to our revised approach. Specifically, we now include a section entitled "Addressing uncertainty through more coordinated modelling efforts" which discusses the deterministic nature of most existing model efforts in the field and the need for greater probabilistic estimates of uncertainty. We draw on lessons learned from the field of climate change science to recommend a united ensemble-like approach, where diverse model types can be used to generate probabilistic projections for coral reef futures. While acknowledging the different goals of individual modeling efforts, we advocate for the need for more correspondence among modelers to select common model metrics and emissions scenarios. This, in turn, would highlight major sources of variation and better characterize the uncertainty of coral reef futures under climate change. This section then goes on to address the diversity of metrics used and discusses coral cover as the most used model metric that can be readily applied to real-world observations. We then use our meta-analytical approach, albeit limited in scope, to showcase existing challenges to consolidate available projections and the need for more greater coordination for uniform metrics and emissions scenarios. We highlight the following excerpts from this revised section: (P15, Lines 363-373) "While the models reviewed here vary in their complexity and underlying approaches, most of them rely on deterministic rules to establish cause-and-effect relationships (Supplementary data 1). A primary example includes the application of thermal thresholds to approximate instances of coral bleaching events. Deterministic approaches are intrinsically limited in their ability to account for the uncertainties that arise from the complex interplay between physical and ecological factors inherent to coral reefs. Models that employ probabilistic relationships, as opposed to cause-and-effect assumptions, are arguably better suited to capture the manifold sources of uncertainty in how coral reefs will respond under future climate¹²³⁻¹²⁵. However, this field still faces significant challenges in

establishing robust connections between key coral reef metrics and satellite-derived data^{72,73,111,112}, which restricts the reliability and usefulness of probabilistic models.”

(P16, Lines 375-388) “In the field of climate change science, atmospheric scientists initially faced issues with fragmented data and disparate deterministic models when modeling the Earth's response to increasing greenhouse gas emissions¹²⁶. By the 1980s, coordinated data collection from weather stations and satellites improved the accuracy of atmospheric-ocean GCM models¹²⁷. By 1988, the IPCC formed and used the Coupled Model Inter-comparison Project (CMIP) to coordinate simulations using the same emissions scenarios and model outputs¹²⁸. This ensemble approach combined diverse deterministic model types across research groups to generate reliable probabilistic statements¹²⁶. While acknowledging that climate scientists only model a single system compared to the thousands of interdependent and locally-adapted species comprising coral reefs, adopting an ensemble approach to generate probabilistic projections for coral reef futures is feasible^{126,129,130}. This, in turn, would highlight major sources of variation and better characterize the uncertainty of coral reef futures under climate change. However, applying an IPCC ensemble-like approach would necessitate improved coordination to select common output metrics and emission scenarios.”

(P17, Lines 404-419) “While coral cover represents the most frequently simulated metric directly linked to real-world observations, its effectiveness as a singular measure is constrained¹⁰⁵. The simplicity and accessibility it provides comes with trade-offs, as it fails to encompass other crucial aspects of reef health, including changes in community compositions of corals, algae, and other key taxa essential for ecosystem functioning. Transitions in coral communities in the Western Indian Ocean and the Great Barrier Reef have marked significant ecological shifts in response to climate change^{132,133}, highlighting the requirement for coordinated simulations of numerous common reef variables to better capture future coral reef vulnerability. Present coral reef assessment and monitoring efforts, however, suffer from differences in methods and the resulting datasets¹⁰⁵. Recommendations for unifying frameworks to select common metrics to capture different dimensions of ecosystem integrity and risk of collapse across ecosystems already exist¹³⁴. However, coordination to select key metrics specific to coral reef ecosystems for this purpose is still lagging. This recommendation is further emphasized by studies utilizing the IUCN RLE classification system to evaluate the risk of coral reef ecosystem collapse, which call for enhanced coordination in monitoring coral reefs and improved data quality and quantity¹⁰³⁻¹⁰⁵.”

We now include an additional section entitled “Reporting uncertainty and metrics of model outputs”, which focuses on the need for clarity over the exact definitions of metrics used and the need to reduce usage of subjective terms that do not have formal definitions (P22-23, Lines 508-538). This section incorporates text previously removed from our original manuscript.

Major comment 1 To me the most valuable contribution of the paper is highlighting that IPCC tend to cite the 'excess heat' type papers and these tend to be over-represented in citations despite their obvious limitations.

We agree that this component of the manuscript is meaningful and now include this aspect in the Abstract as a major finding. (P1-2, Lines 13-31)

“Climate change impact syntheses, such as those by the Intergovernmental Panel on Climate Change (IPCC), consistently assert that limiting global warming to 1.5°C is unlikely to safeguard most of the world’s coral reefs. This prognosis primarily stems from 'excess heat' threshold models, which assume that widespread coral bleaching predictably occurs when temperatures accumulate beyond a specific threshold. Our systematic review of research projecting coral reef futures to climate change (n=79) revealed that 'excess heat' models constituted only one third (32%) of all studies but attracted a high proportion (68%) of citations in the field. We observed that, irrespective of the approach, most methods employed deterministic cause-and-effect rules rather than probabilistic relationships, impeding the field's ability to estimate uncertainties of coral reef futures. In attempting to assess the consistency of projected impacts, we aimed to identify common coral reef metrics under uniform emissions scenarios. However, disparate choices in metrics and emissions scenarios hindered a cohesive synthesis and limited the exploratory analysis to a small fraction of available studies. We found significant discrepancies in expected impacts to coral reefs, suggesting that 'excess heat' models may project more extreme impacts than other methods. Drawing on lessons from the field of climate change science, we propose that an IPCC ensemble-like approach to generating probabilistic projections for coral reef futures is feasible. Successful implementation will require improved coordination among modeling efforts to select common output metrics and emission scenarios, addressing existing geographical biases, among other gaps in current modeling efforts.”

Specific comment 1, I don't think it's correct to suggest that most studies claim that
Line 13 1.5 C warming is unsustainable for reefs.

We agree with Reviewer 1 that most projection many studies do not assert the unsustainability of coral reefs under 1.5°C warming. We have revised this sentence to make it clear that we are referring to climate change impact syntheses, including reports such as the IPCC’s AR6 Report, the IPCC’s Special Report on

1.5°C of Global Warming, and other syntheses of published data, such as those exploring climate change tipping points, exemplified by Armstrong et al. (2022)¹.

The revised sentence reads as follows (P1, Lines 13-15): “Climate change impact syntheses, such as those by the Intergovernmental Panel on Climate Change (IPCC), consistently assert that limiting global warming to 1.5°C is unlikely to safeguard most of the world’s coral reefs.”

Specific comment 2,
Line 74

Yes, agreed several studies have found DHW to be lacking but others have found strong predictive relationships. It's likely to be context dependent. Try to be a little more balanced here.

We appreciate the suggestion from Reviewer 1 and acknowledge that the disparities among investigations regarding the relationship between these metrics and bleaching instances suggest that the strength of these relationships is likely dependent on the specific context. We have revised the text to acknowledge that other investigations have found strong correlations, which now reads: (P4, Lines 79-85): “Put simply, these methods operate under the notion that widespread bleaching predictably occurs when temperatures accumulate beyond a specific threshold. While some investigations show that ‘excess heat’ threshold metrics have strong predictive relationships with bleaching events^{12,28,29}, several others have found these metrics to have limited predictive power when applied to historical bleaching records^{24,30,31}. Despite these inconsistencies, which suggest that the effectiveness of ‘excess heat’ threshold metrics may depend on the specific context, this approach continues to prevail in the field²⁴.”

Specific comment 3,
Line 315

The real problem is that the efficacy of DHW type measures is bound to decline in future as most reefs have a complex legacy of exposure to disturbance. We definitely need more studies that explore the phenotypic variety of outcomes - much like Humanes et al PRS

We thank Reviewer 1 for highlighting this key point. We added text to address the need for more studies addressing how thermal tolerances of corals and other coral reef taxa are likely to shift as coral reefs are exposed to increasing levels of stress. This section of text reads as follows (P15, Lines 348-360): “Another major consideration is the future efficacy of threshold-based metrics in reliably approximating instances of coral bleaching or changes in other coral reef metrics. This is because most coral reefs have already experienced a complex legacy of exposure to disturbance and it is presently unclear by how much and to what extent organisms have adapted or will adapt in future^{120,121}. For example, a recent study examined intrapopulation variability of heat tolerance in corals from the Western Pacific Ocean¹²². The study demonstrated that the most heat-tolerant corals in their study required double the heat stress to induce bleaching compared to their least-tolerant corals. When these differences in heat tolerance were translated into contrasting DHW thresholds and applied to an ambitious emissions scenario (SSP2 -4.5), the study reported that the most heat-tolerant corals could potentially experience annual bleaching events up to 17 years later than their less-tolerant counterparts¹²². Thus, greater confidence in coral reef

projections will likely also depend on greater knowledge of how thermal tolerances of corals and other coral reef taxa are likely to shift as coral reefs are exposed to increasing levels of stress over time.”

Specific comment 4,
Line 325 I don't really agree that there such huge problems in metrics.
Almost all studies project coral cover (at least - though not the exceedance studies)

Among the studies in our database, fewer than one-third projected changes in future coral cover. We revised this section to eliminate text addressing minor discrepancies in metrics representing coral cover. Instead, we now clarify that coral cover represents the most frequently simulated metric directly linked to real-world observations and include an additional section discussing the importance of uniform metrics.

The revised section now reads (P16-17, Lines 390-419): “Despite the growing body of studies forecasting coral reef futures, there is presently no broad consensus on the optimal variables for projecting coral reef vulnerability¹⁰³⁻¹⁰⁵. This is reflected in the diversity of variables used and the large proportion of studies delivering projections with metrics that prove challenging to translate to real-world observations (Supplementary Table 1). Establishing a connection between modeled metrics and real-world observations is not only crucial for enhancing the practicality and usefulness of modeled projections but also enables future assessments of the models' ability to simulate past conditions. More than half of the studies employing 'excess heat' thresholds presented their projections in terms of fractions of reef cells at risk (52%), while SDMs typically provided estimates in terms of fractions of reef cells with suitable habitats or relative changes in habitat suitability (69%) (Supplementary Table 1). Although coral cover serves as a widely used and accessible indicator for this purpose^{105,131}, projections of coral cover were delivered in less than a third of all published studies in our database (29%).

While coral cover represents the most frequently simulated metric directly linked to real-world observations, its effectiveness as a singular measure is constrained¹⁰⁵. The simplicity and accessibility it provides comes with trade-offs, as it fails to encompass other crucial aspects of reef health, including changes in community compositions of corals, algae, and other key taxa essential for ecosystem functioning. Transitions in coral communities in the Western Indian Ocean and the Great Barrier Reef have marked significant ecological shifts in response to climate change^{132,133}, highlighting the requirement for coordinated simulations of numerous common reef variables to better capture future coral reef vulnerability. Present coral reef assessment and monitoring efforts, however, suffer from differences in methods and the resulting datasets¹⁰⁵. Recommendations for unifying frameworks to select common metrics to capture different dimensions of ecosystem integrity and risk of collapse across ecosystems

already exist¹³⁴. However, coordination to select key metrics specific to coral reef ecosystems for this purpose is still lagging. This recommendation is further emphasized by studies utilizing the IUCN RLE classification system to evaluate the risk of coral reef ecosystem collapse, which call for enhanced coordination in monitoring coral reefs and improved data quality and quantity¹⁰³⁻¹⁰⁵.”

Specific comment 5,
Lines 340 and 422

I find the narrative around validating projections of coral and the importance of having high power to be odd. You cannot - by definition - validate a prediction made in the future, though you can validate historical projections. Also, I would rarely if ever link power to modelling because you wouldn't use statistics for a model output (you can just add more and more runs to obtain minute confidence intervals and you typically model the entire population).

We agree and removed text referring to statistical power. The revised text on reporting standards on P22, Lines 508-518 now reads:

“One of the fundamental, yet basic steps towards improving future syntheses of modelled projections is the adherence to essential reporting standards. While all studies in our database provided ample data to facilitate interpretation of the study outcomes, most (89% of studies) failed to report basic metrics for model outputs or sufficient extractable data for measures of variation to be converted into uniform metrics. In many cases, challenges arose from the display of results in figures and geographical maps that were not accompanied by adequate supplemental information reporting extractable values. Recognizing the necessity for reporting standards, other fields focused on projecting climate change impacts to biological systems have implemented agreed-upon standards^{67,68,126}. For instance, the International Union for Conservation of Nature (IUCN) established preliminary reporting standards for species threat assessments based on SDMs¹⁴⁰, which have been further refined in subsequent publications^{67,68}. ”

The reviewer is correct in pointing out that you cannot directly validate predictions made for the future. We revised this section of the text to clarify that establishing a connection between model metrics and real-world observations permits future assessments of the models' ability to simulate past projections. This section of the text reads as follows (P16, Lines 390-402):

“Despite the growing body of studies forecasting coral reef futures, there is presently no broad consensus on the optimal variables for projecting coral reef vulnerability¹⁰³⁻¹⁰⁵. This is reflected in the diversity of variables used and the large proportion of studies delivering projections with metrics that prove challenging to translate to real-world observations (Supplementary Table 1). Establishing a connection between modeled metrics and real-world observations is not only crucial for enhancing the practicality and usefulness of modeled projections but also enables future assessments of the models' ability to simulate past conditions. More than half of the studies employing 'excess heat' thresholds presented their projections in terms of fractions of reef cells at risk (52%), while SDMs typically provided estimates in terms of fractions of reef cells with suitable habitats or relative changes in habitat suitability (69%)

(Supplementary Table 1). Although coral cover serves as a widely used and accessible indicator for this purpose^{105,131}, projections of coral cover were delivered in less than a third of all published studies in our database (29%).

Specific comment 6,
Lines 523 -529

In several places the narrative complains about studies having ignored some factors (e.g., fishing) or only being done in a single locale. Yet those studies may have justified why they ignored some of these factors and why they worked at one location. Sure, that doesn't make them a great source of data for an analysis of global projections of coral reefs but they were never designed for that. So I think the narrative needs to be tightened up and make it less 'whingy'.

As suggested, we have revised this section of the text to provide a more balanced perspective. This section now reads:

(P26-27, Lines 619-662) “Coral reef research has allocated significant effort to forecasting and understanding the combined impacts of climate change and ocean acidification on coral reefs (Supplementary Tables 1 & 4). On the other hand, our analysis revealed that 16 studies in the database considered pollution to some extent, and four studies considered fishing pressure in their projections (Supplementary Tables 1 & 4). Although ocean acidification will undoubtedly have discernable effects on coral reefs¹⁶⁴, there are no practical solutions available to mitigate ocean acidification, apart from the urgent reduction of greenhouse gas emissions^{165,166}. In contrast, elevated nutrients and fishing pressure are now well recognized to increase the susceptibility of coral reefs to heatwaves^{161,167,168}, and practical measures to address these pressures are accessible^{169,170}. Local-scale management actions to minimize pollution and regulate fishing have already demonstrated success in reducing cumulative impacts to coral reefs¹⁶⁹⁻¹⁷², particularly in Pacific nations where actions to manage reefs have been implemented for centuries¹⁷³.

A similar pattern exists for evaluating how pest species and disease will interact with climate change to shape the future of coral reefs. In our analysis, we found only two investigations that delved into the role of coral disease outbreaks in influencing coral reef futures under climate change (Supplementary Tables 1 & 4)^{35,43}, with one of these studies being limited to a simulation of a single reef. The global study highlighted that future warming is likely to heighten coral susceptibility to disease and identified specific locations where targeted management could be implemented⁴³. Although excluded from our analysis due to the absence of future climate change projections, numerous predictive models serving as early warning systems for coral diseases exist¹⁷⁴⁻¹⁷⁶. These early warning system models have identified crucial drivers of disease outbreaks in various regions, which could prove useful for refining existing models projecting coral disease outbreaks under future climate change scenarios. Similarly, we identified only one study that simulated the impact of a pest species in climate change scenarios for coral reefs (Supplementary Tables 1 & 4). This study assessed the potential effectiveness of management

strategies in addressing outbreaks of CoTS and reducing cumulative impacts on the Great Barrier Reef⁵⁵. The urgency to address this area of uncertainty is underscored by the ongoing coral disease outbreak in the Gulf of Mexico, which poses a severe threat to coral reefs in the region^{177,178}. Disease outbreaks are becoming increasingly concerning, affecting not only coral reefs but also other marine life^{179,180}, highlighting the need for urgent attention and action.

With the growing recognition of the need for intervention measures, particularly in line with the Kunming-Montreal biodiversity framework's objective of restoring 30% of degraded habitats by 2030, projection models are likely to play a crucial role in guiding these endeavors. Our analysis points towards a possible need to shift the focus of future modelling experiments to better guide actions to manage and restore coral reefs. This does not imply that modeling studies should neglect stressors like ocean acidification, which are expected to have long-term impacts and limited practical solutions. Instead, modelers could consider prioritizing the inclusion of management and intervention scenarios, including coral reef restoration, that integrate the modeled effects of global and regional pressures. Just three of the 79 studies reviewed here included potential intervention scenarios. Two of these studies explored unconventional geoengineering solutions^{2,3}, while one simulated the potential benefits of demographic restoration and assisted evolution in enhancing reef resilience⁴."

Reviewer 2

Overall summary

I reviewed a previous version of the manuscript. This is a very different and improved paper as it is more transparent about the sources of information and they attempt to calculate effect sizes from the data that they can find in papers. There are a few areas that need to be improved. These are that the figures 3 and 4 are still not that clear and need more explanation to understand what is being presented, the abstract needs to be rewritten to be clearer about specifics, and the meta-analysis category is hard to understand how a summary of the literature would lead to predictions in the future.

We thank Reviewer 2 for their constructive feedback, which has greatly improved our manuscript. We respond to each of the points raised by the review in a point-by-point manner below.

Briefly, we re-wrote the Abstract to be clearer about the specifics of our findings and conclusions, and edited the text to better explain the meta-analysis component and provided more detail in the figure captions.

We are particularly grateful Reviewer 2's input regarding the deterministic nature of most studies in the dataset and now include text in the manuscript to address and discuss this. We now draw on lessons learned from the field of climate change science to recommend a united ensemble-like approach, where diverse model types (including disparate deterministic studies) can be used to generate probabilistic projections for coral reef futures. We discuss, however, that this approach requires the selection of

The authors seem to think or assume the models in use are probabilistic, but they are simple deterministic models and therefore not easily amenable to assigning probabilities. They use deterministic equations, and they make vague categorical statements like “at risk” from these models based on some possibly arbitrary categories. While this is a problem is there really some clear ways to improve what is being done? You can ask people to try but they are constrained by relying on simple deterministic models and do not have data to estimate errors. Data are from satellites without error estimates and models are deterministic equations. None of this is measuring what is of concern, which might be coral cover in the future. Global coral cover and even bleaching compilations are poorly connected to the satellite data. Thus probabilities would be speculative until addressing their relationships.

common model output metrics and emissions scenarios. This, in turn, would highlight major sources of variation and better characterize the uncertainty of coral reef futures under climate change. This section then goes on to address the diversity of metrics used and highlights coral cover as the most used model metric that can be readily applied to real-world observations. We then use our meta-analytical approach, albeit limited in scope, to showcase existing challenges to consolidate available projections and the need for more greater coordination for uniform metrics and emissions scenarios.

We include the following excerpts from the new section entitled “Addressing uncertainty through more coordinated efforts”, which address the key points raised:

(P15-16, Lines 363- 388) “While the models reviewed here vary in their complexity and underlying approaches, most of them rely on deterministic rules to establish cause-and-effect relationships (Supplementary data 1). A primary example includes the application of thermal thresholds to approximate instances of coral bleaching events. Deterministic approaches are intrinsically limited in their ability to account for the uncertainties that arise from the complex interplay between physical and ecological factors inherent to coral reefs. Models that employ probabilistic relationships, as opposed to cause-and-effect assumptions, are arguably better suited to capture the manifold sources of uncertainty in how coral reefs will respond under future climate¹²³⁻¹²⁵. However, this field still faces significant challenges in establishing robust connections between key coral reef metrics and satellite-derived data^{72,73,111,112}, which restricts the reliability and usefulness of probabilistic models.

In the field of climate change science, atmospheric scientists initially faced issues with fragmented data and disparate deterministic models when modeling the Earth's response to increasing greenhouse gas emissions¹²⁶. By the 1980s, coordinated data collection from weather stations and satellites improved the accuracy of atmospheric-ocean GCM models¹²⁷. By 1988, the IPCC formed and used the Coupled Model Inter-comparison Project (CMIP) to coordinate simulations using the same emissions scenarios and model outputs¹²⁸. This ensemble approach combined diverse deterministic model types across research groups to generate reliable probabilistic statements¹²⁶. While acknowledging that climate scientists only model a single system compared to the thousands of interdependent and locally-adapted species comprising coral reefs, adopting an ensemble approach to generate probabilistic projections for coral reef futures is feasible^{126,129,130}. This, in turn, would highlight major sources of variation and better characterize the uncertainty of coral reef futures under climate change. However, applying an IPCC ensemble-like

approach would necessitate improved coordination to select common output metrics and emission scenarios.”

Major comment 1,
Abstract

I don't find the abstract a good representative of the paper. Compared to the main text, I find the text vague and sometimes repetitive with other critiques, i.e., the need to coordinate and down scale models. Is this really that important as opposed to picking the right variables and the model structures? One of the main problems with the models is nearly all based on the excess heat model, which has low certainty in predicting coral cover. The high certainty of the IPCC is based on the fact the excess heat will increase in the future, but this says very little about how corals will react.

As suggested, we rewrote the Abstract to focus on the key points suggested by Reviewer 2. (P1-2, Lines 13-31)
“Climate change impact syntheses, such as those by the Intergovernmental Panel on Climate Change (IPCC), consistently assert that limiting global warming to 1.5°C is unlikely to safeguard most of the world's coral reefs. This prognosis primarily stems from 'excess heat' threshold models, which assume that widespread coral bleaching predictably occurs when temperatures accumulate beyond a specific threshold. Our systematic review of research projecting coral reef futures to climate change (n=79) revealed that 'excess heat' models constituted only one third (32%) of all studies but attracted a high proportion (68%) of citations in the field. We observed that, irrespective of the approach, most methods employed deterministic cause-and-effect rules rather than probabilistic relationships, impeding the field's ability to estimate uncertainties of coral reef futures. In attempting to assess the consistency of projected impacts, we aimed to identify common coral reef metrics under uniform emissions scenarios. However, disparate choices in metrics and emissions scenarios hindered a cohesive synthesis and limited the exploratory analysis to a small fraction of available studies. We found significant discrepancies in expected impacts to coral reefs, suggesting that 'excess heat' models may project more extreme impacts than other methods. Drawing on lessons from the field of climate change science, we propose that an IPCC ensemble-like approach to generating probabilistic projections for coral reef futures is feasible. Successful implementation will require improved coordination among modeling efforts to select common output metrics and emission scenarios, addressing existing geographical biases, among other gaps in current modeling efforts.”

Specific comment 1,
Line 15

L15 -Does the IPCC really fail to address uncertainty? The 2018 IPCC report states that “multiple lines of evidence indicate that the majority (70–90%) of warm water (tropical) coral reefs that exist today will disappear even if global warming is constrained to 1.5°C (very high confidence)” (Hoegh-Guldberg et al. 2018a). This seems clear about their confidence. Isn't the question whether there is evidence for this high confidence or not?

We agree with Reviewer 2 that this statement, as it was written, was misleading. We rewrote the abstract and removed the assertion that the IPCC fails to address uncertainty (P1-2, Lines 13-31).

Specific comment 2, Line 18	is coordination the right word here or is it correspondence? Important to be clear if the problem is methodological or conclusions that differ.	We have refined the Abstract to explicitly convey that enhanced coordination among modeling efforts is essential for selecting common coral reef metrics and emissions scenarios. We have chosen the term "coordination" based on the Oxford dictionary's definition, which defines it as "the act of making parts of something, groups of people, etc., work together in an efficient and organized way." We believe that "coordination" accurately captures the need for improved collaboration and organization among diverse modeling approaches to ensure a cohesive and consistent evaluation of coral reef futures under climate change.
Specific comment 3, Line 22	One wonders what the attributes of these key studies are?	Due to the constraints of the Abstract word count, we aimed to succinctly convey that excess heat threshold models tended to project more extreme impacts compared to other model types. However, we acknowledge the importance of providing more detailed information on the attributes of these key studies. In the main text, specifically on Pages 21-22, lines 464-505, we delve into a more comprehensive discussion of the underlying assumptions of the studies.
Specific comment 4, Line 27	I believe most projects are not even based on coral reef metrics but rather satellite observation of temperature. As written, this sounds vague.	As mentioned above, we rewrote the Abstract and removed this component.
Specific comment 5, Line 30	Many papers mentioned geography and scale but few consider the model variables and structure, which can be more important than scale and location.	We agree. The Abstract now focuses on the major findings of our systematic review and synthesis. Hence, the focus on geography and scale has largely been removed from the Abstract.
Specific comment 6, Lines 32	What do authors think are the gaps?	This statement was removed from the Abstract and details of the identified gaps are addressed explicitly in the manuscript text.
Specific comment 7, Introduction	The word "elements" is quite vague to me, what are they?	We have revised the first paragraph of the text to provide a clearer understanding of the term 'tipping elements' and clarify the meaning of global and regional elements. The first paragraph now reads (P3, Lines 33-47): "Anthropogenic climate change is anticipated to propel large components of the Earth's system beyond critical climate tipping points (CTPs), initiating self-perpetuating change and impacts across biophysical systems ¹ . These components, known as 'tipping elements,' are distinguished by their significance in Earth's system functioning, their substantial contributions to human well-being, or their unique value within the Earth's system ¹ . A notable example is the projected dieback of the Amazon rainforest that could release gigatons of carbon into the atmosphere and accelerate global warming ¹⁻³ . Although the concept of CTPs has been subject to debate ^{4,5} , a recent synthesis delivered a shortlist of nine

		global and seven regional elements at risk¹. Global tipping elements, such as the Amazon rainforest and West Antarctic Ice Sheet, refer to components spanning subcontinental scales that could alter the operation of Earth's system¹. Regional tipping elements represent biospheres expected to exhibit self-sustaining feedback at confined scales that have the potential to occur synchronously across subcontinental scales, including for example, the simultaneous melting of alpine glaciers^{1,6}. Among the shortlisted elements at risk are warm-water coral reefs, which represent a regional tipping element vulnerable to exceedance if global warming surpasses 1.5°C above preindustrial levels^{1,4,7}."
Specific comment 8, Line 58	L58 – how are they independent if they all used the same excess heat metric?	We agree and replaced the word 'independent' with 'several' to avoid implying that the modeling efforts are entirely distinct from each other (see P4, Line 64). Indeed, these studies largely follow the same approach.
Specific comment 9, Line 86	Is it a narrow set of projections, variables, or methods. This could be better articulated.	We agree with Reviewer 2 that this sentence was unclear. This statement has since been removed from the aims paragraph, which now reads (P5, Lines 94-102): "Despite the growing body of literature projecting coral reef futures and their prominent role in assessments of climate change impacts, a comprehensive evaluation of available projections is lacking. Here, we address this requirement by conducting a systematic review of published projections of coral reef futures under climate change in isolation or in combination with other pressures. We first review existing approaches to project coral reef futures and their use in the scientific literature, and then identify key gaps in knowledge that currently contribute to uncertainties. We also unpack how lessons from the field of climate change science can provide a pathway for improving coordination of modelling efforts towards greater certainty in projections of coral reef futures."
Specific comment 10 'Approaches' section	One problem with SDMs is that they are not very predictive because they often chose variables that are not what control the species. i.e., mean temperature. So, they often do not know what is "suitable". Corals are driven more by shapes of the temperature or aragonite distributions than by their means, see Environmental variability and threshold model's predictions for coral reefs. Frontiers in Marine Science 8: 1774	We agree and edited the text to acknowledge this common limitation. The revised paragraph reads as follows (P9, Lines 175-188): "While climate-related data (e.g., mean SST) are used in SDMs applied to coral reefs, other physical parameters such as light availability, current speed, and water depth can also be included ^{64,66,69} . The most common physical parameters employed in the SDMs within our database were SST and aragonite saturation ^{65,66,70,71} , which were commonly represented by their means (Supplementary data 1). By primarily relying on means of physical parameters, such models overlook the well-recognized importance of environmental variability in influencing coral reef responses to climate change, which includes

capturing the shapes and distributions of these parameters^{72,73}. Another major limitation arises from the assumption that the physical environment alone largely governs the natural distribution of warm-water reefs. This key assumption overlooks key biological and ecological processes, such as the influential role of larval dispersal and retention in shaping reef distributions^{74,75}, as well as the role of top-down controls in the food web⁷⁶⁻⁷⁹. Significantly, the biogeographic approach of inferring past ecology largely disregards the potential for species' niches to evolve through adaptive processes. This limitation can lead to an underestimation of future distributions⁸⁰.”

Specific comment 11, 'Approaches' section
I find the meta analysis section a bit hard to follow, consider more explicit writing. It is not clear how a summary of findings would result in making global projections. It is not a meta analysis of projects but of some variables?

We revised the meta-analysis section to be more explicit. The revised section now clarifies how these meta-analyses consolidate data from published experiments to deliver projections (P10, Lines 217-232): “An emerging approach to forecast coral reef futures consolidates data from published experimental manipulations. Representing a minority of the reviewed studies (5%) (Figure 1), the identified meta-analyses^{20,90-92} shared a common goal of projecting the dual impacts of ocean warming and acidification on biological processes within reefs. For this, they compile data from experiments that measure how corals and other coral reef species respond to conditions that simulate future warming and acidification scenarios. These data are then utilized to parameterize models for estimating future coral responses under various representative concentration pathway (RCP) scenarios. The specific purposes of these meta-analyses vary, ranging from estimating changes in numerous biological responses of corals²⁰ to those that exclusively focused on alterations in coral calcification processes⁹² or reef-wide calcium carbonate production^{90,91}. Although data from coral reef monitoring, rather than controlled experiments, arguably offer more realistic insights into how reefs will respond to further warming, our understanding of how reef organisms will react to ocean acidification is primarily based on manipulative experiments^{93,94}. Thus, one significant advantage of these approaches is their capacity to consolidate the wealth of data derived from experiments to estimate how future warming and acidification will interact and impact the biological responses of reef organisms.”

Specific comment 12, Line 237
What method? Authors are discussing other methods, what specifically are these, larval dynamics?

As suggested, we have revised this section to provide more specific descriptions of the 'other' approaches mentioned in the manuscript. This section of the text reads as follows (P11, Lines 245-253): “There are several other emerging approaches to assess coral reef vulnerability to climate change. In addition to the five major approaches outlined above, one study adopted a spatial modeling approach to project the combined effects of warming and sea-level rise on the future coral reef growth rates in the South China

Sea (SCS)⁹⁸. Another integrated linear extension rates of corals from three different islands in the same region with future SSTs to forecast coral growth rates⁹⁹. One study used historical bleaching and sea surface temperature records to project future bleaching probabilities in the Indo-Pacific¹⁰⁰, while another regional-scale study focused on larval connectivity and identified conservation areas with lower risks of coral bleaching in the Amani Islands of southern Japan¹⁰¹.”

Specific comment 13,
Line 309 Aligned how? This sentence seems important, but I don't understand it.

We agree with Reviewer 2 that this was unclear and revised the paragraph to provide clarity. It now explains that our observation aligns with the findings of a recent review of studies aiming to explain historical bleaching patterns, indicating a consistency in the limited use of variable selection procedures in threshold-based studies. The revised sentences read (P14, Lines 336-339): “However, we observed few threshold-based studies that applied variable selection procedures. This observation aligns with the findings of a recent review, which similarly found that studies utilizing threshold-based approaches to explain historical bleaching patterns seldom employed variable selection protocols²⁴.”

Specific comment 14,
Line 315 What are new data streams? Do you mean variables of something else?

As suggested, we now clarify the meaning of ‘new data streams’ by providing a timely example of higher resolution satellites that are coming online to provide higher resolution SST data. The section of the text now reads (P15, Lines 341-346): “One of the main challenges lies in the limited temporal resolution of SST data, which ultimately impact measures of SST variability and rates of SST rise²⁴. Greater confidence in coral reef projections will likely depend on new data streams, including higher-resolution satellites (e.g., Himawari¹¹⁹) coming online to provide improved SST data resolution, alongside an increase in projections derived from methods that incorporate variable selection procedures.”

Specific comment 15,
Line 326 I am not sure what this group is, is it an IUCN group? “Red List of Ecosystems-based studies”. What is coordinated monitoring? Is it how, when, or the methods that need coordinating? There is the GCRMN but most scientists find these data impenetrable in terms of the methods and analyses. Most groups measure coral cover and sometimes “macroalgae” but not much else, which may be a problem.

We thank Reviewer 2 (and Reviewer) 4 for highlighting the lack of clarity regarding the IUCN RLE methodology and studies discussed in the manuscript. To address this, we first provide a detailed explanation of the IUCN RLE approach and provide an example of how this standardized method has been applied to project future coral reef vulnerabilities in the ‘Approaches’ section of the manuscript (P11, Lines 255-2682). In this section, we further clarify the meaning of “more coordinated monitoring efforts” to explain that the issues lie in inconsistency of coral monitoring methods, datasets, and the interpretation of results.

This revised text reads as follows (P11, Lines 255-268): “Articles applying a standardized method for assessing the risk of ecosystem collapse, the International Union for Conservation of Nature (IUCN) Red List of Ecosystems (RLE), represent an especially promising method that seeks to harmonize a diversity of coral reef variables to classify ecosystem collapse risk¹⁰²⁻¹⁰⁵. The RLE offers a standardized classification system that utilizes thresholds for key variables to integrate diverse data, making it valuable for policy applications¹⁰⁶⁻¹⁰⁸. For instance, one investigation applying this approach to coral reefs in the western Indian Ocean used metrics measuring historical changes in ecosystem extent and ecosystem functioning to reveal varying levels of regional vulnerability to future ocean warming¹⁰⁵. These studies shed light on inconsistencies in coral reef monitoring methods, resulting datasets, and the interpretation of results¹⁰²⁻¹⁰⁵. In turn, they emphasize the need for improvements in the consistency of monitoring efforts and advocate for the development of a unifying framework to enable more conclusive risk assessments. There are at least four studies applying this method to coral reefs in the Caribbean^{102,103}, meso-America¹⁰⁴, and the western Indian Ocean¹⁰⁵.
”

We agree with Reviewer 2 that most groups typically measure coral cover and, in some cases, macroalgae. While coral cover provides a fundamental measure of the presence and status of hard corals, we agree that these metrics often lack crucial information about community composition and other vital indicators of coral reef 'health.' We have revised the text to maintain the point that coral cover remains the most common and straightforward metric used in model outputs, which can be verified against real-world observations. Unlike outputs such as 'percent reef cells at risk.' However, we now include additional text to acknowledge the limitations of coral cover as the only common model output that can be verified against real-world monitoring efforts.

This section of text now reads (P17, Lines 404-419):

“While coral cover represents the most frequently simulated metric directly linked to real-world observations, its effectiveness as a singular measure is constrained¹⁰⁵. The simplicity and accessibility it provides comes with trade-offs, as it fails to encompass other crucial aspects of reef health, including changes in community compositions of corals, algae, and other key taxa essential for ecosystem functioning. Transitions in coral communities in the Western Indian Ocean and the Great Barrier Reef have marked significant ecological shifts in response to climate change^{132,133}, highlighting the requirement

for coordinated simulations of numerous common reef variables to better capture future coral reef vulnerability. Present coral reef assessment and monitoring efforts, however, suffer from differences in methods and the resulting datasets¹⁰⁵. Recommendations for unifying frameworks to select common metrics to capture different dimensions of ecosystem integrity and risk of collapse across ecosystems already exist¹³⁴. However, coordination to select key metrics specific to coral reef ecosystems for this purpose is still lagging. This recommendation is further emphasized by studies utilizing the IUCN RLE classification system to evaluate the risk of coral reef ecosystem collapse, which call for enhanced coordination in monitoring coral reefs and improved data quality and quantity¹⁰³⁻¹⁰⁵.”

Specific comment 16, Figure 3 shows the effect sizes on what? Coral cover, reef cells at risk, etc? I am not sure what (g) is, can it be explained in the legend? Does this have some time scale, such as by 2050? It is not very clear. Are the error bars based on the scenarios?

We thank Reviewer 2 for highlighting that the text was not clear in addressing the effect sizes used. We edited the caption of Figure 3 to explain how the effect sizes should be interpreted and clarify that the projections represent only the end-of-century (2090-2100) (P18, Lines 421-430), which is also detailed in the Methods section (P29, Lines 679-748). We further edited the main text to explain how effect sizes are used to resolve differences in output units and permit an assessment of magnitude (P18-19, Lines 432-449).

Specific comment 17, What does “at risk” mean in these reports?
Line 389

We edited the text to clarify (P21, Lines 473-481), which reads as follows:
“According to the Summary for Policy Makers in the IPCC’s Sixth Assessment Report, coral reefs are expected to ‘decline’ by 70-90% at 1.5°C global warming (relative to pre-industrial levels), and the estimates exceed 99% at 2°C¹³⁹. While there is no formal definition of ‘decline’ in the context of projection statements within the IPCC reports, the assessments integrated projections of the percentage of reef cells at risk, drawing from the findings of Schleussner et al. (2016) and Frieler et al. (2013) as foundational evidence. Importantly, both studies utilized the same DHM thresholds and applied the same frequency of heating events expected to impede reef recovery in estimating the fraction of reef cells at risk of long-term degradation.”

Specific comment 18, The problem with all these models is that they are deterministic and not probability models. So, no certainty can be calculated unless probabilities are applied to the deterministic variables that are used and yet this is unknown.
Line 419

As detailed above, in response to Reviewer 2’s overall summary, we now address the deterministic nature of most studies in the dataset. We draw on lessons learned from the field of climate change science to recommend a united ensemble-like approach, where diverse model types (including disparate deterministic studies) can be used to generate probabilistic projections for coral reef futures. We discuss, however, that this approach requires the selection of common model output metrics and emissions scenarios (see P15-16, Lines 363- 388).

Specific comment 18, Line 446	L446 – can the authors briefly differentiate “statistical and dynamic downscaling procedures, so it is clearer to the reader	As suggested, we now include a brief description of statistical and dynamical downscaling procedures in the text. This section of the text reads as follows (P24, Lines 551-555): “There are two main approaches to improve the spatial resolution of global and regional models: statistical and dynamic downscaling procedures ¹⁴⁵ . Statistical downscaling estimates local-scale climate variables from larger-scale climate models using statistical methods, whereas dynamic downscaling uses regional numerical models to simulate local conditions at a higher spatial resolution based on global climate model outputs ¹⁴⁶ .”
Reviewer 4		
Overall summary	This paper presents a systematic review of studies projecting coral reef futures under climate change scenarios and highlights gaps and areas for improvement for future research to be applicable to management. The authors undertook significant revisions to address reviewer comments in a previous round of review, which included a systematic review of studies. Their responses to reviewers were thorough and thoughtful, and the edits are well done. This study is interesting and important, and it points out areas that are really valuable for future research. In their response, the authors have noted areas in need of additional information, as well as inconsistencies among studies, which might be resolved by greater transparency and standardization in uncertainty measures. This was an interesting read, and I think it represents an important contribution to the field with urgent implications for ongoing and future research. The authors should be commended on their revisions and overall product. I recommend it for publication with minor revisions.	We thank Reviewer 4 for their feedback and positive assessment of our work. Reviewer 4’s constructive critique has been invaluable in shaping the manuscript. We have responded to Reviewer 4’s remaining comments in a point-by-point manner below.
Specific comment 1, Abstract	I would recommend including more information on the meta-analysis component (8 studies) and highlighting major findings from this.	As suggested by all reviewers, we have rewritten the Abstract to include the major findings of our systematic review and meta-analysis. The abstract now reads (P1-2, Lines 13-31) “Climate change impact syntheses, such as those by the Intergovernmental Panel on Climate Change (IPCC), consistently assert that limiting global warming to 1.5°C is unlikely to safeguard most of the world’s coral reefs. This prognosis primarily stems from ‘excess heat’ threshold models, which assume that

widespread coral bleaching predictably occurs when temperatures accumulate beyond a specific threshold. Our systematic review of research projecting coral reef futures to climate change (n=79) revealed that 'excess heat' models constituted only one third (32%) of all studies but attracted a high proportion (68%) of citations in the field. We observed that, irrespective of the approach, most methods employed deterministic cause-and-effect rules rather than probabilistic relationships, impeding the field's ability to estimate uncertainties of coral reef futures. In attempting to assess the consistency of projected impacts, we aimed to identify common coral reef metrics under uniform emissions scenarios. However, disparate choices in metrics and emissions scenarios hindered a cohesive synthesis and limited the exploratory analysis to a small fraction of available studies. We found significant discrepancies in expected impacts to coral reefs, suggesting that 'excess heat' models may project more extreme impacts than other methods. Drawing on lessons from the field of climate change science, we propose that an IPCC ensemble-like approach to generating probabilistic projections for coral reef futures is feasible. Successful implementation will require improved coordination among modeling efforts to select common output metrics and emission scenarios, addressing existing geographical biases, among other gaps in current modeling efforts.”

Specific comment 2,
Introduction

The explanation of climate tipping points, global and regional elements at risk, and regional tipping elements is a bit confusing in the first paragraph and could use some rephrasing.

As suggested, we revised this paragraph of the Introduction provide an explicit explanation of the term 'tipping elements' and clarify the meaning of global and regional elements. The first paragraph now reads (P3, Lines 33-47): “Anthropogenic climate change is expected to propel components of the Earth’s system beyond climate tipping points (CTPs) that, once exceeded, are expected to induce self-perpetuating change and impacts across biophysical systems¹. These components, termed ‘tipping elements’, are considered to be essential in the functioning of the Earth system, contribute substantially to human welfare, or have immense value as unique features of the Earth system¹. A notable example is the projected dieback of the Amazon rainforest that could release gigatons of carbon into the atmosphere and accelerate global warming^{1,5,6}. Although the concept of CTPs has been subject to contention^{7,8}, a recent synthesis delivered a shortlist of nine global and seven regional elements at risk¹. Global tipping elements, such as the Amazon rainforest and West Antarctic Ice Sheet, refer to components spanning subcontinental scales that could alter the operation of Earth’s system¹. Regional tipping elements represent biospheres expected to exhibit self-sustaining feedback at confined scales that have the potential to occur synchronously across subcontinental scales, including for example, the simultaneous melting of alpine glaciers^{1,9}. Among the shortlisted elements at risk are warm-water coral reefs, which

		represent a regional tipping element susceptible to exceedance if global warming exceeds 1.5°C above preindustrial levels ^{1,7,10} .”
Specific comment 3, Lines 36-37	Expected is used 2x in this sentence	We revised this sentence to remove the duplicated word, which reads as follows (P3, Lines 33-35): “Anthropogenic climate change is anticipated to propel large components of the Earth’s system beyond critical climate tipping points (CTPs), initiating self-perpetuating change and impacts across biophysical systems ¹ .”
Specific comment 4, Line 43	Exceedance and exceed are used back-to-back in this sentence – consider rewording	We reworded this sentence, as suggested (P3, Lines 45-47): “Among the shortlisted elements at risk are warm-water coral reefs, which represent a regional tipping element vulnerable to exceedance if global warming surpasses 1.5°C above preindustrial levels ^{1,4,7} .”
Specific comment 5, Lines 41-43	This sentence is confusing and would benefit from a rewrite.	We revised the first paragraph to clarify the meaning of global and regional tipping elements and edited the text to avoid word repetition. The revised paragraph reads as follows (P3, Lines 33-47): “Anthropogenic climate change is anticipated to propel large components of the Earth’s system beyond critical climate tipping points (CTPs), initiating self-perpetuating change and impacts across biophysical systems ¹ . These components, known as ‘tipping elements,’ are distinguished by their significance in Earth’s system functioning, their substantial contributions to human well-being, or their unique value within the Earth’s system ¹ . A notable example is the projected dieback of the Amazon rainforest that could release gigatons of carbon into the atmosphere and accelerate global warming ¹⁻³ . Although the concept of CTPs has been subject to debate ^{4,5} , a recent synthesis delivered a shortlist of nine global and seven regional elements at risk ¹ . Global tipping elements, such as the Amazon rainforest and West Antarctic Ice Sheet, refer to components spanning subcontinental scales that could alter the operation of Earth’s system ¹ . Regional tipping elements represent biospheres expected to exhibit self-sustaining feedback at confined scales that have the potential to occur synchronously across subcontinental scales, including for example, the simultaneous melting of alpine glaciers ^{1,6} . Among the shortlisted elements at risk are warm-water coral reefs, which represent a regional tipping element vulnerable to exceedance if global warming surpasses 1.5°C above preindustrial levels ^{1,4,7} .”
Specific comment 6, Lines 45-47	(1st sentence): Improper grammar – revise	The revised sentence now reads (P3, Lines 49-58): “Low-latitude reefs, as some of Earth’s most biodiverse ecosystems ⁸ , have reached a critical juncture where further deterioration could compromise global food supply, coastline protection, economic revenue, and the livelihoods of up to one billion people ⁹⁻¹¹ .”

Specific comment 7, Line 47	Again, reference to regional tipping element needs more explanation in the beginning	We now provide a detailed explanation of regional tipping elements and provide an example. The text reads (P3, Lines 45-49): “Global tipping elements, such as the Amazon rainforest and West Antarctic Ice Sheet, refer to components spanning subcontinental scales that could alter the operation of Earth’s system ¹ . Regional tipping elements represent biospheres expected to exhibit self-sustaining feedback at confined scales that have the potential to occur synchronously across subcontinental scales, including for example, the simultaneous melting of alpine glaciers ^{1,9} .”
Specific comment 8, Lines 55-56	What do high and medium confidence refer to in the text?	We now clarify that the confidence levels used by the most recent CTP synthesis follow the same confidence rating system used by the IPCC and provide a supporting citation that explains, in detail, what these confidence levels refer to. The edited text reads as follows (P3-4, Lines 60-65): “The most recent CTP synthesis followed the same confidence rating system used by the Intergovernmental Panel on Climate (IPCC) ^{1,21} . It identified a CTP of 1.5°C (1 to 2°C, high confidence) for tropical coral reefs, with an estimated timescale of 10 years for dramatic change (with medium-confidence) ¹ . In high agreement with findings of the IPCC ^{22,23} , the synthesis cited several modelling efforts using similar ‘excess heat’ modelling approaches as the basis of the assessment ¹⁴⁻¹⁷ .”
Specific comment 8 Line 66	What does “among the closest elements” refer to? What are the other elements? Other ecosystems?	We agree with Reviewer 4 that this was unclear. We revised the following sentence to be more specific (P4, Lines 71-76): “The resulting CTP of 1.5°C (1 to 2°C) places warm-water reefs among the six elements at risk of exceeding their tipping points within the global warming range set by the Paris Agreement (1.5 to <2°C) ¹ . This finding aligns with the conclusions of the Working Group II’s contribution to the IPCC’s AR6 report ²² and raises concern over imminent impacts to marine biodiversity, human livelihoods, and the effectiveness of interventions to alleviate further coral reef degradation.” We also revised the first paragraph of the introduction to provide further explanation of global and regional tipping elements (P3, Lines 33-47).
Specific comment 9, Line 87	Projections used 2x in this sentence. Should second use be “studies” instead?	As suggested, we revised the sentence to remove the duplicated word (P5, Lines 94-96): “Despite the growing body of literature projecting coral reef futures and their prominent role in assessments of climate change impacts, a comprehensive evaluation of available projections is lacking.”
Specific comment 10, Figure 1	Figure 1: Interesting figure! Extra spaces in the legend text.	We thank Reviewer 4 for their feedback on Figure 1. There was an issue when we converted the figure to pdf, which affected the alignment of text in the legend. This has now been corrected.

Specific comment 11, What were the unique methodologies?

Line 228

We edited the text to describe the two methodologies in more detail. The revised section of text reads as follows (P11, Lines 245-253): “There are several other emerging approaches to assess coral reef vulnerability to climate change. In addition to the five major approaches outlined above, one study adopted a spatial modeling approach to project the combined effects of warming and sea-level rise on the future coral reef growth rates in the South China Sea (SCS)⁹⁸. Another integrated linear extension rates of corals from three different islands in the same region with future SSTs to forecast coral growth rates⁹⁹. One study used historical bleaching and sea surface temperature records to project future bleaching probabilities in the Indo-Pacific¹⁰⁰, while another regional-scale study focused on larval connectivity and identified conservation areas with lower risks of coral bleaching in the Amani Islands of southern Japan¹⁰¹.”

Specific comment 12, What is the Red List of Ecosystems? Can you explain here?

Line 233

As suggested, we now include an explanation of the Red List of Ecosystems and provide an example of how this standardized method has been applied to project future coral reef vulnerabilities. This revised section of text reads (P11, Lines 255-268): “Articles applying a standardized method for assessing the risk of ecosystem collapse, the International Union for Conservation of Nature (IUCN) Red List of Ecosystems (RLE), represent an especially promising method that seeks to harmonize a diversity of coral reef variables to classify ecosystem collapse risk¹⁰²⁻¹⁰⁵. The RLE offers a standardized classification system that utilizes thresholds for key variables to integrate diverse data, making it valuable for policy applications¹⁰⁶⁻¹⁰⁸. For instance, one investigation applying this approach to coral reefs in the western Indian Ocean used metrics measuring historical changes in ecosystem extent and ecosystem functioning to reveal varying levels of regional vulnerability to future ocean warming¹⁰⁵. These studies shed light on inconsistencies in coral reef monitoring methods, resulting datasets, and the interpretation of results¹⁰²⁻¹⁰⁵. In turn, they emphasize the need for improvements in the consistency of monitoring efforts and advocate for the development of a unifying framework to enable more conclusive risk assessments. There are at least four studies applying this method to coral reefs in the Caribbean^{102,103}, meso-America¹⁰⁴, and the western Indian Ocean¹⁰⁵.”

Specific comment 13, Can you explain “Red List of Ecosystem-based studies”

Line 326

In addition to the revised text on Page 11, Lines 255-268, which provides a detailed explanation of the IUCN RLE approach (detailed above), this sentence was edited to clarify the meaning of RLE-based studies (P17, Lines 416-419): “This recommendation is further emphasized by studies utilizing the IUCN RLE classification system to evaluate the risk of coral reef ecosystem collapse, which call for enhanced coordination in monitoring coral reefs and improved data quality and quantity¹⁰³⁻¹⁰⁵.”

Specific comment 14, Figure 3	Interesting figure! What does “variable continuous” refer to in the technique? Should this be continuous variable?	We thank Reviewer 4 for highlighting this error. Indeed, the legend of Figure 3 and Figure 4 should read “continuous variable”. We corrected this oversight and inserted new versions of Figure 3 and 4 on Pages 18 and 20, respectively.
Specific comment 15, Figure 4	It is not clear to me from the caption why the % coral cover is pulled out and not included as the other projection variables are.	We agree with Reviewer 4 that this was unclear. Negative values for habitat change and reef cells at risk indicate negative ecological impacts compared to a baseline of 0%, whereas positive values indicate a positive effect direction, such as projections estimating increased habitat availability. However, for % coral cover, positive values do not necessarily indicate a positive ecological effect. Percent coral cover estimates are represented by Red Bars in insets to avoid misinterpretation. We edited the caption of Figure 4 to explain why % coral cover estimates are presented in insets. The revised figure caption reads as follows (P20-21, Lines 451-): “Projection estimates utilized as the foundational data for the effect size analysis depicted in Figure 3. Baseline projection estimates and their uncertainties (± 1 standard deviation [std]) were sourced from published scenarios spanning the years 2010-2014, aligning with historical global warming levels of 0.86°C to 0.96°C. The future ‘Warming’ projection estimates ± 1 std were derived from end-of-century scenarios (2090-2100) categorized by future warming levels: 1.5 - 2°C, 2 - 4°C, and >4°C. Negative values for habitat change and reef cells at risk signify adverse ecological impacts compared to a baseline of 0% (no effect), while positive values indicate a positive effect direction, such as projections estimating increased habitat availability. For % coral cover, positive values do not necessarily denote a positive ecological effect, as they reflect absolute values of % coral cover rather than relative increases or decreases. Percent coral cover estimates are visually depicted by red bars and symbols in insets to prevent misinterpretation. A comprehensive list of individual scenario descriptions is available in Supplementary Table 3.”
Specific comment 16, Line 385	Why is decline in quotes? Also, given the prior discussion of how “coral reefs” are measured, it would be good to include here what the definition is in the IPCC (i.e., % coral cover).	The word decline is in quotes because the IPCC does not provide a definition as to the unit of these projections. Specifically, in some reports, the IPCC delivers these projections to refer to ‘corals’ and in other reports, the projections for refer to coral reefs. Unfortunately, we cannot provide a definition provided and edited the text to explain this (P21, 473-478): “According to the Summary for Policy Makers in the IPCC’s Sixth Assessment Report, coral reefs are expected to ‘decline’ by 70-90% at 1.5°C global warming (relative to pre-industrial levels), and the estimates exceed 99% at 2°C¹³⁹. While there is no formal definition of ‘decline’ in the context of projection statements within the IPCC reports, the

assessments integrated projections of the percentage of reef cells at risk, drawing from the findings of Schlessner et al. (2016) and Frieler et al. (2013) as foundational evidence.”

Specific comment 17, Caldwell et al. 2016 (<https://www.mdpi.com/2072-4292/8/2/93>)
Line 528 may be relevant here.

We thank Reviewer 4 for highlighting this study. We found several other modeling studies that serve as early warning systems for coral disease outbreaks. We updated the paragraph to acknowledge the existence of several modeling studies that function as early warning systems for coral disease outbreaks and highlight their potential utility in improving our projections of coral disease susceptibility under future climate change scenarios.

The revised paragraph now reads (P27, Lines 633-650): “A similar pattern exists for evaluating how pest species and disease will interact with climate change to shape the future of coral reefs. In our analysis, we found only two investigations that delved into the role of coral disease outbreaks in influencing coral reef futures under climate change (Supplementary Tables 1 & 4)^{35,43}, with one of these studies being limited to a simulation of a single reef. The global study highlighted that future warming is likely to heighten coral susceptibility to disease and identified specific locations where targeted management could be implemented⁴³. Although excluded from our analysis due to the absence of future climate change projections, numerous predictive models serving as early warning systems for coral diseases exist¹⁷⁴⁻¹⁷⁶. These early warning system models have identified crucial drivers of disease outbreaks in various regions, which could prove useful for refining existing models projecting coral disease outbreaks under future climate change scenarios. Similarly, we identified only one study that simulated the impact of a pest species in climate change scenarios for coral reefs (Supplementary Tables 1 & 4). This study assessed the potential effectiveness of management strategies in addressing outbreaks of CoTS and reducing cumulative impacts on the Great Barrier Reef⁵⁵. The urgency to address this area of uncertainty is underscored by the ongoing coral disease outbreak in the Gulf of Mexico, which poses a severe threat to coral reefs in the region^{177,178}. Disease outbreaks are becoming increasingly concerning, affecting not only coral reefs but also other marine life^{179,180}, highlighting the need for urgent attention and action.”

Specific comment 18, Semicolon at end of sentence should be colon
Line 615

This typo was corrected (see Page 30).

Specific comment 19, Please double check variance formula as written.
Line 641

We thank Reviewer 4 for highlighting this error. All formulae were checked and corrected, accordingly.
See P30-31, Lines 734-746.

Specific comment 20, Really thorough, clear, and well done.
Supplemental

We thank Reviewer 4 for this comment and previous recommendations, which significantly improved the supplemental information.

References:

- 1 Armstrong McKay, D. I. *et al.* Exceeding 1.5°C global warming could trigger multiple climate tipping points. *Science* **377**, eabn7950, doi:doi:10.1126/science.abn7950 (2022).
- 2 Kwiatkowski, L., Cox, P., Halloran, P. R., Mumby, P. J. & Wiltshire, A. J. Coral bleaching under unconventional scenarios of climate warming and ocean acidification. *Nature Climate Change* **5**, 777-781 (2015).
- 3 Couce, E., Irvine, P. J., Gregoire, L., Ridgwell, A. & Hendy, E. Tropical coral reef habitat in a geoengineered, high-CO₂ world. *Geophysical Research Letters* **40**, 1799-1805 (2013).
- 4 DeFilippo, L. B. *et al.* Assessing the potential for demographic restoration and assisted evolution to build climate resilience in coral reefs. *Ecological applications* **32**, e2650 (2022).
- 5 Gatti, L. V. *et al.* Amazonia as a carbon source linked to deforestation and climate change. *Nature* **595**, 388-393, doi:10.1038/s41586-021-03629-6 (2021).
- 6 Steffen, W. *et al.* Trajectories of the Earth System in the Anthropocene. *Proceedings of the National Academy of Sciences* **115**, 8252-8259 (2018).
- 7 Lenton, T. M. *et al.* Climate tipping points—too risky to bet against. *Nature* **575**, 592-595 (2019).
- 8 Gaucherel, C., Hély, C. & Moron, V. Tipping point interactions may also generate weakening cascades. (2022).
- 9 Huss, M. & Hock, R. Global-scale hydrological response to future glacier mass loss. *Nature Climate Change* **8**, 135-140, doi:10.1038/s41558-017-0049-x (2018).
- 10 Schellnhuber, H. J., Rahmstorf, S. & Winkelman, R. Why the right climate target was agreed in Paris. *Nature Climate Change* **6**, 649-653, doi:10.1038/nclimate3013 (2016).

REVIEWERS' COMMENTS

Reviewer #1 (Remarks to the Author):

In my original review I'd noted that much of the content of the paper wouldn't surprise most modellers of coral reefs, though setting out the use of various methods and over emphasis of 'impact frequency' metrics in global analyses was useful. There was originally a section that attempted to explore the issues around uncertainty and I'd commented that merely quantifying uncertainty is one thing but incorporating that uncertainty into the USE of models is currently lacking. This was identified as a novel area that the paper could contribute to. The first revision then removed the entire section on uncertainty. This has now been revised but again the focus is still on how to measure uncertainty, and the possibility of taking an ensemble approach.

There's nothing inherently wrong here and I clearly the authors do not wish to get into the question of 'how to use uncertainty' but to me that leaves the paper of limited novelty. I certainly see value in its publication.

My only other comment is that I feel there is too much emphasis on undermining the DHW approach. Reading it fresh, I did not get the sense that the analysis was all that balanced, despite raising this in the previous review. For example, complaining about the poor temporal resolution of SST data. Really? It's acquired twice a week - globally! I'm aware of its limitations and geographical variation in its apparent efficacy as a stress metric. But these limitations are well known. And until there is an alternative with a more compelling archive then we're stuck with it or at least an algorithm using the same dataset. A constructive approach might advocate that people adapt the use of the algorithm (or underlying data) to create appropriate stress metrics for their geography. Otherwise, we see a unilateral lack of acceptance of metrics even in areas where they might be appropriate.

Reviewer #2 (Remarks to the Author):

Nature communication 402170

The paper is further revised and improved. I have several issues around clarity and improved presentation that would benefit the paper listed below. This and cited paper 24 are useful reviews and a starting place to begin a new round of analysis of climate change and coral reefs. I see it as an important contribution.

L34 – self implies causation. Feedback may be a better term.

L37 – and is better than or, as all these things are important.

L46 – I would use marine rather than regional. Each region or province of the world with coral reefs are experiencing climate change differently. It is important to not treat coral reefs as a uniform region or ecosystem.

L56 – I see it as more synchronized than uniform.

L100 – unpack is a colloquialism. I would suggest disarticulate

L172 – cost effective but not very accurate. See this study.

Lee-Yaw, J. A., J. L. McCune, S. Pironon, and S. N. Sheth. 2021. Species distribution models rarely predict the biology of real populations. *Ecography* 44.

L260-266 – this section is not that clear to me, too much jargon. I thought they used a threshold model for various locations or countries. So, that this is more of a local downscaled application of the threshold model. My perspective is the threshold model they used had very little connection to the empirical coral cover data, is that the insight? That the model is not making good predictions of the coral cover?

L288 – defy might be too emotive a word, as in disobey or resist. I would just say that the study combined multiple sources of data, environmental variables, and analytic techniques (i.e. regression and association). It is interesting that the most empirical field based study (72) is unusual. Thus, much of the work to date is quite theoretical with a slow movement

from a simple theory to field-based empiricism.

L334 – many papers believe the problem is resolution of data but ignore the variable choice problem. I don't think the authors should repeat this too often after they summarize the problem in the text above. I see resolution as a distraction from model structure and variable choice.

Figure legends and text still need some work to be clear.

The first sentence in the legends of figure 3 and 4 should be very clear about how these two figures differ. Some words like foundational data, % variable, projected impacts, projected estimates are not well-defined words that are making these figures unclear to readers. In figure 4, is this the percent change from a baseline. Why not present the direction of the changes for the van Woesik is projecting in coral? It is confusing to have % coral cover, but I think the authors mean % change in coral cover and do not distinguish positive and negative change. I wonder about the spatial area of each point, global or some smaller area? Could the symbols be modified to distinguish global from regional predictions?

The Frieler et al. 2013 paper is an outlier in figure 3 but less so in figure 4. Is this because it shows a rapid decline for small warming? What about Maynard? Given that these papers are cited for most future projections for planetary boundaries, etc. assume it is built into the model by multiple assumptions. The paper might benefit from a bit more depth of analysis on this issue of why these papers show such high sensitivity and others do not. I suspect it is because all the spatial cells behave the same, so there is no variability, just a threshold decline, and the threshold decision determines the predictions. What specifically distinguished Teneva from Frieler? I think with better analytical writing these sections could be both shorter and clearer.

L496 - It might be worth some short text about the debate around using the extreme scenarios in most papers, as in the 4.5C. See: Burgess, M. G., J. Ritchie, J. Shapland, and R. Pielke. 2020. IPCC baseline scenarios have over-projected CO2 emissions and economic growth. *Environmental Research Letters* 16:014016.

L512 – not sure what a uniform metric is.

An interesting development in science is the requirement to deposit empirical data in public locations but not model results. Seems this might be a call for doing more of this sharing of model outputs for comparison purposes.

L523 – I am not sure what “meaning of projection metrics” is. Does this refer to functions or relevance to ecology or people? The examples given are good, but this introductory term could be clearer.

P551 – the empirical environmental data for most models is at around 5 to 10 km², so that can set a limit to what is achievable regardless of the statistical or dynamic approaches. I believe the problem is less of scale than finding and integrating the variables that have predictive strength. Consider a table that lays out the differences between statistical and dynamic downscaling as the text here is not very explicit about specific of the differences and yet this understanding is important for making good recommendations for improvement of modelling.

L570 – why the difference?

L575 – I assumed the authors were referring to ocean and not atmospheric models.

Figure 5 is a nice addition. I wonder if it would be easier to follow if a panel a was global and panel b was sub global.

S610 – remember that most readers are quite aware of these shortcomings of models. So, I would keep it brief.

Reviewer #4 (Remarks to the Author):

I am satisfied with the authors' responses to my previous comments and the resulting revisions. I appreciate their thoroughness in addressing other reviewers' comments as well.

LOCATION	COMMENT	RESPONSE AND ACTION(S)
Reviewer 1		
Overall Comment, Part I	In my original review I'd noted that much of the content of the paper wouldn't surprise most modelers of coral reefs, though setting out the use of various methods and over emphasis of 'impact frequency' metrics in global analyses was useful. There was originally a section that attempted to explore the issues around uncertainty and I'd commented that merely quantifying uncertainty is one thing but incorporating that uncertainty into the USE of models is currently lacking. This was identified as a novel area that the paper could contribute to. The first revision then removed the entire section on uncertainty. This has now been revised but again the focus is still on how to measure uncertainty, and the possibility of taking an ensemble approach.	We extend our thanks to Reviewer 1 for their constructive critique of our manuscript, which has greatly improved the article and the recommendations therein. We now explicitly address the question of how to use uncertainty in models projecting coral reef responses. To do so, we first explain the various sources of uncertainty in the models. We then explain that most of the models in our database represent deterministic approaches that are inherently limited in their ability to directly incorporate uncertainty. While probabilistic models may be deemed more appropriate for the purpose of generating probabilistic statements of coral reef futures, the field currently lacks data to support their reliability and utility. We therefore focus on methods to account for, and use, uncertainty caused by model inputs in deterministic models. We discuss two main approaches: uncertainty analyses (UA) and sensitivity analyses (SA) and provide examples of how this can be conducted from the coral reef literature. Given that both of these approaches rely on a substantial number of models runs that can be burdensome, we also discuss techniques to reduce model runs and provide specific examples. We then link this text to our recommendation for a multi-model ensemble approach by explaining that such model sensitivity analyses (UA & SA) offer ways to estimate and use uncertainty caused by model's input values and parameters within an individual study, but it is possible to address system and model uncertainty using a multi-model ensemble approach. Excerpts from this new text read: (P13, Lines 353-359): "Uncertainties in coral reef projections stem from various sources that compound in the steps involved in generating the projections¹²². The sources range from variations in the climate system that impact various modeling tools, such as General Circulation Models (GCMs)^{122, 123}, to uncertainties in how future socio-economic policies and technologies will affect future emissions trajectories¹²⁴. Uncertainties related to the models themselves pertain to the model structure and parameter settings used^{122, 125}, which both rely on knowledge of the specific physical and ecological processes affecting how coral reefs will respond in the future."

(P14-15, Lines 373-399): “The question of how to account for uncertainty of deterministic models poses a significant challenge. Uncertainty associated with the model structure, specifically uncertainty about the cause-and-effect relationships, is often difficult to quantify because this requires the comparison of model outputs with real-world observations¹²⁷. However, it is possible to evaluate and then use uncertainty caused by the model’s input values and parameters. The probable range of model outputs can be examined by analyzing how these outputs behave when model input values are changed within plausible ranges^{127, 130}. Some studies have evaluated how choices in model inputs and different assumptions affect the outputs of coral reef models (e.g., ^{55, 72, 130, 131, 132, 133}), though differences in model outputs are used to produce formal estimates of uncertainty arising from model inputs. One approach to doing this is to conduct a formal uncertainty analysis of different model outputs. A straightforward method for conducting such an analysis is by applying Monte Carlo methods, where variations in model inputs are drawn randomly, and the resulting model outputs are treated as a random sample of the model output distribution¹²⁷ (e.g., ^{55, 130}). Although effective in helping to incorporate uncertainty into deterministic models, this approach requires a substantial number of model-runs.

Another approach is to apply a sensitivity analysis – a common method to understand how changes in input values and/or parameters of a model affect its output¹²⁷. Essentially, these analyses aim to pinpoint the input parameters to which the model output is most sensitive. For example, if plausible changes in an input parameter value induce large variations in the model output, this indicates that the parameter value is highly uncertain. Conversely, if model outputs remain stable, the analysis will indicate that the parameter value has low uncertainty. These analyses can become computationally expensive when all possible parameter values and their interactions are tested in a step-wise manner. However, there are techniques to reduce the number of model-runs^{127, 130}. For instance, sensitivity analyses have been applied to coral reef models by testing only the highest and lowest plausible values of the biological parameters and adjusting single parameter values by $\pm 10\%$ ^{130, 131}.”

Overall Comment,
Part II

There's nothing inherently wrong here and I clearly the authors do not wish to get into the question of 'how to use uncertainty' but to me that leaves the paper of limited novelty. I certainly see value in its publication.

As stated above, we now discuss how to evaluate and use uncertainty caused by the model’s input values and parameters, which returned some themes from earlier drafts (see Pages 14-15). We thank for Reviewer 1 for acknowledging the value of our work.

Overall Comment,
Part III

My only other comment is that I feel there is too much emphasis on undermining the DHW approach. Reading it fresh, I did not get the sense that the analysis was all that balanced, despite raising this in the previous review. For example, complaining about the poor temporal resolution of SST data. Really? It's acquired twice a week - globally! I'm aware of its limitations and geographical variation in its apparent efficacy as a stress metric. But these limitations are well known. And until there is an alternative with a more compelling archive then we're stuck with it or at least an algorithm using the same dataset. A constructive approach might advocate that people adapt the use of the algorithm (or underlying data) to create appropriate stress metrics for their geography. Otherwise, we see a unilateral lack of acceptance of metrics even in areas where they might be appropriate.

We acknowledge Reviewer 1's concerns about an unbalanced discussion of degree-heating metrics. We revised text throughout the manuscript to address this and added text to the "How Heat Stress is Modelled" section to acknowledge that threshold metrics remain vital for established programs forecasting bleaching and briefly review the work that has been done to optimize the operational DHW algorithm, acknowledging papers that show marked improvements in hit rates. We thank Reviewer 1 for their nice suggestion to advocate that researchers consider adapting the use of algorithms to create tailored stress metrics for their geography. We take this point one step further, but suggesting that such customized algorithms could then be confidently applied to regional projection models for their focal geography.

This new section reads as follows (P12-13, Lines 323-335): "While thermal threshold metrics have acknowledged limitations, they remain vital for established programs forecasting coral bleaching risk using satellite-based products. Work has already been done to test how well different degree heating algorithms explain coral bleaching patterns at local, regional, and global scales in an effort to improve their efficacy (e.g., ^{112, 113, 114, 115}). New configurations of the operational DHW algorithm hold promise in improving their ability to predict instances of observed bleaching^{112, 113, 116}, although the extent to which adapted algorithms improve predictability depends on the focal region and spatial scale of the test. This suggests that researchers could consider adapting different degree heating algorithms to pinpoint the most appropriate stress metric for their geography. Subsequently, these customized algorithms could be confidently applied to projection models for the focal region. Several studies in our database have shown how threshold choice affected their model outputs^{26, 40, 46, 112, 117, 118}. For example, one study reported divergent estimates of bleaching onset timing when different inter-annual variation thresholds were used¹¹⁷.

"

We also provide some examples where text was changed throughout the manuscript to ensure a balanced argument.

E.g. 1. We revised the abstract to clarify that a small subset of 'excess heat' threshold models applying degree-heating metrics may project more severe consequences. (P2, 18-32): "Climate change impact syntheses, such as those by the Intergovernmental Panel on Climate Change, consistently assert that

limiting global warming to 1.5°C is unlikely to safeguard most of the world's coral reefs. This prognosis is primarily based on a small subset of available models that apply similar 'excess heat' threshold methodologies. Our systematic review of 79 articles projecting coral reef responses to climate change revealed five main methods. 'Excess heat' models constituted one third (32%) of all studies but attracted a disproportionate share (68%) of citations in the field. Most methods relied on deterministic cause-and-effect rules rather than probabilistic relationships, impeding the field's ability to estimate uncertainty. To synthesize the available projections, we aimed to identify models with comparable outputs. However, divergent choices in model outputs and scenarios limited the analysis to a fraction of available studies. We found substantial discrepancies in the projected impacts, indicating that the subset of articles serving as a basis for climate change syntheses may project more severe consequences than other studies and methodologies. Drawing on insights from other fields, we propose methods to incorporate uncertainty into deterministic modeling approaches and propose a multi-model ensemble approach to generating probabilistic projections for coral reef futures."

E.g., 2. In this context, we omitted the discussion of efficacy concerns to avoid redundancy and overemphasis. (P11, Lines 287-293): "Our analysis revealed that more than half of all the studies (53%) employed thermal threshold techniques as the primary basis for their projections (Supplementary Data 2). Besides the exclusive use of this method in 'excess heat' threshold models, around 40% of population dynamic and eco-evolutionary models also relied on thresholds as the basis for their projections (Figure 2). Across the five major approaches (Figure 2), the most common threshold metrics applied were degree heating weeks (DHWs) or months (DHMs), which calculate values representing both the intensity and duration of heat stress events in singular metrics."

Reviewer 1 is correct that global-scale sampling of SST is sampled at impressive time-windows. We believe that further improvements in temporal resolution will be critical for future modelling of marine heatwave characteristics, such as rates of heating, especially in the context of increasing reports of rapid coral mortality over recent years. Although we consider this point significant, we agree to omit it this statement since it would require a greater word count to address it appropriately. Indeed, this point is not the focus of our manuscript and we understand Reviewer 1's perspective on the text as it was written.

Overall Comment	The paper is further revised and improved. I have several issues around clarity and improved presentation that would benefit the paper listed below. This and cited paper 24 are useful reviews and a starting place to begin a new round of analysis of climate change and coral reefs. I see it as an important contribution.	We thank Reviewer 2 for their time, constructive critique and insights, which have greatly strengthened the overall merit of our manuscript.
Specific Comment 1, Line 34	Self implies causation. Feedback may be a better term.	As suggested, we edited the sentence to replace the term “self-perpetuating” with “feedback-driven” so as not to imply causation. The sentence now reads (P3, Lines 34-36): “Anthropogenic climate change is anticipated to propel large components of the Earth’s system beyond critical climate tipping points (CTPs), initiating feedback-driven change and impacts across biophysical systems ¹ .”
Specific Comment 2, Line 37	And is better than or, as all these things are important.	As suggested, we replaced the word “or” with “and”. The sentence now reads (P3, Lines 36-38): “These components, known as 'tipping elements,' are distinguished by their significance in Earth's system functioning, their substantial contributions to human well-being, and their unique value ¹ .”
Specific Comment 3, Line 46	I would use marine rather than regional. Each region or province of the world with coral reefs are experiencing climate change differently. It is important to not treat coral reefs as a uniform region or ecosystem.	We agree with Reviewer 2’s point that coral reefs should not be considered a uniform region or ecosystem. However, after careful consideration, we have decided to retain the term ‘regional tipping element’ to be consistent with the paper cited and the definitions therein. We realized that the way the sentence was written was confusing and have now reworded Lines 46-28 to clarify that the term regional element refers to the shortlist and terms used by Armstrong et al (2022). ¹ The revised sentence reads as follows: (P2, Lines 39-41) “Among the shortlisted regional elements at risk are warm-water coral reefs, which are deemed vulnerable to exceedance if global warming surpasses 1.5°C above preindustrial levels ^{1,4,7} .”
Specific Comment 4, Line 56	I see it as more synchronized than uniform.	To avoid double use of the term synchronize in the same sentence, we replaced the word “uniform” with “concomitant”. (P3, Line 57) “Although coral bleaching is regarded as a localized process, near-synchronous bleaching events on many of the world’s reefs have occurred as a result of concomitant increases in ocean temperatures across the tropics ¹³ .”

Specific Comment 5, Line 100	Unpack is a colloquialism. I would suggest disarticulate	As suggested, we replaced the word “unpack” with “disarticulate”. The revised sentence reads (P5, Lines 101-103): “We also disarticulate how lessons from the field of climate change science can provide pathways for improving coordination of modelling efforts towards greater certainty in projections of coral reef futures.”
Specific Comment 6, Line 172	Cost effective but not very accurate. See this study. Lee-Yaw, J. A., J. L. McCune, S. Pironon, and S. N. Sheth. 2021. Species distribution models rarely predict the biology of real populations. Ecography 44.	We thank Reviewer 2 for highlighting this important study. We revised the following text to acknowledge the findings of Lee-Yaw and colleagues (P7, Lines168-172): “This approach permits large-scale projections that consider multiple physical parameters, even with limited field sampling ^{63, 64, 65} , making SDMs cost-effective tools. Although the widespread adoption of SDMs amplifies their value for comparing a diverse range of responses across marine and terrestrial ecosystems ^{66, 67} , a recent systematic review showed that SDMs may have significant limitations in accurately predicting the biology of real-world populations ⁶⁸ .”
Specific Comment 7, Lines 260-266	This section is not that clear to me, too much jargon. I thought they used a threshold model for various locations or countries. So, that this is more of a local downscaled application of the threshold model. My perspective is the threshold model they used had very little connection to the empirical coral cover data, is that the insight? That the model is not making good predictions of the coral cover?	We appreciate Reviewer 2's feedback and acknowledge the confusion in the original paragraph. We edited the text to clarify that the paper uses a standardized classification system that assigns thresholds to harmonize different reef variables. The paper uses an ecosystem model to estimate the risk of ecosystem collapse using this approach. We clarify that our central point is not related to the alignment of estimates of ecosystem collapse risk with empirical coral cover data, but rather issues in data scarcity, which resulted in important parameters being excluded from the ecosystem model discussed. We thus conclude that more conclusive risk assessments requires improved consistency in monitoring efforts and the development of a unifying framework across regions and countries. The revised paragraph reads as follows (P10, Lines 252-264): “A standardized method for assessing the risk of ecosystem collapse, the International Union for Conservation of Nature (IUCN) Red List of Ecosystems (RLE), represents an emerging method ^{101, 102, 103, 104} . The RLE offers a standardized classification system that utilizes thresholds for key variables to integrate diverse data ^{101, 105, 106} . One study applying this method used various coral reef datasets to model interactions within Western Indian Ocean reefs under future warming ¹⁰⁴ . The study reported varying levels of regional vulnerability to ecosystem collapse, ranging from ‘critically endangered’ to ‘vulnerable’ across the 11 eco-regions examined ¹⁰⁴ . However, the ecosystem model's assessment excluded data on fishing pressure and rates of sedimentation, among other variables, due to data scarcity across countries and regions. There are at least four studies applying this method to coral reefs in the Caribbean ^{101, 102} , meso-America ¹⁰³ , and the western Indian Ocean ¹⁰⁴ .”

		Together, they emphasize the need for improvements in the consistency of monitoring efforts and advocate for the development of a unifying framework to enable more conclusive risk assessments.”
Specific Comment 8, Line 288	Defy might be too emotive a word, as in disobey or resist. I would just say that the study combined multiple sources of data, environmental variables, and analytic techniques (i.e., regression and association). It is interesting that the most empirical field-based study (72) is unusual. Thus, much of the work to date is quite theoretical with a slow movement from a simple theory to field-based empiricism.	We revised this section of text according to Reviewer 2’s suggestion (P11, Lines 283-285): “Only one study in our database could not be classified as using either technique. The study integrated various data sources, environmental variables, and analytic techniques, including regression and association methods for projecting future coral cover ⁷² .”
Specific Comment 9, Line 334	Many papers believe the problem is resolution of data but ignore the variable choice problem. I don’t think the authors should repeat this too often after they summarize the problem in the text above. I see resolution as a distraction from model structure and variable choice.	We agree and have revised this section of text to exclude the discussion of resolution problems, avoiding distraction from the main message. As advised, issues with resolution are now addressed solely on pages 21-22. The revised text reads as follows (P12, Lines 312-321): “We found only one study projecting impacts on coral reefs that directly compared model outputs derived from both thermal threshold and continuous variable techniques ⁷² . This study revealed that a DHW-based model projected more severe declines in coral cover in the Indian ocean compared to the multivariate approach that integrated variables characterizing historical and future patterns of stresses. The findings suggested that patterns of acute and chronic stresses could be more influential than cumulative heat stress in predicting future coral cover in certain regions ⁷² , further highlighting the importance of variable selection procedures in the modelling process. Although there is substantial uncertainty in how climate change will morph future thermal regimes, global databases of marine environmental data provide many useful exposure and modifying variables for this purpose ^{72, 111} .”
Specific Comment 10, Figure 3 & 4	Figure legends and text still need some work to be clear. The first sentence in the legends of figure 3 and 4 should be very clear about how these two figures differ. Some words like foundational data, % variable, projected impacts, projected estimates are not well-defined words that are making these figures unclear to readers. In figure 4, is this the percent change from a baseline. Why not present the direction of the changes	We originally used % coral cover (not % coral cover change) because van Woesik et al. 2018 presented their projections in units of absolute percent coral cover, and these data were used for the effect size analysis. However, we realize that this adds an unnecessary layer of confusion. As suggested, we have revised Figures 3 & 4 to present % coral cover change and now indicate the geographic scale of the projections. Additionally, we have modified the figure captions for Figures 3 & 4 to explicitly state the difference between the two figures. The revised figure captions read as follows:

for the van Woesik is projecting in coral? It is confusing to have % coral cover, but I think the authors mean % change in coral cover and do not distinguish positive and negative change. I wonder about the spatial area of each point, global or some smaller area? Could the symbols be modified to distinguish global from regional predictions?

Figure 3 Comparative effect-size analysis of projected impacts on coral reefs among a small subset of available studies and three warming scenarios. Calculated mean effect sizes (Hedges' $g \pm 95\%$ CIs) represent the magnitude of projected impacts on model outputs (i.e., coral reef metrics) across three global warming scenarios (1.5 °C - 2 °C, 2 - 4 °C, and >4 °C). Model outputs (mean \pm 1std) used in this analysis were extracted from $n=39$ individual modelled scenarios across eight published studies, and represented in Figure 3. Mean effect sizes were derived from differences between projected estimates of coral reef metrics for the end-of-century (2090-2100) and the baseline period (2000-2015) (cf. Methods). Hedges' g , a common effect size metric ranging from $-\infty$ to $+\infty$, signifies no impact at zero, positive values indicate ecological benefits, and negative values signify adverse effects. The 95% CIs represent variability among scenarios within each study and warming scenario (Supplementary Data 3). Analyzed coral reef metrics include percent reef cells at risk (black), percent habitat change (blue), and percent coral cover change (red). Circles and triangles denote studies using thermal threshold and continuous variable techniques for modeling heat stress, respectively. Open symbols represent global-scale projections, while closed symbols denote regional-scale projections. Source data are provided as a Source Data file.

Figure 4 Percent changes in coral reef metrics representing the model outputs used in the analysis of Figure 3. Percent change in mean estimates (± 1 standard deviation) of model outputs (i.e., coral reef metrics) used in the analysis presented in Figure 3. Model outputs were extracted from $n=39$ modelling scenarios across eight published studies and converted into percent change for ease of interpretation. Mean estimates of coral reef metrics for historical global warming levels of 0.86°C to 0.96°C represent the baseline period of the years 2000-2015. Mean estimates of coral reef metrics categorized into for future warming scenarios of 1.5 - 2°C, 2 - 4°C, and >4°C represent projections at the end-of-century (years 2090-2100). Negative values for percent reef cells at risk (black), percent habitat change (blue), and percent coral cover change (red) signify adverse ecological impacts compared to a baseline of 0% (no effect), while positive values indicate a positive effect direction, such as projections estimating increases in reef cell habitat availability. Circles and triangles denote studies using thermal threshold and continuous variable techniques for modeling heat stress, respectively. Open symbols represent global-scale projections, while closed symbols denote regional-scale projections. Supplementary Data 3 provides a comprehensive list of individual scenario descriptions. Source data are provided as a Source Data file.

Specific Comment 11,
Figure 3 & 4

The Frieler et al. 2013 paper is an outlier in figure 3 but less so in figure 4. Is this because it shows a rapid decline for small warming? What about Maynard? Given that these papers are cited for most future projections for planetary boundaries, etc. assume it is built into the model by multiple assumptions. The paper might benefit from a bit more depth of analysis on this issue of why these papers show such high sensitivity and others do not. I suspect it is because all the spatial cells behave the same, so there is no variability, just a threshold decline, and the threshold decision determines the predictions. What specifically distinguished Teneva from Frieler? I think with better analytical writing these sections could be both shorter and clearer.

As recommended, we revised this section to provide a more in-depth analysis of Figures 3 &4 while ensuring conciseness. Specifically, we discuss the reasons why papers, including Frieler et al. and Maynard et al., were outliers in Figure 3. We also contrast the approach of Teneva et al. with other threshold studies to suggest that certain aspects of their method probably resulted in more modest and variable projections. We further show that studies applying the same methods and assumptions to global scale projections of reef cells at risk (e.g., Frieler et al. & Schleussner et al.) generate similar results and are often selected as a basis for IPCC and CTP assessments. Throughout this section of text, we relate the effect sizes back to the end-of-century estimates reported by the individual studies and conclude that there is high variation in the expected outcomes for coral reefs.

Excerpts from this revised section read (P18-19, Lines 494-541)

“...Among the threshold studies in the 2 -4 °C scenarios, Teneva et al.¹¹⁷ produced a relatively small effect size, aligning with the study’s less severe and more variable projections of reef cells at risk (Figures 3 & 4). The projections by Teneva et al.¹¹⁷ cannot be easily compared with other threshold studies reviewed here because of various methodological differences. In contrast to the other studies¹⁴, which applied global temperature thresholds to estimate future bleaching frequencies, Teneva et al.¹¹⁷ used bleaching observations from Reef Base to test prediction methods in which thermal thresholds were determined by historical SST variability. Accounting for historical climate experience might explain why the projections by Teneva et al.¹¹⁷ deviated from most other threshold studies in the analysis (Figure 4). Importantly, Teneva et al.¹¹⁷ also defined reef cells at risk as grid cells characterized by at least a 50% probability of experiencing 5-year mild or severe bleaching events by 2100.”

“...However, the study by Maynard et al.⁴² generated a notably smaller effect size than Frieler et al.¹⁴ and Anthony et al.⁵⁵ (Figure 3). Given that the effect sizes were based on relative differences between the baseline and end-of-century scenarios and weighted for variance, this discrepancy may be explained by the more severe and variable baseline impacts modelled by Maynard et al.⁴² (Figure 4). The relatively large effect size for Frieler et al.¹⁴’s projections occurred because of the absence of any variance with the study’s drastic projections of reef cells at risk (-100%, ± 0 Std) (Figure 4).”

“...The main reason for the high coherence between Schleussner et al.¹⁵ and Frieler et al.¹⁴ is the minimal differences in their approaches to modeling the frequency of bleaching events across global reef cells.

Both articles used the same model type and made analogous assumptions, including their selection of global thermal thresholds and frequency of heating events expected to impede reef recovery. Overall, there are several factors that may explain the high variation in expected outcomes for coral reefs. In contrast to Schleussner et al.¹⁵ and Frieler et al.¹⁴, differences in model types, model parametrization, and assumptions are likely important factors explaining differences in the extent of expected impacts.”

Given the numerous factors affecting the distribution of effect sizes, we also revised the preceding text to provide a better overview of how the analysis was done. Specifically, we now explain how the model outputs were aligned to the baseline period of 2000-2015 and then mapped to the three future scenarios, based on the categorization of emissions scenarios.

This section of text reads (P17, Lines 479-490):

“Figure 3 illustrates mean effect sizes representing the direction and magnitude of expected impacts on coral reef metrics across the selected studies. The distribution of effect sizes among the studies is influenced by a combination of factors: (1) varying assumptions on key drivers, such as choices in future emissions scenarios, (2) methodological differences, including the choice of simulated coral reef metric and the type and parametrization of the model, and (3) how the results were reported, such as the number of, and agreement among individual scenarios within each study. To help disentangle these factors, we aligned model outputs to baselines years between 2000-2015 (0.86-0.96 °C) and three end-of-century warming scenarios (1.5 – 2 °C, 2 – 4 °C, and >4 °C), which categorized the various emissions scenarios used (Supplementary Data 3). We further categorized each study based on whether it employed a thermal threshold or continuous variable technique in modeling heat stress and whether it presented global or regional scale projections (Figure 3).”

Specific Comment 12,
Line 496

It might be worth some short text about the debate around using the extreme scenarios in most papers, as in the 4.5C. See: Burgess, M. G., J. Ritchie, J. Shapland, and R. Pielke. 2020. IPCC baseline scenarios have over-projected CO₂ emissions and economic growth. *Environmental Research Letters* 16:014016.

We thank Reviewer 2 for this nice suggestion. We included a brief paragraph discussing the use of extreme scenarios in most studies and new evidence that such scenarios diverge from observed global CO₂ trends.

The paragraph reads as follows (P16-17, Lines 450-464): “There is currently no formal consensus on the most suitable emissions scenarios for modeling coral reef futures. While the number and type of emissions scenarios varied, the most frequently used scenario in our database was RCP8.5 (CMIP5), representing a high-emission scenario of ~4.5°C global warming by the end of the 21st century

(Supplementary Data 2). Most studies applied two emissions scenarios, typically comparing RCP8.5 (CMIP5) with a scenario of lower radiative forcing such as RCP2.6 or RCP4.5 (CMIP5) (Supplementary Data 2). Though subject to debate, recent analyses show that observed trends in global CO₂ emissions are substantially lower than those simulated by high-emission baseline scenarios such as RCP8.5 (CMIP5)^{142, 143, 144, 145}. These studies suggest that this divergence could widen throughout this century and conclude that such scenarios should no longer serve as reference high-emission scenarios^{142, 143, 144, 145}. Given these developments and the release of the IPCC's 6th Assessment Report (AR6)²², there is a pressing need for coordination to select the common emissions pathways for modeling coral reef futures. This urgency is underscored by the introduction of new socioeconomic pathways representing novel levels of radiative forcing (1.9, 3.4, 7.0 W m⁻²), already incorporated into recent projections for coral reefs (e.g.,^{17, 146})."

Specific Comment 13, Not sure what a uniform metric is.
Line 512

We agree this was unclear and replaced the term "uniform metrics" with "the same units" to clarify. The revised sentence now reads (P23, Lines 583-586): "While all studies in our database provided ample data to facilitate interpretation of the study outcomes, most (89% of studies) failed to report basic metrics for model outputs or sufficient extractable data for measures of variation to be converted into the same units."

Specific Comment 14 An interesting development in science is the requirement to deposit empirical data in public locations but not model results. Seems this might be a call for doing more of this sharing of model outputs for comparison purposes.

We thank Reviewer 2 for this nice recommendation and edited the text in two sections to make this recommendation.

(P20, Lines 550-558): "In many cases, challenges arose from the display of results in figures and geographical maps that were not accompanied by adequate supplemental information reporting extractable values. While there is an increasing emphasis on depositing empirical data into online repositories (e.g., Dryad, Figshare, and Zenodo), this is rarely required for model outputs. Recognizing the necessity for reporting and metadata availability standards, other fields focused on projecting climate change impacts to biological systems have implemented agreed-upon standards^{66, 67, 134}. For instance, the International Union for Conservation of Nature (IUCN) established preliminary reporting standards for species threat assessments based on SDMs¹⁵¹, which have been further refined in subsequent publications^{66, 67}."

(P21, Lines 580-583): "In summary, establishing and adhering to standards for the comprehensive reporting and communication of projections, including associated uncertainties, would facilitate more

conclusive syntheses of coral reef projections in the future. This may also involve setting standards for publishing metadata.”

Specific Comment 15,
Line 523

I am not sure what “meaning of projection metrics” is. Does this refer to functions or relevance to ecology or people? The examples given are good, but this introductory term could be clearer.

We edited the text to clarify that there is a lack of clarity over the ecological or biological meaning of the projected variables.

The revised text reads as follows (P20-21, Lines 563-583): “However, a lack of clarity over the ecological or biological meaning of the projected variable and the exact outcomes anticipated for coral reefs constrains the usefulness of projections in guiding effective-decision making, management, and conversation efforts.”

While the vast majority explicitly define the metrics simulated, some earlier studies provide indistinct descriptions (Supplementary Data 2). For instance, several ‘excess heat’ threshold models simulate the frequency of severe bleaching events to deliver projections as the proportion of coral reef cells (e.g., 1° x 1° grid cells on the Earth’s surface) at risk of ‘long-term degradation’ or ‘severe bleaching events’^{14, 15, 16, 17}. However, there is presently no agreed nomenclature of such states for coral reefs, raising uncertainty as to their exact meaning and the consequences involved. In some cases, subjective terms affect the communication of projections in influential assessments of climate change impacts, where terms like ‘losses of coral reefs’¹⁵², ‘corals being lost’²³, and ‘coral reefs at risk’²³ are used interchangeably without accompanying definitions. These terms could, in theory, be understood to imply a range of outcomes for coral reefs, ranging from reductions in live coral to the ecological collapse of entire reef ecosystems. Clear and well-defined nomenclature is especially important to in the context executive summaries addressing policy-makers and other stakeholders. In summary, establishing and adhering to standards for the comprehensive reporting and communication of projections, including associated uncertainties, would facilitate more conclusive syntheses of coral reef projections in the future. This may also involve setting standards for publishing metadata.”

Specific Comment 16,
Line 551

The empirical environmental data for most models is at around 5 to 10 km², so that can set a limit to what is achievable regardless of the statistical or dynamic approaches. I believe the problem is less of scale than finding and integrating the variables that have

As recommended, we have incorporated a table delineating the main differences between statistical and dynamical downscaling procedures. The table outlines key advantages and disadvantages, helping readers to discern the most suitable technique based on their specific needs. Please see Table 1 on Page 44 of the

predictive strength. Consider a table that lays out the differences between statistical and dynamic downscaling as the text here is not very explicit about specific of the differences and yet this understanding is important for making good recommendations for improvement of modelling.

manuscript document, captioned “Advantages and limitations of statistical and dynamical downscaling procedures”.

We agree with Reviewer 2 that variable selection and predictive strength is a dominant problem. So as not to detract from the discussion of variable selection, we removed text addressing resolution on pages 13-14 (see also response Reviewer 2, Specific Comment 9). We now discuss resolution as an in issue solely in this section and edited the text to include information regarding new data streams becoming available, including higher-resolution satellites. This edited section reads as follows:

(P22, Lines 617-624) “Although these results suggest that dynamical downscaling may outperform statistical methods, further assessments of the relative costs and benefits of the two techniques are warranted (Table 1). Downscaling techniques, however, ultimately introduce an additional source of uncertainty. Fortunately, the spatial resolution of global models is expected to improve in the near term with the introduction of new data streams, including higher-resolution satellites (e.g., Himawari¹⁶⁰) coming online. This enhancement will improve sea surface temperature (SST) data resolution and reduce the reliance on downscaling approaches¹⁶¹.”

Specific Comment 17, Why the difference?
Line 570

As suggested, we now explain why projections from the statistical downscaling procedure could not detect the earlier onset of annual bleaching. The additional text reads as follows (P22, Lines 617-624): “While there was a high level of agreement between the projections produced by the two techniques, the dynamically downscaled model detected an earlier onset of annual severe bleaching linked to future changes in regional currents. In contrast, the statistical procedure failed to detect these changes due to its inability to capture local-scale features, such as eddies, which influence warming levels leading to coral bleaching¹⁵⁶.”

Specific Comment 18, I assumed the authors were referring to ocean and not
Line 575 atmospheric models.

Reviewer 2 is correct that this was an error, and the word ‘atmospheric’ was deleted. The revised sentences read as follows (P22, Lines 620-624): “Fortunately, the spatial resolution of global models is expected to improve in the near term with the introduction of new data streams, including higher-resolution satellites (e.g., Himawari¹⁶⁰) coming online. This enhancement will improve sea surface temperature (SST) data resolution and reduce the reliance on downscaling approaches¹⁶¹.”

Specific Comment 19, Figure 5	Figure 5 is a nice addition. I wonder if it would be easier to follow if a panel a was global and panel b was sub global.	We thank Reviewer 2 for their feedback and have split Figure 5 into two panels, as recommended. See Figure 5 caption on page 46 of the manuscript file.
		The caption of Figure 5 now reads: Figure 5 Association between major coral reef provinces and applied approaches used to project coral reef futures. (a) represents the distribution of modelling approaches used at a global-scale, and (b) represents the association between coral reef provinces and the main methodologies used. The specific flow width is proportional to the number of research articles applying each of the five main methods, while the numbers in parentheses indicate the total count of articles that generated projections for global reefs (a) or each reef province (b). See Supplementary Data 1 for a full description of the focal geographic regions for each study included in database (n=74). This diagram has been generated using the online tool: Visual Paradigm (https://online.visual-paradigm.com). Source data are provided as a Source Data file.
Specific Comment 20, Line 610	Remember that most readers are quite aware of these shortcomings of models. So, I would keep it brief.	We agree and edited the text to acknowledge that most readers are already aware of this issue and kept this section brief. The revised text reads (P23, Lines 646-651): “Coral reef scientists are increasingly aware of this issue. Addressing these gaps necessitates targeted efforts to enhance the resolution and accuracy of global-scale projections, while simultaneously expanding the scope and diversity of regional and local-scale projections and monitoring efforts. Such efforts are already underway and essential in providing decision-makers with actionable information to manage climate change impacts on coral reefs at global, regional, and local scales ^{167, 168.} ”
Reviewer 4		
Overall Comment	I am satisfied with the authors' responses to my previous comments and the resulting revisions. I appreciate their thoroughness in addressing other reviewers' comments as well.	We thank Reviewer 4 for their review of our work, which greatly improved the manuscript. We appreciate the time dedicated and their thoughtful suggestions throughout the review process.
Additional edits		
Minor fix to effect sizes in Figure 3		We identified a small coding error that led to the presentation of negative effect sizes for Cacciapaglia et al. 2015 and 2018 in Figure 3, whereas they should have been positive. Importantly, this coding oversight did not alter the text or affect any of the study's conclusions. We have rectified the error, conducted a thorough double-check of all analysis components, and can confirm that Figure 3 is now accurate.

References:

1. Armstrong McKay DJ, *et al.* Exceeding 1.5oC global warming could trigger multiple climate tipping points. *Science* **377**, eabn7950 (2022).